# Analysis of medieval burials from Ibiza reveals genetic and pathogenic diversity during the Islamic period

Ricardo Rodríguez-Varela [1,2] ✉, Zoé Pochon [1,2], Alex Mas-Sandoval[3], Reyhan Yaka [1,2], Cesar A. Fortes-Lima [4,5], Almudena García Rubio[6], Nicholas Márquez-Grant[7,8], Juanjo Marí[9], Glenda Graziani[10], Antoni Ferrer Abárzuza[11], Mário Vicente[1,2], Lander Lorca-Francisco [1], Anna Linderholm [1,12], Vendela K. Lagerholm[1,2], Lara R. Arauna [13], Patxi Pérez-Ramallo [14,15,16], Maja Krzewińska [1,2], Carina M. Schlebusch [5,17,18] & Anders Götherström [1,2] ✉

Ibiza, an island in present-day Spain, was conquered in 902 CE by the Umayyad Emirate of Córdoba. The island remained under Islamic rule until 1235. Here, we analyse the genetic and metagenomic profiles of 13 individuals from an Islamic cemetery in Ibiza, dated to 950–1150 CE. Genome-wide analyses reveal heterogeneity, with ancestry components from Europe, North Africa, and Sub-Saharan Africa. Our analyses estimate that North African gene flow occurred two to seven generations before these individuals lived, suggesting admixture following the Islamic conquest of Iberia and potentially on Ibiza itself. Notably, two individuals trace their Sub-Saharan origins to distinct regions, Senegambia and present-day southern Chad, providing direct evidence of trans-Saharan connections via military and slave networks documented in contemporary Arabic sources. Metagenomic analyses detect several pathogens in this community, with one individual carrying *Mycobacterium leprae*, offering insight into the presence of leprosy in Ibiza. Our findings align with the historically documented two-pulse demographic model, indicating an initial settlement following the early tenth-century conquest and a second influx associated with Almoravid movements in the twelfth century. These securely dated genomes offer insights into medieval population dynamics and health in the Balearics.

Medieval Iberia was a crossroads of diverse cultures, religions, and ethnicities, shaped by centuries of migration, trade, and conquest. The Islamic expansion of the early 8th century marked the establishment of al-Andalus, a region that, under the Emirate and Caliphate of Córdoba (756–1031 CE), fostered significant advancements in science, art, and intellectual thought[1,2]. The demographic composition of Iberia expanded even further during this period, with Imazighen (sing. Amazigh; often referred to as "Berbers") forming the majority of new arrivals alongside Arab elites, as well as sporadic groups of Sub-

Saharan Africans and Slavic peoples, many of whom were likely mercenaries or enslaved individuals[3,4]. These groups contributed to a diverse and, in some periods, pluralistic society that encompassed the existing Hispano-Roman population, already fused with the Visigoths, as well as long-established Jewish communities.

After the fragmentation of the Córdoba Caliphate in the 11th century, al-Andalus splintered into smaller *Taifa* Kingdoms, where alliances often transcended religious boundaries, reflecting the complex and dynamic social structure of medieval Iberia[1,2]. Two Amazigh

empires, the Almoravids (1040–1147 CE), originating from what is now Mauritania and the Western Sahara, and later the Almohads (1121–1269 CE), who controlled the north-western coast of Africa, unified and took control of al-Andalus after the fall of the Umayyad dynasty[1,2].

The island of Ibiza (*Eivissa*), part of the Balearic archipelago (present-day Spain), was integrated into the broader Islamic Iberian realm in 902 CE by 'Isam al-Khawlâni (Fig. 1a), and remained under Islamic rule until 1235 CE. The colonisation, led primarily by Imazighen clans with participation from some Arab groups, occurred between 902 and 940 CE, spanning scarcely more than a single generation[5,6]. Following the collapse of the Caliphate of Córdoba, Ibiza came under the control of the Taifa of Dénia, which established a more stable state presence on the island. By the 12th century CE, one can already speak of a *madina* (urban settlement or fortified town), albeit a modest one[7]. Situated along key Mediterranean trade and migration routes, the island soon became a hub of active cultural exchange.

The archaeological site located on Bartomeu Vicent Ramon street corresponds to a sector of the "*Maqbara of Madina Yabisa*", a main urban Muslim cemetery. The cemetery was discovered during construction work, and a subsequent rescue excavation uncovered 125 individual burials dated between 925 and 1150 CE. The graves, consisting of simple earth pits (darih type), align with Islamic funerary law prescribing unadorned burials, with the exception of one burial (UE 153)[8], which yielded personal ornaments (two silver rings) (Fig. 1b)[9]. Most individuals were placed on their right side, facing southeast toward Mecca, and the cemetery shows evidence of prolonged use, with occasional overlapping burials (Fig. 1b). Osteological assessment indicates a demographically mixed population, with both sexes and all age groups represented, good preservation, and limited evidence of trauma or skeletal pathologies[9,10]. This funerary context, together with the analysis of the human remains, offers valuable insights into the region's demographic diversity during the Islamic period. Previous archaeological and isotopic analyses revealed complex cultural and dietary practices, as well as evidence of mobility and non-local origins among the individuals buried there[11,12]. These findings align with archaeological syntheses indicating that Ibiza before 902 CE was likely very sparsely populated at the time and is scarcely mentioned by Andalusi writers, suggesting that the newcomers encountered largely uninhabited areas during this initial Islamic migration wave[6,7]. A second demographic pulse reached the Balearic Islands following the Almoravid conquest of Mallorca in 1115-1116 CE[13].

The advent of genome-wide analyses and genotype imputation techniques[14,15] has enabled the reconstruction of diploid genomes and haplotypes, providing more precise insights into population history and revealing the genetic landscapes of past societies. Genetic studies have shown that North African ancestry has left a lasting impact on modern Iberian populations[16,17], with evidence of a significant increase of gene flow from North Africa during the medieval period[18,19]. In parallel, ancient metagenomics provide valuable insights into the health and disease environments of past societies. Pipelines such as HOPS[20], aMeta[21] or the combination of mapping followed by ngsLCA[22] and bamdam (https://github.com/bdesanctis/bamdam) enable the identification of microbial communities and ancient pathogens in archaeological human remains. Previous studies have documented pathogens including *Mycobacterium leprae*, *Variola virus*, *Yersinia pestis*, Hepatitis B virus and human parvovirus B19 in historical populations[23–27]. These genomic and metagenomic approaches provide new opportunities to explore genetic ancestry and health in past populations, as well as potential links between human mobility and the spread of infectious diseases.

Previous population genetic studies on Ibiza have focused either on modern populations or on ancient DNA (aDNA) samples dating back to the Bronze Age and Phoenician period[28–31]. However, no archaeogenetic studies have examined the medieval Islamic period, despite scarce written sources and its historical importance. Our study directly addresses this gap by generating genomic data from medieval Islamic Ibiza. Access to individuals from this period is essential, as reconstructing it from modern populations is particularly challenging due to the substantial genetic transformation that followed the 1235 CE conquest of Ibiza by the Crown of Aragon and the subsequent settlement of Catalan populations[30].

In this study, we investigate the genetic and metagenomic landscape of individuals from the medieval Islamic necropolis, "Maqbara of Madina Yabisa," in Ibiza. Our objectives are to: (i) characterize genetic ancestry components and admixture patterns, with a focus on the timing and origin of North African gene flow; (ii) explore kinship and consanguinity; (iii) assess pathogen presence; and (iv) combine genomic data with dated archaeological context and relevant historical background to investigate population history. We achieve this by analysing genome-wide shotgun sequencing data along with comprehensive modern reference datasets, followed by genotype imputation using a recently curated reference panel[32]. This approach enables haplotype-based analyses to investigate the timing and dynamics of admixture events and to resolve fine-scale ancestry components.

## Results
### Osteological and archaeological analysis
Permission was granted to study 41 individuals, but due to bad preservation and tooth availability, only 30 were sampled (Supplementary Data 1) from the 125 burials excavated at the Maqbara of Madina Yabisa (Ibiza, Spain). From these, sufficient autosomal DNA was successfully recovered from 13 individuals, enabling population genomic and metagenomic analyses (Fig. 1b). The majority of the graves are oriented (W-E) along an axis between 250° and 284° W, showing no fixed spatial order within the necropolis, which likely reflects its long-term and somewhat irregular use[9]. However, a group of ten graves stands out due to its distinct arrangement: they are aligned parallel to each other, evenly spaced, and oriented NW–SE, with the individuals facing toward the southeast (115°–137°). Among the thirteen individuals that yielded ancient DNA, two belong to this southeast-oriented group (s. 107 and s.103) (Fig. 1b). We present the estimated age at death and stature based on osteological examination of these 13 skeletons (see Methods). Our results indicate that all individuals were adults at the time of death, with estimated statures ranging from 145 cm to 176 cm (Supplementary Data 1). No grave goods, trauma or diagnostic skeletal pathology were observed in the sequenced individuals. However, the absence of certain elements, such as the facial bones in individual s.313, makes it difficult to rule out the presence of diseases like leprosy. For the genetic analyses, only teeth were sampled.

### General validation and radiocarbon dates
In the 13 individuals that yielded DNA, the human autosomal genome coverage ranged from 0.1× to 6.3× (mean: 1.05×; median: 0.53×). All libraries showed short fragment lengths, characteristic damage patterns, and low levels of mitochondrial and X-chromosome contamination (0–3% and 1–6%, respectively; Supplementary Data 1), supporting the authenticity and endogenous origin of the data.

Twelve of the thirteen individuals with sufficient DNA for genomic analyses were directly radiocarbon dated at the Tandem Laboratory, Uppsala University, to the 10th–12th centuries CE. Individual s.155, which yielded lower genomic coverage, was not dated (Supplementary Data 1 and 2). Because they derive from an island context where both direct consumption of marine resources and indirect intake via seaweed- or shell-fertilized crops were plausible, as previously suggested[11], a potential marine reservoir offset cannot be excluded[33]. To assess whether individuals s.117 and s.197, who have the most recent median radiocarbon dates (1099 and 1113 CE, respectively), might belong to the post-1115 CE Almoravid influx, we applied marine reservoir corrections to the radiocarbon ages of all individuals. These corrections were based on their estimated dietary contributions,

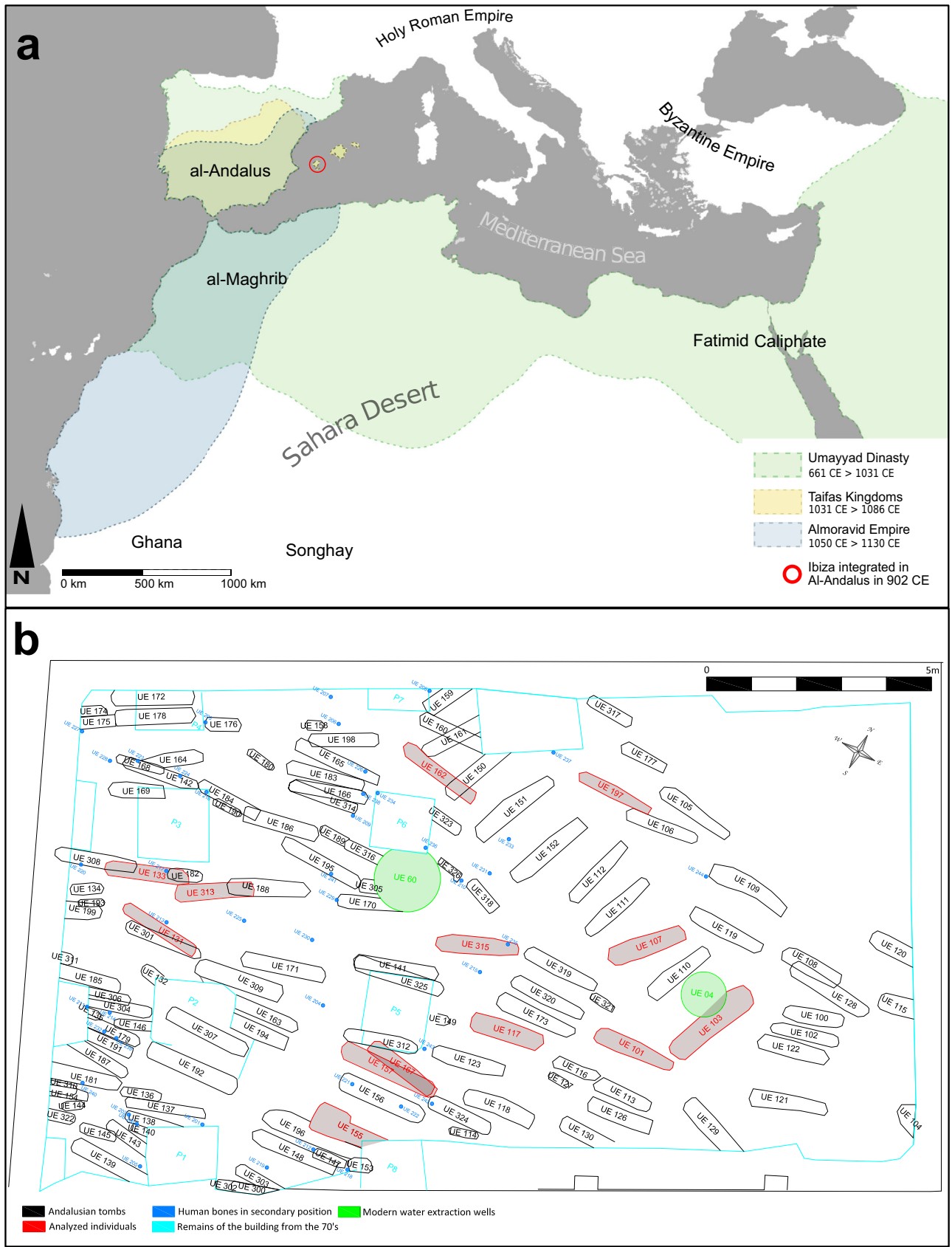

**Fig. 1 | Geographic context and excavation plan. a** Map of the Mediterranean and trans-Mediterranean Islamic domains in Iberia at their greatest extent. Sourced from refs. 85,175,176. **b** Plan of the rescue excavation at 33 Bartomeu Vicent Ramon Street, Ibiza town, corresponding to a sector of the maqbara (cemetery) of Madina Yabisa. The labels of the analysed burials are shown in red.

reconstructed from $\delta^{13}C$ and $\delta^{15}N$ values obtained from the same dentine collagen used for aDNA and radiocarbon analyses, using the Bayesian mixing model *simmr*[34] (see Supplementary Discussion and Supplementary Data 2). Taken together, these results indicate with high probability (>80% using the broader prior and >95% using the constrained prior) that both individuals postdate 1115 CE and thus likely belong to the period following the Almoravid influx (see Supplementary Discussion and Supplementary Fig. 1).

## Sex identification and uniparental markers

Four individuals were genetically identified as females (XX) and nine as males (XY) (Supplementary Data 1). The Y-chromosome haplogroup analysis reveals a diverse genetic landscape, reflecting the multi-ethnic environment of medieval Islamic Iberia (Supplementary Data 1). The haplogroup E1b1b1b1a1 is present in five individuals, while E1b1b1a1a1c2 is present in one of the individuals. Both haplogroups are common in North African Amazigh populations and are also present in Iberian populations[18,35-37], supporting the significant Amazigh influence in mainland al-Andalus and in particular, in the necropolis of the Island of Ibiza. Haplogroup E1b1a1a1a, found in one individual (s.117), is common in Sub-Saharan African populations, which suggests long trans-Saharan networks across North Africa. Other haplogroups, such as R1b1a1b1b3 found in one individual, are common in Western Europe[18], indicating a persistent presence of indigenous Iberian populations. In agreement with the Y-chromosome haplogroups, the mitochondrial haplogroups are also diverse, pointing to European, Middle Eastern, and North African origins (H1, J1, J2, and K1; Supplementary Data 1). Notably, we found haplogroups that are common in Sub-Saharan Africa in two individuals[31]: L3e1c (s.117) and L3b2 (s.197). No genetic kinship was detected among the individuals (see Methods); full summary statistics are provided in Supplementary Data 3.

## Uncovering global ancestry patterns

We projected the retrieved genomic data onto a worldwide population's Principal Component Analysis (PCA) using smartpca[38]. This first approximation to the data indicates a significant genetic diversity within the studied individuals (Fig. 2). Two individuals (s.157 and s.313) aligned within the PCA space associated with European populations (in blue), while eight individuals occupied intermediate positions between European, Middle Eastern, and North African populations. One individual overlapped closely with North African populations (in orange). Notably, two males (s.117 and s.197), were positioned within the PCA space corresponding to Sub-Saharan African populations (in red), in agreement with their uniparental markers (Supplementary Data 1).

We then projected all ancient individuals, excluding the two individuals with evidence of Sub-Saharan African ancestry, onto a PCA structured around three major modern reference groups: Europeans (blue), Middle Eastern (including the Caucasus region) populations (purple), and North Africans (red) (Fig. 3a and Supplementary Fig. 2). Most of the individuals share the PCA space with previously published Islamic-period Iberians from Spain. However, two individuals are positioned closer to both modern and ancient Europeans, suggesting absent or reduced levels of North African ancestry. In contrast, one individual occupies a PCA space near pre-European contact Canary Island populations, close to modern North African groups.

To further explore the genetic ancestry of these individuals, we performed genotype imputation on the ancient Ibiza samples alongside two controls: a pre-European contact individual from Tenerife (Canary Islands) (gun011[39]) representing a North African source, and a pre-Islamic early medieval individual from Las Gobas (Condado de Treviño, Burgos) in northern Iberia (ldo039, individual 26[23];) representing a local Iberian source. We then merged the imputed genomes with a combined reference panel from the HGDP[40] and the 1000 Genomes Project[41] (hereafter referred to as 1K-HGDP[32]), resulting in a diploid database containing 5.7 million SNPs. The results of a PCA

performed with this dataset closely mirrored those obtained from the pseudohaploid data with the Human Origins (HO) SNP panel, confirming the accuracy of the imputed datasets (Fig. 3b). The only exception was the individual with the lowest coverage, s.155, whose position differed between the two PCAs (Fig. 3) and had too low genome coverage (0.11x; Supplementary Data 1) for a reliable imputation[15].

To validate these results, we performed an unsupervised ADMIXTURE analysis (K = 2 to K = 9) using the pseudohaploid HO_1240K dataset (see Methods). At K = 7, the analysis revealed clear differentiation among North African, Middle Eastern, European, Asian, and various African populations, including distinct components for rainforest hunter-gatherer groups (RHG; e.g., Mbuti) versus East and West African non-RHG groups (Supplementary Fig. 3). To further investigate, we conducted a supervised ADMIXTURE analysis at the previously selected K = 7, using modern populations that maximized each of the seven ancestral components (see Methods), and including all relevant ancient reference individuals (Fig. 4). Both the unsupervised and supervised ADMIXTURE analyses yielded highly similar genetic ancestry profiles for both the modern and ancient Ibiza samples.

In the published ancient genomes, we observe the trajectory of North African ancestry (highlighted in yellow) over time in Iberian populations (Fig. 4). Notably, almost no North African ancestry is detected in Iron Age Iberians or north Iberian Visigoths, but it appears in significant proportions among individuals found in Roman sites in Iberia and in the Islamic-era individuals. The genetic ancestry proportions of the Ibiza individuals are consistent with the PCA results and previously published Iberian Islamic individuals[18], revealing that while all individuals carry both North African and European ancestry, the proportion of each varies between individuals. These results are confirmed using an unsupervised ADMIXTURE with our imputed samples and the 1K-HGDP dataset (Supplementary Fig. 4).

Individual s.107 exhibits the highest proportion of North African ancestry, consistent with the PCA results, and may represent an Amazigh individual. This interpretation is supported by the similarity in ancestry composition to pre-European individuals from the eastern Canary Islands, who also carried a small proportion of European ancestry in ADMIXTURE analyses (Fig. 4)[42,43]. This aligns with existing knowledge that even pre-Islamic North African populations already carried a small proportion of European and Levantine ancestry[44,45].

Interestingly, the two Sub-Saharan African individuals from Ibiza exhibit distinct ancestry profiles. Individual s.117 exhibits evidence of West African and East African ancestry (red and brown components, respectively; Fig. 4). However, this result may reflect the limited availability of more appropriate representative African populations in this reference panel. In contrast, individual s.197 shares most of his ancestry with West African populations. Olalde et al. (2019)[18] reported two Islamic-period individuals from Granada (southern Spain), dated to the 10th century CE (I7427, "Spain Islamic" in Fig. 4) and the 16th century CE (I3810, "Spain Late Islamic" in Fig. 4), who also exhibit a notable component of Sub-Saharan African ancestry, albeit at lower levels than those observed in the sample presented here.

## Local ancestry inference

To gain deeper insights, we applied haplotype-based inference methods using RFMix v1.5.4[43] based on the imputed dataset. We excluded individual s.155 due to low coverage and consequently unreliable imputation. This method infers local ancestry by segmenting each chromosome into windows and using a conditional random field (CRF) informed by random forests (RF) trained on reference panels[43]. We used two sets of reference populations depending on the genetic ancestry of the target individuals. For most individuals, we employed a reference panel representing Europeans and North Africans. For the two individuals with Sub-Saharan African ancestry, we used an additional reference panel comprising East and West African populations. All reference groups were drawn from the 1K-HGDP dataset. To

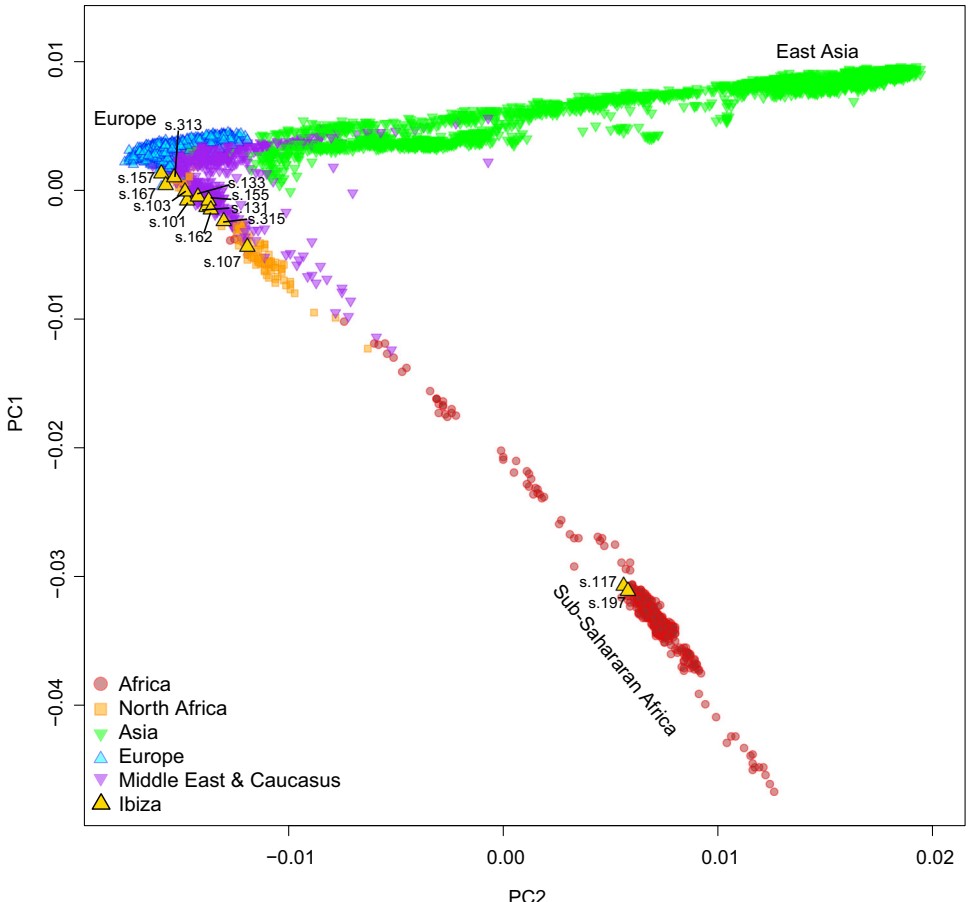

**Fig. 2 | Principal component analysis of medieval Iberian genomes.** Genomic data of the medieval individuals projected onto the first two principal components of modern worldwide populations from the Human Origins (HO) dataset.

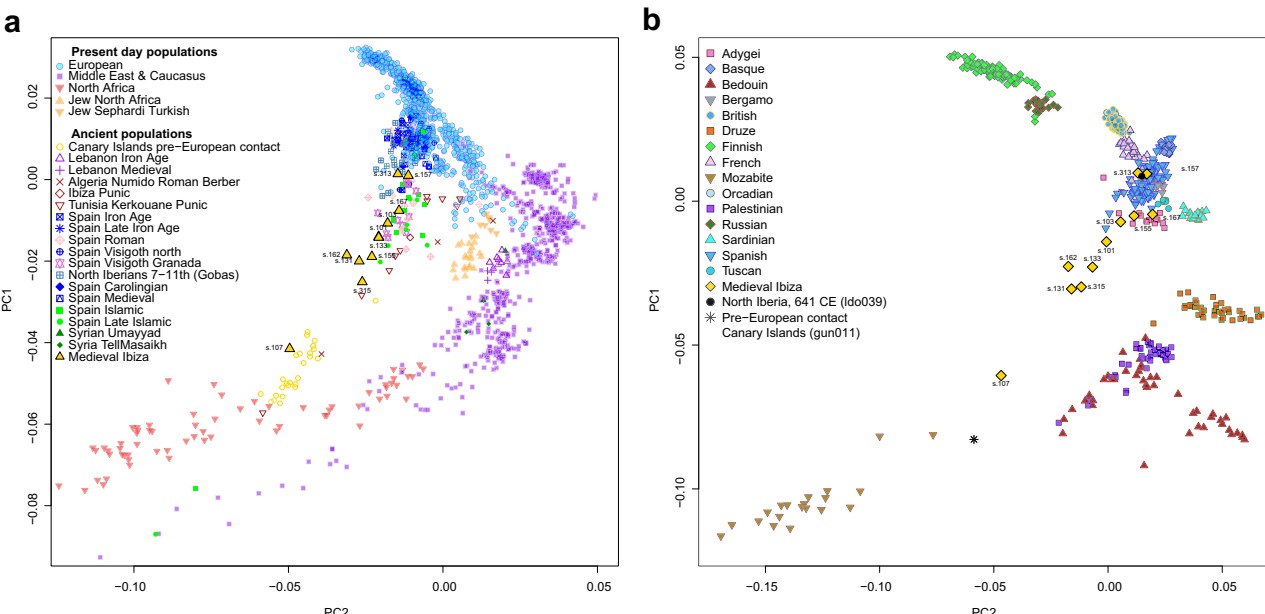

**Fig. 3 | Eurasian PCA with the ancient individuals from Ibiza (excluding the two Sub-Saharan African individuals). a** Ancient pseudohaploid individual from Ibiza, along with other ancient reference individuals, projected on a PCA based on the "1240 K + HO" dataset. **b** Imputed ancient DNA samples projected on a background of modern Eurasian populations, PCA based on the 1K-HGDP dataset.

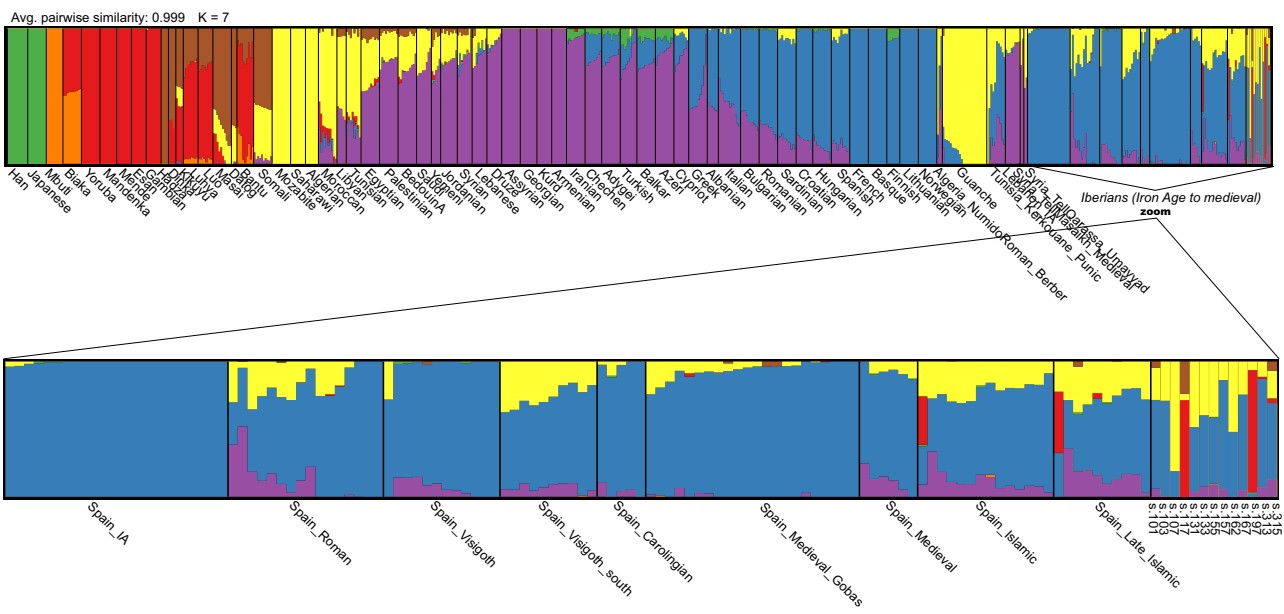

**Fig. 4 | Supervised ADMIXTURE analysis.** Results for $K = 7$ (best-supported solution; 10/10 runs) using the combined 1240 K + HO reference dataset and pseudohaploid ancient genomes. Inset: zoom-in on the ancient individuals, with ancient Iberian references ordered by estimated date.

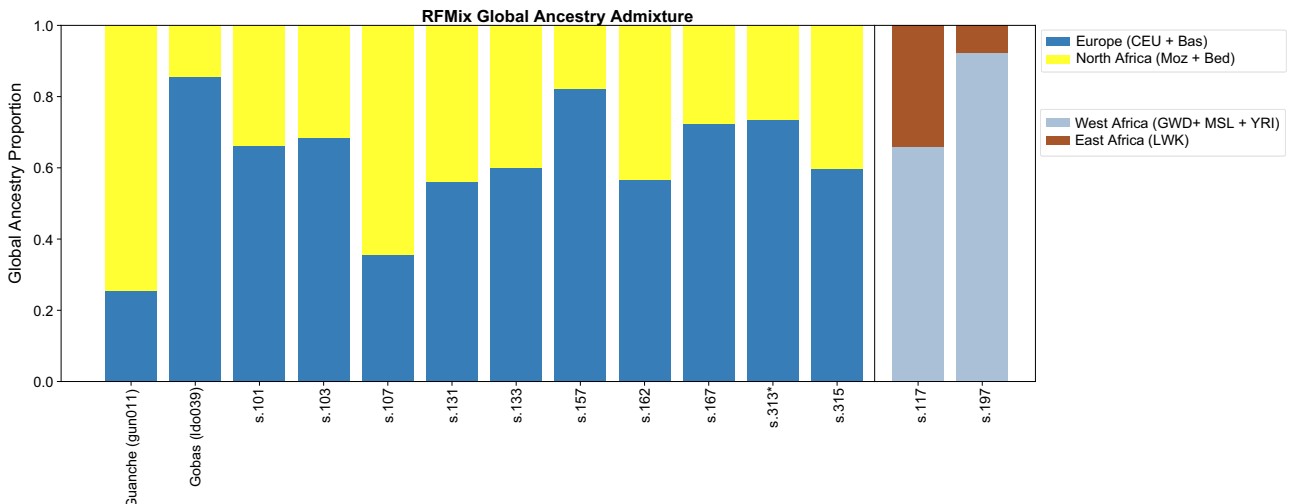

**Fig. 5 | Whole-genome RFMix analysis showing ancestry proportions for all individuals with sufficient data.** *Indicates a sample with fewer SNPs than the others, analysed separately.

represent European ancestry, we included CEU and Basque populations as proxies for pre-admixture Iberians, given that modern Iberians (IBS) already exhibit North African admixture. North African and Middle Eastern ancestries were represented by a combined sample of Mozabite ($n = 22$) and Bedouin ($n = 43$) individuals. East African ancestry was represented by the Luhya, the only reference population from East Africa available in this dataset. However, we acknowledge the limitations of this approach: since the Luhya are a Bantu-speaking group with predominantly West African ancestry and limited East African admixture, they are not fully representative of East African ancestry. Moreover, only four African populations were used in this analysis, selected from the limited Sub-Saharan African representation in the 1K-HGDP dataset based on PCA and ADMIXTURE results. In contrast, West African ancestry was represented by individuals from Yoruba, Gambia, and Sierra Leone, capturing a broader spectrum of West African genetic diversity.

The results corroborated the pseudohaploid analyses, providing greater resolution for the Sub-Saharan individuals (Fig. 5). Specifically,

individual s.117 exhibited a mix of Luhya and West African ancestries, while individual s.197 showed predominantly West African ancestry (Fig. 5).

RFMix results provide ancestry resolution at the chromosome level, enabling more precise insights into the timing and patterns of admixture. For example, individuals with similar overall proportions of European and North African/Middle Eastern ancestry (e.g., s.131 and s.133) carry long, uninterrupted chromosomal segments from both genetic ancestries, indicating recent admixture events. The presence of these extended ancestry tracts suggests that admixture occurred relatively recently, likely within the past few generations, as recombination has not yet had sufficient time to break down these segments into shorter fragments (Fig. 6).

**Timing of North African gene flow into Iberia**

To investigate the timing of the admixture observed between local Iberians and North African Imazighen in the medieval Ibiza individuals, we estimated admixture dates based on the decay of local ancestry

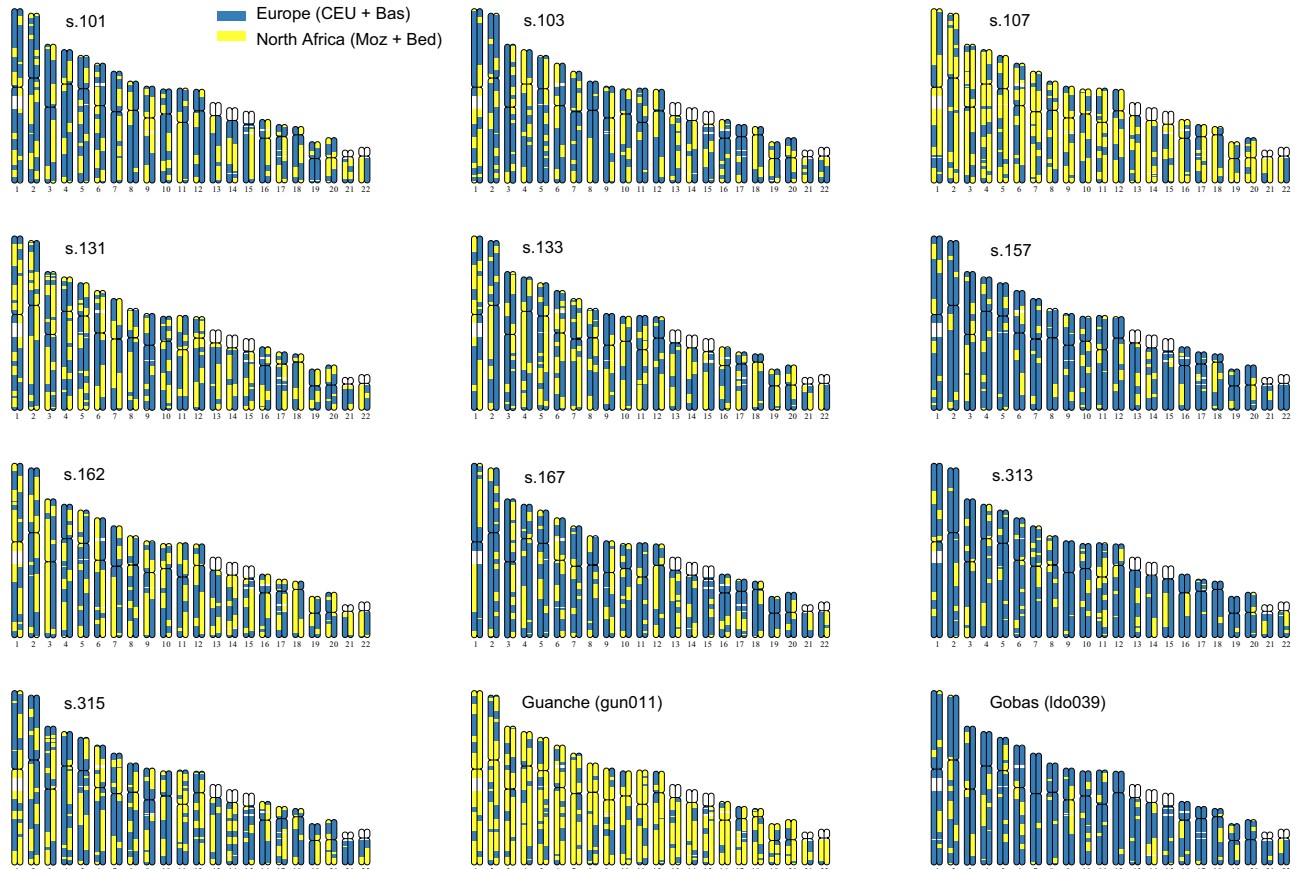

**Fig. 6 | Local ancestry painting (LAP) of the Ibiza individuals, shown alongside two ancient reference individuals (gun011 and ldo039).** Each panel represents one individual, with chromosomal segments coloured according to the inferred ancestry across the genome. Colours correspond to the reference populations used in the RFMix analysis. These profiles illustrate the mosaic structure of individual genomes and the varying degrees of admixture among them. Numbers on the x-axis indicate chromosomes 1 to 22.

covariance along the genome, using ancestry tracts inferred with RFMix. The analysis measures how the local ancestry assignments of two SNPs on the same haplotype covary as a function of genetic distance (more specifically, as a function of the probability of an odd number of recombination events between both SNPs). We fitted a covariance decay curve for each individual and a population-based curve grouping seven individuals with radiocarbon dates ranging from 1073 to 1084 CE (mean date: 1080 CE) (Fig. 7). We inferred an admixture date of 7.84 (95% CI: 7.63, 8.05) generations before 1080 CE (Fig. 7a) from the population-based curve (Fig. 7A). Using a generation time of 26.9 years[46], the admixture dates back to 869 CE (95% CI: 863.5, 874.8). Nevertheless, the individual-based inferred dates of admixture cover a range from 2.49 to 7.81 generations (Fig. 7b), pointing to a complex admixture process, implying continuous gene-flow or ancestry-related assortative mating. These results are compatible with substantial gene flow from North Africa into Ibiza after the island was incorporated into al-Andalus in 902 CE. In addition, the individual s.157 presents a differentiated individual-based admixture date, dating back to 519 CE (16.27 generations before the radiocarbon date of 957 CE). Interestingly, this individual shows an ancestry profile with high European ancestry similar to the Iberian pre-Islamic sample (ldo039) from mainland Spain (Las Gobas, Burgos)[23], which points to a minor and more ancient gene-flow from North Africa before the Islamic period, probably related to the demographic changes in the Mediterranean associated to the fall of the Roman Empire[17].

The local ancestry analyses using RFMix yielded results broadly consistent with the ADMIXTURE and PCA patterns obtained from three independent datasets (Human Origins, 1K-HGDP, and the extended Africa2 panel), supporting the robustness of our main ancestry inferences. However, we acknowledge that using Mozabites and Bedouins as the sole North African reference populations and proxies for medieval Berber and Arab populations entering into Iberia is a limitation. Mozabites genomes include both European and Near Eastern components. Consequently, some ancestry segments may have been misassigned as "European". In the ADMIXTURE analyses of this study using Human Origins and the 1K-HGDP datasets, the pre-European individual from Tenerife (Guanche, gun011) also shows small percentages of European ancestry (Supplementary Figs. 3 and 4), in agreement with the RFMix results (Figs. 5 and 6). We interpret these not as evidence of recent European admixture, but rather as reflecting either (i) ancient European-related ancestry already present in pre-European Canary Islanders[39,42], or (ii) reference bias due to shared European ancestry in Mozabites. Another plausible explanation, consistent with the first hypothesis, is that these small segments reflect ancient European gene flow into North Africa during earlier periods (e.g., the Roman era), which may have persisted in the ancestors of gun011 before their isolation in the Canary Islands. Thus, both biological and methodological factors could contribute to the apparent European signal. We therefore stress that our RFMix results should be read in comparative rather than absolute terms. Importantly, these potential mis-assignments do not critically affect our temporal inferences: tract-length patterns remain informative for admixture dating, as our approach, based on ancestry covariance decay, is less sensitive to local misclassification than methods relying directly on tract-length distributions.

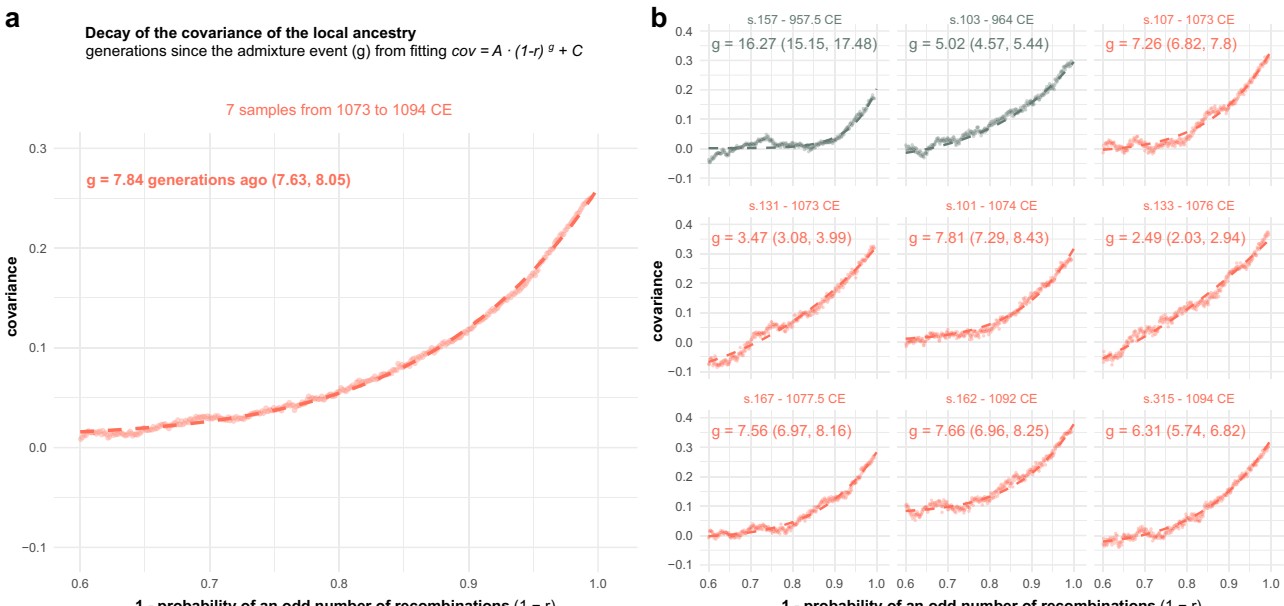

**Fig. 7 | Estimated admixture dates in generations since admixture.** Admixture dates inferred from the decay of local ancestry covariance with increasing genomic distance between SNPs on the same chromosome, using RFMix ancestry tracts. Each point represents the mean covariance for all within-chromosome SNP pairs within a discretized r window of 0.001. The dashed line is the curve fitting the equation cov=$A \cdot (1-r)^g + C$. The mean estimated g parameter and the 95% confidence interval, in brackets, are computed over 100 SNP resampling. Each plot shows the covariance decay and the fitted curve for one of the resamplings. **a** Population-level curve combining seven individuals with radiocarbon dates ranging from 1073 to 1084 CE. **b** Individual curves are shown separately for nine individuals, including the seven grouped individuals.

We confirmed the RFMix results of the two Sub-Saharan individuals from Ibiza using a PCA based on the imputed diploid genotypes, including modern Sub-Saharan reference populations from the 1K-HGDP diploid dataset. One individual, s.197, clustered with Gambian populations, while the s.117 positioned between East and West African groups (Supplementary Fig. 5). This is consistent with the RFMix and ADMIXTURE analyses, which suggest that this individual may have ancestral origins from populations located between Western and Eastern Africa, regions not represented in this limited dataset.

## Tracing Sub-Saharan African origins

To overcome the limited representation of African references in previously used datasets and to better investigate the genetic origins of the two Sub-Saharan individuals, we merged both the imputed and pseudohaploid data with SNP array data from two comprehensive reference panels (Africa1 and Africa2 datasets). This approach also enabled us to confirm the genetic affinities between the remaining individuals and North African populations.

The first panel, the Africa1 dataset, includes a broad range of African populations[47–60]. The second panel, Africa2 dataset, comprises Sub-Saharan Africans, Europeans, Middle Easterners, and North Africans, including Amazigh groups from Morocco[47,48,50,51,54,55,61–64]. Using these comprehensive datasets, we performed PCA, ADMIXTURE, and $f_3$-statistic analyses to investigate the genetic origins of the two ancient individuals with Sub-Saharan African ancestral origins.

The PCA and ADMIXTURE analyses suggest that s.197 is genetically close to Senegal Bedik and Gambian individuals speaking non-Bantu Niger-Congo languages, consistent with the results of the $f_3$-statistics (Fig. 8 and Supplementary Figs. 6–11). Note that for individual s.197, we detected small levels of contamination on the X chromosome (~6%), while no evidence of contamination was observed in the mitochondrial genome (Supplementary Data 1). This low-level contamination may account for the slight affinity of s.197 toward Eurasian populations observed in some analyses (Fig. 8d; Supplementary

Fig. 9b). On the other hand, s.117 shows stronger genetic affinity with present-day populations from southern Chad (Sara and Laal speaking a Nilo-Saharan language and an endangered isolate language, respectively), as evidenced by ADMIXTURE, PCA, and an MDS plot based on outgroup $f_3$-statistic (Fig. 8 and Supplementary Fig. 9a). These results indicate that s.117 most likely has a genetic origin in southern Chad, rather than representing recent admixture between West African and East African populations. The Africa2 dataset further confirms and refines, at the individual level, the genetic cline observed between European populations and Moroccan Imazighen in non-Sub-Saharan African individuals from Ibiza. In particular, individual s.107 clusters closely with Moroccan Imazighen, suggesting that the European ancestry detected in this individual (see Figs. 4 and 5) likely originates within the Afro-Asiatic-speaking Amazigh populations rather than from recent admixture with local Ibiza or Iberian individuals.

## Genetic consanguinity

Genetic consanguinity was assessed using the frequency and length of runs of homozygosity (ROH). We used the hapROH software[65], which detects ROH in individuals with coverage ≥ 0.3× using reference haplotypes, to estimate consanguinity in all ancient Ibiza individuals and comparative ancient Iberian individuals. This analysis was based on over 400,000 SNPs (the recommended threshold) from the 1240 K dataset[66], except for individual s.315, who had only 191,950 SNPs available; thus, results for this individual should be interpreted with caution (Supplementary Data 4).

We identified two individuals with high levels of genetic consanguinity (s.131 and s.315) despite the overall high diversity of ancestries in the group (Fig. 9 and Supplementary Fig. 12). The ROH pattern in s.131 is consistent with parental relatedness at the first-cousin level (Supplementary Fig. 12a). Combined with an admixture timing of 3.47 generations, this suggests that both parents were already part of the admixed local gene pool of Ibiza.

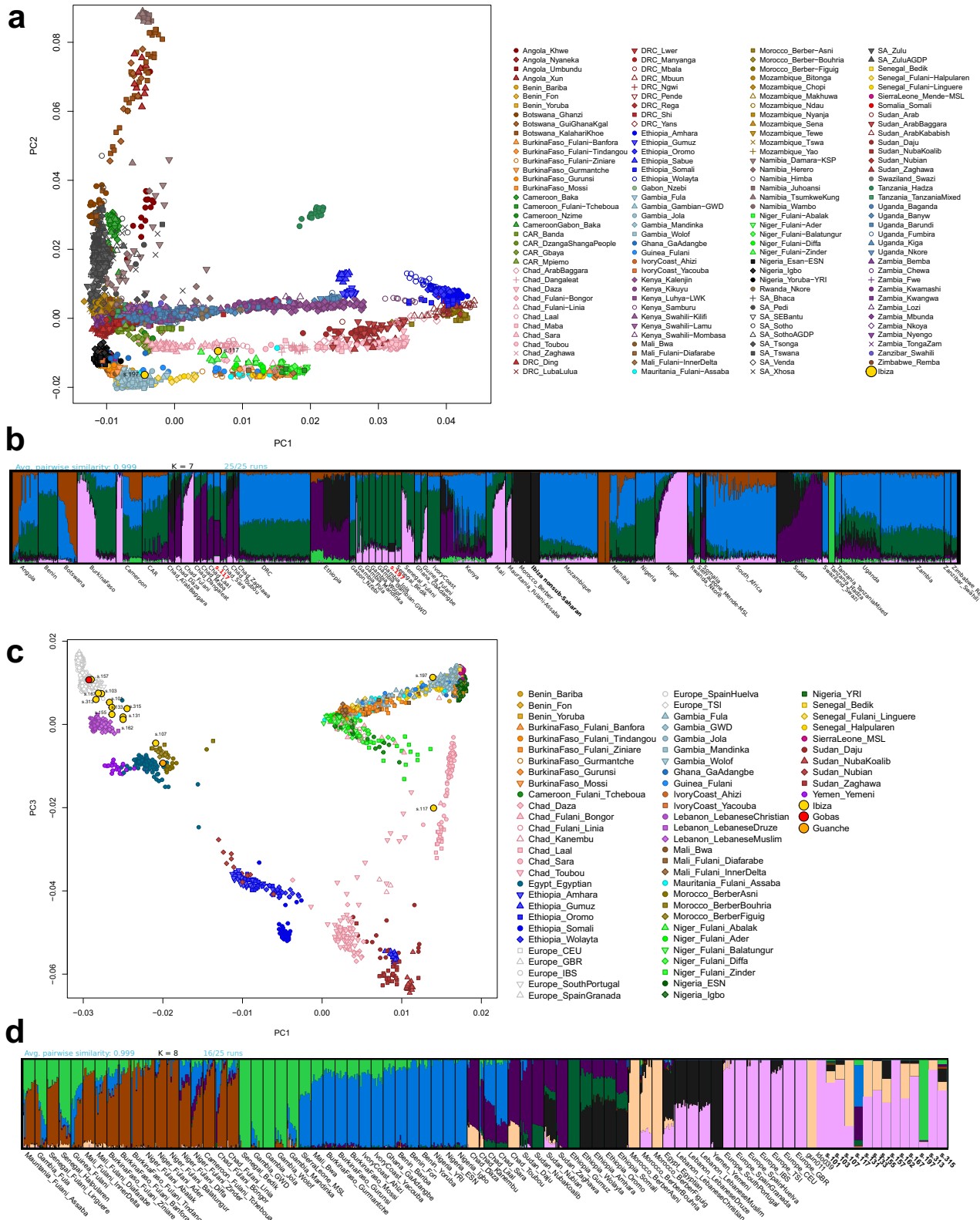

**Fig. 8 | Population structure of ancient Ibiza individuals in the context of Sub-Saharan African reference datasets.** The bars representing the ancient samples are visually enlarged to improve clarity, as each represents a single individual. **a** and **c**. Principal Component Analysis (PCA) showing the projection of ancient pseudo-haploid Ibiza samples onto modern African genetic variation. **a** uses the Africa1 dataset, while **c** uses Africa2 dataset. **b** and **d** Unsupervised ADMIXTURE analysis of the same ancient Ibiza samples alongside modern individuals from the respective reference datasets. **b** corresponds to Africa1, and **d** to Africa2 dataset.

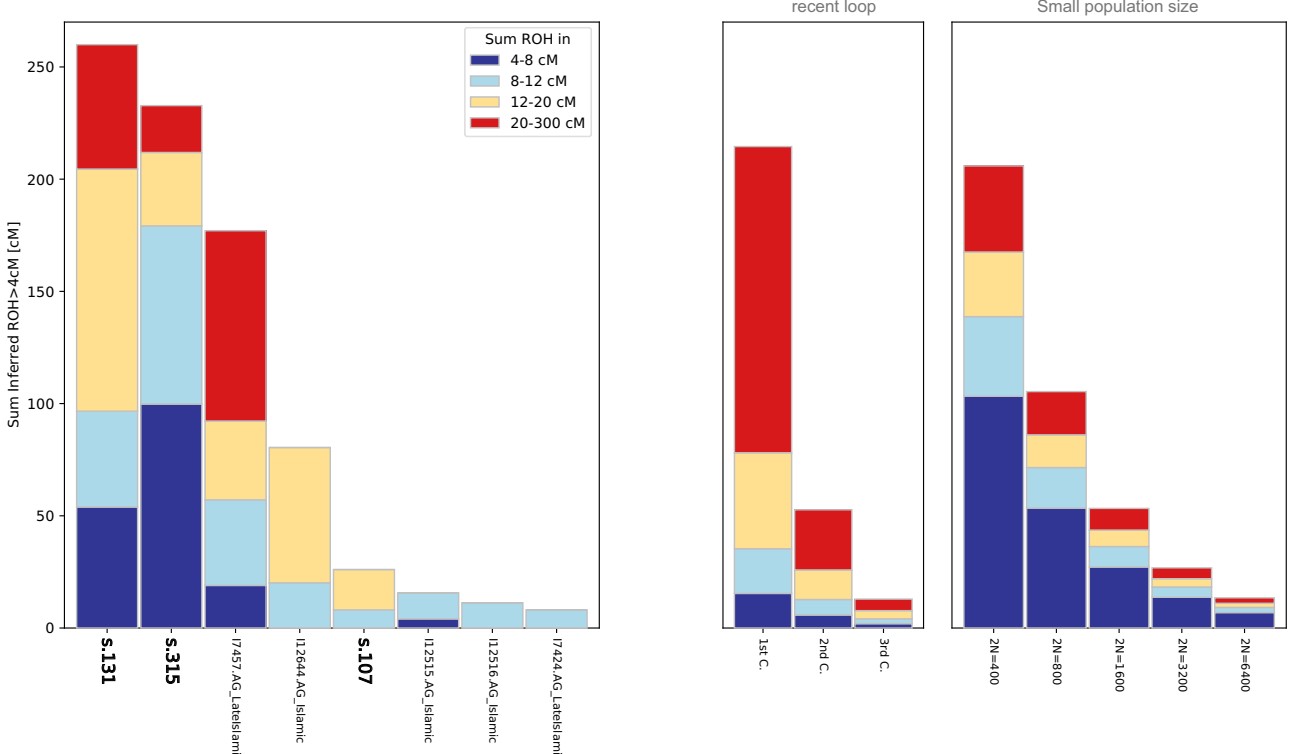

**Fig. 9 | Runs of homozygosity (ROH).** ROH analysis for individuals from Ibiza and other Islamic Iberian individuals, exhibiting cumulative ROH lengths exceeding 4 cM.

## Metagenomic analysis

To explore potential ancient pathogens and reconstruct aspects of individual health, we performed metagenomic screening based on shotgun sequencing data. Since our aim was to recover pathogens spanning a wide range of genomic characteristics (e.g., genome size, partial single-strandedness, divergence from reference due to mutation rate, pathogen load), we applied two layers of authentication: a first one using aMeta, and a second one using an evenness of coverage proxy based on the KrakenUniq results generated within aMeta, similarly to[23]. In addition, whenever a pathogen is attested at the site, we reevaluate its presence at low coverage across all individuals by analysing all the reads assigned to it via MALT[67], when the evenness of coverage proxy is bigger than 1. For the first layer of authentication, we focused on microbial taxa authenticated through a combination of metrics summarized into authentication scores by aMeta[21] (see Methods). Among taxa with authentication scores ≥ 8 (Supplementary Fig. 13), several were identified as likely environmental contaminants (e.g., *Burkholderia lata*, *Clostridium tetani*, and *Ralstonia solanacearum*). One authenticated microbe was part of the oral microbiome and is a known periodontal pathobiont associated with periodontitis[68], *Parvimonas micra,* in individual s.133 (Supplementary Figs. 14, 15). Among the authenticated taxa, only *Streptococcus pneumoniae* and *Mycobacterium leprae* met our criteria for pathogen detection, in individuals s.155 and s.313, and were confirmed by visual inspection of the full authentication plots generated by aMeta and classified as a potential pathogen and a primary pathogen, respectively (Supplementary Fig. 16, 17). Notably, *S. pneumoniae*, a pathogen known to cause pneumonia, is typically carried asymptomatically by ~10% of healthy adults[69]. Summary statistics for those three host-associated species with a score of ≥ 8 can be found in Supplementary Data 5-7.

To increase detection of viral pathogens and potential eukaryotic parasites, we performed a second layer of authentication below the aMeta authentication score thresholds using an evenness of coverage proxy (*k*-mer/reads ratio; see Methods). This approach led to the detection of hepatitis B virus (HBV) in individual s.167 (reads ≥ 200; *k*-mer/reads ratio > 1.5) (Supplementary Fig. 18). Additional potential detections were identified in s.155, s.117, s.315, s.133, s.101 and s.197 (*k*-mer/reads ratio ≥ 1), with HBV reads extracted using MaltExtract[20] (Supplementary Data 8). Those reads were submitted to BLASTn validation (e-value 1e-5; bit delta ≥ 6 interspecies for viruses), yielding in total 525, 143, 50, and 7 non-ambiguous HBV-validated reads from s.167, s.155, s.117 and s.315, respectively. (Supplementary Data 8)[20,70,71]. We implemented this second layer of authentication because the aMeta scoring system can be less sensitive for viral taxa with particular genetic characteristics (see end of section). We mapped all extracted reads with bwa aln[72], and generated deamination plots with DamageProfiler[73] (Supplementary Figs. 19, 20). We also performed de novo assembly of the extracted reads with MEGAHIT[74] and used BLASTn[70] on the longest HBV contigs, which indicated genotype D (likely D4) for s.167 and genotype A for s.155 (both with e-value 0 and 98.4% identity). Due to low coverage, damage plots could only be generated for the partial genomes of s.167 and s.155 to validate ancient status (Supplementary Figs. 19, 20). The longest contigs measured 1669 bp for s.167 and 1586 bp for s.155 out of ~3,200 bp genome length.

Similarly, human parvovirus B19 (primate erythroparvovirus 1; B19V) was detected in individuals s.131, s.133, s.157 and s.315, with 1339, 551, 457 and 477 reads respectively, extracted through MaltExtract and confirmed by BLASTn (e-value 1e−5; bit delta ≥ 6 interspecies for viruses) (Supplementary Table 9)[20,70,71]. Traces of B19V reads (<10) were found in 5 more individuals and BLASTn-validated (4, 1, 5, 2 and 1) without ambiguous assignments (bit delta ≥ 6 interspecies) (Supplementary Data 9). Authentication plots for s.131 (Supplementary Fig. 21), together with complementary damage plots for all four individuals (Supplementary Figs. 22-25), confirm the ancient origin of the partial genomes of B19V. De novo assembly using MEGAHIT[74] yielded the longest contigs of 342 bp for s.131 and 210 bp for s.315. BLAST analysis of these contigs indicated genotype 2. The longest contigs from the other two individuals (s.133: 174 bp; s.157: 304 bp) were insufficient for confident genotype attribution by BLAST.

Although both HBV and B19V yielded authentication scores below 8 in aMeta, hits with sufficiently high read counts can still be considered genuine, as a low score is consistent with their genome architecture. HBV's partially single-stranded genome and B19V's single-stranded genome with palindromic terminal repeats produce atypical breadth of coverage plots and incur a two-point penalty for uneven coverage distribution in the authentication score system. Note that, in double-stranded libraries, single-stranded regions are rarely recovered. As a result, the double-stranded portion of HBV and the hairpin-stabilised, effectively double-stranded termini of B19V are over-represented. Moreover, due to their higher evolutionary rates, viruses often display lower average nucleotide identity (e.g., 94.5% for B19V) and divergent edit distance curves compared to bacterial pathogens, which can reduce the aMeta score by up to an additional three points. Nonetheless, the deamination patterns observed for both viruses support their authenticity.

In the case of *Streptococcus pneumoniae*, the evenness of coverage proxy did not increase the detection of additional ancient pathogens. Reads assigned by MALT to this species were extracted from most individuals in this study ($k$-mer/reads ratio > 1). The BLASTn validation confirmed accurate species detection in s.155, which had an aMeta score of 9. Among the extracted reads, 1902 were validated as *S. pneumoniae*, 871 were classified as ambiguous and 95 were assigned to another species (e-value 1e-5; bit delta ≥ 5 interspecies for bacteria; Supplementary Data 6). The other individuals did not yield a majority of *S. pneumoniae*-assigned reads compared to either the ambiguous category or the reads assigned to other species.

Similarly, *Parvimonas micra* was validated in both FASTQ batches of individual s.133, which had an aMeta authentication score of 8 (s.133_A and s.133_B), with a majority of reads validated by BLASTn, whereas the reads extracted for this taxon in other individuals were predominantly assigned to other species (e-value 1e-5; bit delta ≥ 5 interspecies for bacteria; Supplementary Data 5).

For *Mycobacterium leprae*, we could only validate the detection in individual s.313 (Supplementary Data 7 and Supplementary Discussion). We extracted 11,887 reads using MaltExtract[20], from which 10,545 were BLAST-validated[73] (e-value 1e-5; bit delta ≥ 5 interspecies for bacteria; Supplementary Data 7). This *M. leprae* detection was low coverage, with at most 29,000 mapped reads and a mean depth of ~0.54× (first eager run, see Methods). This sample was therefore selected for capture enrichment.

### Phylogenetics of human parvovirus B19

We performed a phylogenetic analysis of ancient and modern human parvovirus B19 (B19V) genomes, including the four partial genomes recovered from Ibiza. Extracted reads were mapped with bwa aln and consensus sequences were generated using ANGSD, followed by realignment of all sequences with MAFFT (see Methods). We included all Ibiza individuals for whom a consensus sequence could be produced, and excluded individuals with <10 BLAST-validated reads.

The final dataset consisted of 92 genomes, including 4 ancient genomes from this study, 11 previously published ancient European genomes[75,76] and modern reference sequences representing genotypes 1–3. The alignment comprised 6992 bp, with 1203 parsimony-informative sites, 890 singleton sites and 4899 constant sites, after removal of sites with only gaps or ambiguous characters. A maximum likelihood tree was inferred in IQ-TREE2[77] under the TIM3 + F + R3 substitution model selected by ModelFinder[78], with 100 bootstrap replicates and midpoint rooting (Supplementary Fig. 26).

Two Ibiza genomes with the longest contigs, from individuals s.133 and s.157, formed a highly supported clade together within genotype 2 (bootstrap 100). The remaining two genomes, from s.131 and s.315, clustered with the medieval individual G83 from Lauchheim, Germany[78] (bootstrap 55). All four Ibiza genomes therefore fall within genotype 2, and their placement reflects the limited resolution typical of ancient genotype-2 lineages, which show short internal branches and incomplete genome recovery as noted previously[75,76]

### Phylogenetics of Mycobacterium leprae

Targeted enrichment of *Mycobacterium leprae* using the myBaits Custom Community Panel from Arbour Biosciences[79] increased the read yield from 29,000 to 128,800 (roughly 4.4-fold increase), enabling a mean genome coverage depth of 3.75× and a breadth of 76.2% after relaxed mapping (3.21× and 72.4% after FASTQ trimming and strict mapping; see Methods). This coverage enabled the placement of the *M. leprae* genome retrieved from individual s.313 into a phylogenetic context with both modern and ancient *M. leprae* genomes (Fig. 10) (see Supplementary Fig. 27 for the uncollapsed tree).

Phylogenetic reconstruction assigned the s.313 leprosy genome to genotype 2F. A Maximum Likelihood (ML) tree, based on 1,821 high-confidence SNPs covered in ≥90% of the genomes in the datasets (see Methods), placed the s.313 leprosy genome within the 2F clade (Fig. 10, Supplementary Fig. 27). This clade includes seven ancient genomes from the Middle Ages (ca. 650 to 1250 CE), spanning sites from the Hospital of Sant Llàtzer/Santa Margarida in Barcelona, Spain (ID: UF800), to Sigtuna, Sweden (ID: 3077)[80,81]. While a few internal nodes within the clade showed weak bootstrap support, the clade as a whole is defined by a long, well-supported branch (bootstrap = 100), emphasising its distinctiveness (Fig. 10).

In parallel, a Maximum Parsimony (MP) tree was generated using a relaxed SNP presence threshold of ≥80%, encompassing 2200 informative SNPs (Supplementary Fig. 28). We used parsimony because it selects the tree with the most parsimonious changes and *M. leprae* changes slowly and mostly by single-base mutations with very limited homologous recombination via horizontal gene transfer. In practice, parsimony and maximum likelihood give very similar trees for *M. leprae*, as shown in ref. 81. Despite this broader SNP inclusion, the MP topology was consistent with the ML tree and similarly placed the s.313 genome within the 2F clade. Notably, the MP tree grouped s.313 and Jorgen749 from medieval Denmark together (bootstrap = 79), suggesting they may be the most closely related within this lineage (Supplementary Fig. 28). This specific relationship was not clearly supported in the ML reconstruction, where the internal topology of clade 2F remained partially unresolved.

## Discussion

The intricate interactions between political, ethnic, cultural, religious, and linguistic groups in al-Andalus, and the extraordinary diversity that emerged from them, make it a distinctive historical case that extends beyond its own time and regional setting. This study combines genomic and metagenomic data from dated burials at an Islamic necropolis in Ibiza with documented archaeological context to characterize disease presence and the demographic landscape of al-Andalus, while providing finer resolution of genetic ancestry.

Our analyses of admixture patterns, across multiple methods and reference panels, confirm that most individuals from the Ibiza necropolis exhibit mixed Iberian and North African genetic ancestry, forming a continuum from predominantly European to predominantly North African. This underscores North Africa's central role in shaping the genetic landscape of al-Andalus, particularly after the Islamic conquest and through subsequent trade and migration networks.

Based on estimates of admixture timing, we show that North African/Middle Eastern gene flow likely occurred between 2.49 and 7.81 generations prior to the lifetimes of the individuals. Assuming a generation time of 26.9 years[46] and using the seven individuals radiocarbon-dated from 1073 to 1094 CE, the gene-flow is estimated to have occurred from 864 CE to 1025 CE. When considering the population-based estimate for these seven individuals, the main admixture event is placed around 869 CE. This is consistent with continuous admixture beginning contemporaneously with Ibiza's

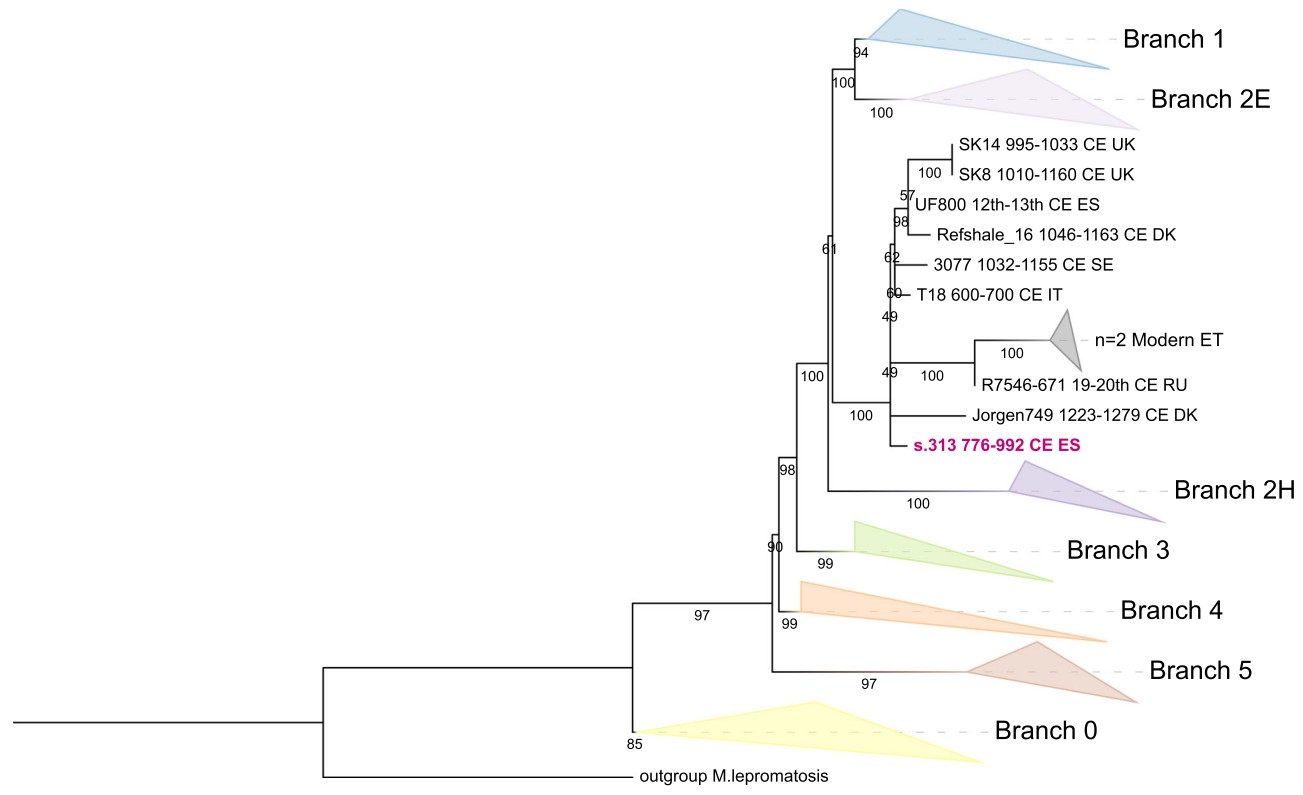

**Fig. 10 | Maximum-likelihood (ML) phylogeny of *Mycobacterium leprae*.** Main branches are collapsed by major lineage, except for branch 2F, which is shown expanded. The s.313 genome is highlighted in red, positioned within clade 2F. Tip labels indicate the sample ID, time period, and country code for ancient genomes. Bootstrap values indicate branch support. The tree was inferred from high-quality informative SNPs (≥3× coverage, genotype quality ≥30, base frequency ≥90%) retained at positions covered in ≥90% of the samples. *Mycobacterium lepromatosis* was used as the outgroup. The phylogeny was reconstructed using the GTR + F model and 100 bootstrap replicates.

incorporation into the Islamic world in 902 CE. These findings suggest that admixture likely occurred either in Ibiza or in mainland al-Andalus during and after the island's conquest. This implies that al-Andalus was not only a crossroads for diverse ethnic groups from across the Islamic world, but also a region where substantial intermixing between those groups actively took place in a short period of time. However, as our study focuses on an urban cemetery, we cannot extrapolate these patterns to contemporary rural (tribal) communities, whose dynamics remain largely unknown.

Individuals s.157 and s.313 have genomes containing only trace amounts of North African ancestry (13% and 11%, respectively) (Fig. 4). For s.157, we were able to estimate an admixture date, likely reflecting a pre-Islamic event (Fig. 7b), while insufficient data prevented dating the admixture in s.313. The presence of both individuals in a Muslim cemetery aligns with the profile of the muladíes (muwalladûn), recently Islamized local Iberians, as described by ref. 82. This suggests that cultural and religious transformations could occur independently of, and potentially more rapidly than, genetic admixture.

The radiocarbon dates and genomic results presented here support and align with contemporary chronicles, indicating two main demographic pulses shaping the genetic landscape of the Balearic Islands. The first followed the Umayyad expedition around 902 CE, when a fleet carrying Arabs, Imazighen, and Islamised Iberians (muladíes) settled the newly conquered islands[83]. The second occurred after the Almoravid conquest of Mallorca in 1115–1116 CE, involving the arrival of Sanhaja garrisons, Andalusi artisans, and Sub-Saharan auxiliaries[84], and was later reinforced by refugees fleeing the collapse of Almoravid rule on the mainland[83,85]. These episodic arrivals, alongside continuous movement along western Mediterranean trade and slave routes, generated the layered genetic signatures observed in the cemetery. This is also consistent with isotope and dietary analyses from medieval Ibiza, indicating the presence of non-local individuals and African dietary influence characterized by the consumption of C4 plants such as millet or sorghum[11,86,87].

Sub-Saharan African ancestry is represented by individuals s.117 and s.197, whose genomes best match present-day populations from southern Chad (e.g., Sara and Laal) and Senegambia (e.g., Bedik and modern Gambians), respectively. The coexistence of these two very different ancestries shows that 11th- to 12th-century Islamic Iberia drew people from both the central and western Sahel regions. The radiocarbon dates of these individuals, corrected for marine reservoir effects, place them after 1115 CE in the second demographic pulse that reached the Balearic Islands following the Almoravid conquest of Mallorca. Their genetic ancestry profiles are consistent with Arabic records describing the northward movement of captives from modern-day Chad, via the Kawar and Fezzan oases (in present-day Niger and Libya, respectively). They also align with accounts of enslaved Senegambian individuals and military auxiliaries referred to as Tukrūrī, a name derived from Takrūr, a 9th- to 13th-century Muslim kingdom located in the middle-lower Senegal Valley (in what is now northern Senegal and southern Mauritania)[85]. The historical accounts report that, three decades before the Mallorca conquest, Yūsuf b. Tāshfīn crossed into Iberia with approximately 4,000 Takrūr cavalry, free or newly manumitted horsemen from the Senegal and Gambia valleys, supported by Sudanese infantry at the Battle of Zallaqa/Sagrajas in 1086 CE[85]. Following Takrūr's royal conversion to Islam around 1030 CE, trans-Saharan networks were further developed to supply both captives and voluntary recruits to Almoravid armies and garrisons across the western Mediterranean. This chronological placement, together with the Chadian- and Senegambian-derived

ancestries observed in these individuals, aligns closely with Almoravid military recruitment, slave trafficking, and resettlement practices documented by sources such as Ibn ʿIdhārī, al-Bakrī, the al-Bayān al-Mughrib tradition, and modern syntheses[83,88,89].

The elevated genetic inbreeding observed in individuals s.131 and s.315 (Fig. 9) may reflect endogamous practices within specific groups, potentially structured along cultural or ethnic lines. Similar patterns have been documented among Imazighen groups and many other communities[50,61], where cousin marriages, especially between first cousins, were historically used to maintain familial ties and reinforce cultural and social cohesion[90]. High levels of genetic consanguinity have also been documented in an early Islamic individual from Valencia[19] and in later Islamic individuals (e.g., I7457) from the Almohad period in Granada[18]. Alternatively, this pattern may reflect restricted marital networks and genetic isolation, as suggested for other small medieval Iberian communities[19,23].

Alongside genetic ancestry, our metagenomic analyses identify authenticated pathogen signals for hepatitis B virus, human parvovirus B19, *Mycobacterium leprae* and *Streptococcus pneumoniae*. The latter can be carried asymptomatically. The probability of pathogen detection is limited by the small sample size (n = 13) and identification of oral microbiome taxa is further constrained by within-tooth sampling, which favours blood-borne pathogens. We authenticate one oral microbe, *Parvimonas micra*, a periodontal pathobiont, associated with periodontitis[68]. Most notably, although osteological evidence of leprosy has been documented in medieval Islamic communities[91], the genome we report provides an opportunity to characterise *Mycobacterium leprae* in this historically understudied context.

The *M. leprae* genome recovered from individual s.313 belongs to genotype 2F and falls within a clade of ancient genomes dated between the 7th and 13th centuries, from sites ranging from Barcelona to Sigtuna[80,81]. The broad distribution of that lineage highlights the long-distance connectivity of disease transmission across medieval Europe and suggests that Ibiza was part of these broader epidemiological networks. Interestingly, the burial of s.313 was indistinguishable from others at the site, with no signs of marginalisation in grave treatment[9]. This pattern may point to a reluctance within Muslim communities at the time to exclude individuals afflicted with leprosy, and underscores the importance placed on fulfilling duties toward the sick[92]. This reflects a broader pattern of inclusion of individuals affected by leprosy in medieval societies, with similar evidence also found in Christian cemeteries (see ref. 81 and references therein). Alternatively, the individual may have died before developing visible symptoms of the disease or may have had a subclinical infection that never progressed to full-blown disease. Although the available skeletal remains, notably lacking certain elements such as the facial bones in individual s.313, are not sufficient to completely exclude the presence of leprosy osteologically, the studied bones do not show any apparent signs of the disease[93].

In addition to *M. leprae*, we identified hepatitis B virus (HBV) and human parvovirus B19 (B19V), respectively responsible for hepatitis B and the fifth disease (colloquially called slapped-cheek syndrome). They were detected using a proxy for evenness of coverage (see Methods). HBV's partially single-stranded genome and B19V's single-stranded genome with palindromic terminal repeats produce atypical coverage profiles, meaning that stringent evenness thresholds can filter them out. Their fast mutation rates can also produce false negatives in identity-based screening. Nonetheless, the deamination patterns observed for both viruses support their authenticity. For HBV, we found genotypes A and D of hepatitis B virus (HBV) in individuals s.155 and s.167. These genotypes fall within the known diversity of HBV strains circulating in medieval Europe[26]. Low-coverage HBV signals were also detected in individuals s.117 and s.315 and validated by BLASTn (50 and 7 reads) with low read ambiguity, which, given the specificity of the viral sequence, suggests genuine pathogen detection.

However, further sequencing would be required to plot reliable deamination patterns.

Human parvovirus B19 (B19V) yielded partial genomes in four individuals: s.131, s.133, s.157 and s.315. Phylogenetic analysis indicated that they all clustered within genotype 2. Dating from the 10th to 12th centuries CE, these four partial to low-coverage genomes help to bridge a temporal gap in the European B19V record between ~950 CE and the 20th century. Their assignment to genotype 2 extends the known persistence of this lineage by over a century[64] and supports the hypothesis that genotype 2 remained widespread and stable throughout the medieval period, maybe even until its replacement by genotype 1 around the 1970s[94]. Traces of B19V were also recovered and BLAST-validated in five more individuals (≤ 5 reads per ancient pathogens) and could be true positives, given the ubiquity of the virus today, but further sequencing would be required for a definitive answer.

Overall, we identified molecular evidence of infection by primary pathogens in at least 3 of the 13 individuals analysed, involving *M. leprae* and HBV. Including the two individuals with low-coverage HBV signals raises this number to 5 and a rate of 38%, which is comparable to proportions reported in broad-spectrum pathogen screening studies at other medieval sites[23,75]. When we also consider likely infections with *S. pneumoniae* and *P. micra*, the number of individuals with at least one detectable pathogen increases to 7, or about 54%.

Because B19V in adults most likely reflects persistence after childhood infection, we treat it separately. In our material, four individuals yielded partial B19V genomes and a further five individuals carried very low coverage traces. This translates into a minimum of 31% (4 of 13) and a maximum of 69% (9 of 13) of individuals with detectable B19V DNA. Given the high prevalence of lifelong B19V persistence in modern populations[95], the true proportion in our sample probably lies toward the higher end of this range. Our metagenomic analyses provide direct evidence for the presence of these pathogens in medieval Ibiza, establishing a foundation for future comparative work to explore their wider historical and epidemiological patterns.

In conclusion, this study offers a high-resolution snapshot of medieval Ibiza's gene pool, revealing pervasive admixture between local Iberian and North African genetic ancestries. This admixture probably began during the island's colonization and continued up to just two or three generations before the lifetimes of some individuals analysed here. In addition, we identify two individuals whose genomes trace to distinct Sub-Saharan regions, southern Chad and Senegambia, providing genetic evidence of the long-distance slave and military Almoravid networks described in Arabic sources.

Together, our findings support a two-pulse demographic model that shaped the Islamic medieval population history of the archipelago. Notably, s.313, who shows minimal North African ancestry and was likely a muladí, carries a *Mycobacterium leprae* genome of genotype 2F. The pathogens detected in this study provide insights into health conditions at the necropolis and establish a baseline for future comparative work. Furthermore, the leprosy genome from s.313 provides a starting point for future studies of leprosy in the medieval Islamic world, which will benefit from additional genome-wide data as more ancient cases are analysed.

Finally, given the scarcity of written and archaeological sources for the first Muslim occupation of the Balearics in the early tenth century CE, these dated genomes provide an unprecedented line of evidence for this dynamic and transformative period and offer a baseline for future population genetics research in the Western Mediterranean.

## Methods
### Ethics Statement
All archaeological human remains analysed in this study originate from a medieval Islamic necropolis discovered fortuitously during

construction works in Ibiza, Balearic Islands, Spain. In accordance with Spanish heritage legislation (Law 16/1985 on Spanish Historical Heritage) and the regional heritage law of the Balearic Islands (Law 12/1998), all archaeological materials are considered part of the public domain and fall under the protection and authority of the regional cultural heritage institutions.

The excavation, recovery, and study of the remains were carried out under the scientific direction of Glenda Graziani and Juanjo Marí, who were officially designated as the director of the archaeological intervention and as the temporary legal custodian of the materials by the Consell Insular de Ibiza. Within this framework, Prof. Anders Götherström and Dr. Ricardo Rodríguez-Varela were authorised, under the official excavation and study permit, to carry out the genetic analyses and radiocarbon dating, including the selection of samples for these specific purposes.

Sampling was conducted during the authorised study period and under the same legal framework. Only small analytical subsamples (e.g., tooth or bone fragments) were exported to Sweden for genomic analysis. These subsamples do not constitute archaeological objects under Spanish heritage law (Art. 40, Law 16/1985, Government of Spain; and Arts. 2 and 46, Law 12/1998, Government of the Balearic Islands). Consequently, their transfer complied fully with Spanish and Swedish regulations.

All samples analysed in this study derive from the original excavation and had not been previously subjected to ancient DNA analysis or published elsewhere. No additional destructive sampling was performed beyond that authorised.

This research was also approved by the Cranfield University Research Ethics System (Ref: CURES/26677/2025).

All human remains were treated with dignity and respect throughout the study, in accordance with the principles outlined in the internationally renowned British Association of Biological Anthropology and Osteoarchaeology (BABAO) Code of Ethics (https://babao.org.uk/wp-content/uploads/2024/01/BABAO-Code-of-Ethics.pdf.

## Osteological Analyses
The analysis of the human remains focused on estimating the minimum number of individuals and reconstructing their biological profiles, including age-at-death, biological sex, stature, and the presence of any pathological alterations or trauma that could inform on past living conditions and funerary practices.

Sex was estimated primarily from pelvic and cranial morphology[94,96,97], complemented by metric assessments[98]. Age-at-death[99] estimation was based on skeletal development[97] and dental formation[100,101] as well as the morphology of the pubic symphysis, auricular surface, and rib ends when preserved[102–104]. These methods were complemented by observations of dental wear, ante-mortem tooth loss, and age-related pathological conditions.

Stature was estimated, when possible, from the maximum length of long bones (preferably the femur), applying the regression equations developed by refs. [105] and [106].

## Radiocarbon dating
Radiocarbon pre-treatment followed the standard Uppsala Tandem Laboratory (Uppsala, Sweden) protocol for bone collagen (report p5077). Surfaces were mechanically cleaned (scraping or sand-blasting), ultrasonically washed in boiled distilled water (pH 3), ground, and demineralized in 0.8 M HCl at -10 °C for 30 min (fraction A). The insoluble residue was gelatinized in distilled water at pH 3 for 10 h at 90 °C, producing an insoluble fraction C and a soluble fraction D. The soluble "fraction D", containing most of the original collagen, was combusted to $CO_2$, graphitized using an Fe catalyst, and dated by AMS at Uppsala Tandem Laboratory.

## DNA extraction and library preparation
Human remains were sampled at the aDNA facilities within the Center for Palaeogenetics, Stockholm University (Sweden). Prior to analysis, the bone surface was removed, and the remaining material was drilled into powder. For tooth samples, the root tips were cut using a multi-tool drill (Dremel) to obtain approximately 80 to 150 mg of bone powder or root tip material. These root tip samples were then placed in Eppendorf tubes containing 1 ml of predigestion buffer [0.45 M EDTA (pH 8.0)] and incubated at 37 °C in a hybridization oven with rotation. After 30 minutes, the supernatant was discarded to minimize exogenous DNA contamination.

Following this predigestion, 1 ml of extraction buffer [0.45 M EDTA (pH 8.0) and proteinase K (0.25 mg/ml)] was added to the samples. They were incubated in the hybridization oven at 37 °C with rotation for 1 to 4 days until the powder or root tip dissolved. DNA extraction followed established protocols[107], using the same reagents and consumables, specifically: 1 ml of the digested extract was combined with 13 ml of binding buffer containing 5 M guanidine hydrochloride, 40% (v/v) isopropanol, 0.05% Tween 20, and 90 mM sodium acetate (pH= 5.2). Fifty-millilitre silica columns (Roche, High Pure Viral Nucleic Acid Large Volume Kit) were used for DNA purification, and the DNA was eluted in 45 µl of Elution Buffer (EB; Qiagen). Blank controls were included at each step of the extraction process.

## Library preparation and sequencing
For library preparation, 20 µl of extract was used to create blunt-end ligation DNA libraries with P5 and P7 adapters and double indexes as described in ref. 108 with the following specific settings: Blank controls were used throughout library preparation and amplification. The optimal number of polymerase chain reaction (PCR) cycles was determined by quantitative PCR. PCR reactions had a final volume of 50 µl, including 5 µl of DNA library and the following final concentrations: 1× AmpliTaq Gold Buffer, 2.5 mM $MgCl_2$, 25 µM of each deoxynucleotide triphosphate, 2.5 U of AmpliTaq Gold (Thermo Fisher Scientific, Waltham, MA), and 200 nM of each index primer. PCR conditions included an activation step at 94 °C for 10 minutes, followed by 8 to 20 cycles of 94 °C for 30 seconds, 60 °C for 30 seconds, and 72 °C for 45 seconds, with a final elongation step at 72 °C for 10 minutes. Four amplification reactions were carried out per library to increase complexity. After amplification, the libraries were pooled and purified with AMPure XP beads (Agencourt, Beckman Coulter, Brea, CA). Fragment size and concentration were verified using BioAnalyzer with the High Sensitivity Kit (Agilent Technologies, Cary, NC).

## Capture enrichment for *M. leprae* in s.313
The shotgun library from individual s.313, which was positive for *Mycobacterium leprae* DNA, was concentrated and enriched using the *M. leprae*-specific myBaits Custom Community Panel from Arbour Biosciences (Design ID: D10227MLPRA)[79]. Two rounds of hybridisation capture were carried out on individual libraries, following the High Sensitivity guidelines in protocol version 5.02 (Arbour Biosciences, 2022), using a hybridisation temperature of 60 °C. To maximise target-bait interactions in low-endogenous ancient samples, hybridisation time was extended to 44 hours, as recommended for challenging aDNA material (Arbour Biosciences, 2018[109]). Post-capture amplification was performed using KAPA HiFi HotStart ReadyMix (Roche), followed by purification with the MinElute PCR Purification Kit (Qiagen). When initial library yields were low, an additional round of PCR was conducted based on qPCR quantification, to ensure sufficient concentrations of target-enriched libraries before sequencing. Final sequencing was performed on the NovaSeq 6000 platform at the Science for Life Laboratory (SciLifeLab) in Stockholm (Sweden).

## Sequencing and human data processing

Purified libraries were pooled in equimolar concentrations and sequenced on an Illumina NovaSeq 6000 at the SciLifeLab National Genomics Infrastructure (SciLifeLab-NGI) in Stockholm using an S4 flowcell with XP clustering; either the S4-300 (v1.5) kit or the S4-200 XP kit was used, both supporting standard paired-end 2×150 bp reads. Sequencing reads were demultiplexed during BCL to FASTQ conversion using bcl2fastq v2.20.0.422 (CASAVA), with a Sanger / phred33 / Illumina 1.8+ quality scale at SciLifeLab-NGI. Adapters were trimmed using Cutadapt v. 2.3[110] (–quality-base 33 –nextseq-trim = 15 –overlap 3 -e 0.2 –trim-n --minimum-length 15:15), and fastq reads were merged using FLASH v. 1.2.11 (–min-overlap 11 --max-overlap 150 –allow-outies). The reads were aligned against the human reference genome build 37 (hs37d5) and build 38 (GRCh38) using Burrows-Wheeler Alignment (BWA) v. 0.7.17 with parameters (-l 16500 -n 0.01 -o 2)[72]. Fastq files from different sequencing runs of the same library were merged using Samtools v. 1.17 (using samtools merge command)[111], followed by a consensus read creation using the script FilterUniqueSAMCons.py (https://bioinf.eva.mpg.de/fastqProcessing/) with the following modifications (addition of a --count_file option to record counts of sequences, tie-breaking in calc_consensus() using random() when multiple bases have equal scores). Reads shorter than 30 bp or with less than 90% identity to the reference were excluded using percidentity_threshold.py[112]. Due to the low coverage of ancient samples, pseudo-haploid genomes were generated by randomly selecting one read with a minimum mapping quality of 30 and base quality of 30.

## Contamination and data validation

All libraries produced short read lengths, and patterns of cytosine deamination were assessed using PMDtools v0.60 (https://github.com/pontussk/PMDtools) with default parameters. Contamination estimates were obtained using two mitochondrial DNA (mtDNA) methods; one following Green et al. (2008)[113] using default settings and contamMix[114] with the following settings (MCMC was run with 100,000 iterations (nitre), 3 chains (nChains), an alpha of 0.1, a base quality score of 30 (baseq), a trimming of 10 bases (trimBases), and the 'transverOnly' option set to FALSE and the "Contamination" program in ANGSD v.0.911 to estimate X chromosome contamination in males[115] with -minMapQ 30 -minQ 30. Contamination estimates were listed (Supplementary Data 1).

## Imputation of ancient genomes

We used TrimBam (default settings) from BamUtils v1.0.15[116] to trim 10 bp from the end of each read in our ancient individuals, thereby eliminating the damage patterns characteristic of ancient and degraded DNA. We imputed the bam files mapped to the human reference genome GRCh38 with GLIMPSE2 v2.0.1[117] using the 1K-HGDP dataset[32] as reference, following the scripts available at https://github.com/PalaeogeneticsandPopulationGenetics/Workshop-for-Genotype-Imputation

## Assembled datasets

The generated BAM files were mapped to the human reference genome build 37 (hs37d5), and then merged separately with reference populations included in datasets: "1240 K + HO" 54.1.p1[66], Africa1 and Africa2. The imputed bam files were mapped to the human reference genome GRCh38, and then merged with populations included in datasets: "1K-HGDP"[32], Africa1, and Africa2.

## Filtering and harmonization of modern and ancient datasets

We used PLINK (v1.90[118];) and bcftools (v1.15) to prepare two datasets for population genetic and local ancestry analyses: (1) ancient, imputed genomes from Ibiza plus two previous published ancient individuals as references; one pre-European contact Canary Island individual gun011 and one early medieval (pre-Islamic) individual from North Spain

(Ido039)[23,39] (2) a reference panel of modern individuals from the 1000 Genomes Project and HGDP datasets[32].

We first filtered the modern reference dataset (1K-HGDP) to retain only biallelic SNPs and excluded variants with a minor allele frequency (MAF) less than 0.05.

The Ibiza dataset was filtered to retain only high-quality genotypes with imputation probabilities ≥ 0.99. Individual s.155 was excluded from local ancestry analysis due to excessive missing data, as RFMix v1.5.4 does not tolerate missing genotypes. Individuals s.313 and s.197 with relatively higher levels of missingness were retained but analysed separately to avoid bias.

To harmonize the two datasets prior to merging, we standardized SNP identifiers to the CHR:POS format, identified the intersecting set of SNPs, and removed duplicated or inconsistent entries (see scripts at ref. 119). We retained only SNPs shared between the two datasets. The harmonized datasets were then merged using PLINK, resolving strand mismatches and excluding problematic variants that could not be reconciled. The resulting merged dataset was further filtered with MAF > 0.05. This harmonized dataset served as the foundation for both global and local ancestry analyses.

To curate the African datasets, we followed the quality-control recommended by ref. 41 for the Africa1 dataset, and by ref. 50 for the Africa2 dataset. Before merging, first- and second-degree relatives were removed, and only SNPs and individuals with high genotyping rates were included (plink --geno 0.1 --mind 0.1, respectively).

## Local ancestry inference

For local ancestry inference, we used RFMix v1.5.4[43], which requires fully phased, high-quality, and complete genotype data. To prepare the input files, we subset the ancient and modern datasets to include only overlapping SNPs, modified variant identifiers to the CHR:POS format, and ensured identical variant order across both files (see scripts at[119]). The final filtered and synchronized VCFs were compressed and indexed to serve as RFMix input.

We selected the following reference populations from the 1K-HGDP dataset to represent plausible ancestral sources for the Ibiza individuals. The Iberian component included 44 CEU and 21 Basque individuals, while North African/Middle Eastern ancestry was represented by 22 Mozabite and 43 Bedouin individuals. Because Sub-Saharan Africa harbours the highest levels of human genetic diversity, we included larger reference panels from this region: 97 Luhya (East Africa), and 97 Yoruba, 97 Gambian, and 98 Sierra Leonean individuals (West Africa). These populations were chosen based on PCA and ADMIXTURE results, which identified them as the most informative sources of African-related ancestry in the ancient individuals. The larger Sub-Saharan African panels ensure adequate representation of regional diversity, whereas the smaller but balanced European and North African panels sufficiently capture variation relevant to our dataset. RFMix v1.5.4 was run with default settings, except for setting the window size to 0.1 cM (-w 0.1) and enabling the forward-backward algorithm (--forward-backward) for probabilistic ancestry inference. The results were plotted using available Python scripts (https://github.com/alisi1989/RFMIX2-Pipeline-to-plot).

## Dating admixture events

We estimated admixture dates using the decay of local ancestry covariance with increasing genomic distance between SNPs on the same chromosome, which reflects the breakdown of continuous ancestry blocks due to recombination events at each generation, since the admixture process started. This analysis was done using the script cov_decay_Eivissa_published.R available at[119].

Local ancestry was first inferred with RFMix v1.5.4, using a merged dataset comprising our ancient Ibiza individuals and relevant modern reference populations. We used phased genotype data and assigned ancestry at each SNP along the genome for each individual.

Then, we computed the covariance between local ancestry assignments of pairs of SNPs located on the same haplotype as a function of the genetic distance between them. Specifically, as a function of r (the probability that an odd number of recombination events has occurred between two loci), $r = (1-e^{-2d})/2$, where d is the genetic distance in Morgans between them.

Finally, we fitted the covariance decay to the equation $cov = A(1-r)^g + C$, where g is the number of generations since the admixture event took place, assuming random mating. We discretized r in 0.001 width windows, and computed the mean local ancestry covariance per window. Then, we performed a nonlinear least squares fitting using the nls2 package with the 'port' algorithm in R. To estimate the uncertainty around the parameter g, we resampled 100 times the 0.5% of SNPs per chromosome (17,616 total), and computed the covariance for all within-chromosome SNP pairs. We calculated the 2.5th and 97.5th percentiles of the resulting distribution of g estimates to obtain an empirical 95% confidence interval.

We generated both individual-level and population-level covariance decay curves. For the population-based analysis, we combined data from seven individuals with radiocarbon dates ranging from 1073 to 1094 CE to increase statistical power and to obtain an averaged admixture date representative of the population.

## Mitochondrial and Y chromosome haplogroups

Mitochondrial haplogroups were determined by filtering mtDNA reads with a minimum mapping and base quality of 30 using Samtools v. 1.17[111]. Consensus calling was performed with bcftools v1.15[111]. Haplogroups were assigned using HaploGrep 3[120] based on PhyloTree Build 17[121] (Supplementary Data 1). For Y chromosome haplogroups, we used pathPhynder v.1a[122] with the "BigTree" Y chromosome dataset as the reference phylogeny.

## Kinship analysis

We tested for genetic kinship among the studied individuals using four different software: *NgsRelate* v2.0[123], *KIN* v3.1.3[124], *READ* v2[125] and *ancIBD* v0.7[126]. To assess consistency and increase confidence in our kinship estimates, we jointly evaluated the results obtained from all four methods. For the *NgsRelate* and *KIN* analysis, we used a panel of 1,554,712 autosomal transversion SNPs from the Estonian Genome Diversity Project (EGDP)[127], following the methods and parameters described in ref. 23. For *READ* analysis, first, the reads mapped to the human reference genome (version hs37d5) were trimmed from both ends by 10 bp using the *trimBAM* command of *bamUtil* software[116] to remove postmortem deamination artifacts. We then genotyped the data using pseudo-haploid calling at 1,150,639 autosomal SNPs from the 1240 K dataset (v62.0)[66] with an in-house pipeline, retaining one random allele per site, as outlined in ref. 23. Finally, we ran *READ* on the studied individuals with the default parameters. We did not find any genetically related pairs among the individuals based on those three tools. Summary statistics of all kinship results are provided in Supplementary Data 3.

For the *ancIBD* analysis, we first imputed the aforementioned trimmed genomes using *GLIMPSE2 v2.0.1*[117] and the 1000 Genomes Project phased data (1000 Genomes Project Consortium) as the reference panel, with the default parameters. We then ran *ancIBD* on the imputed data using default parameters[126] to infer pairwise shared identical by descent (IBD) segments. No shared IBD segments longer than 8 cM were identified between any pair of individuals.

## Principal component analysis

PCA was performed based on the different datasets using the *smartpca*[38] module in EIGENSOFT v5.0.1 with the options lsqproject and shrinkmode set to YES, thus enabling projection of ancient individuals onto the PC space calculated using modern reference data.

## Model-based clustering analysis

ADMIXTURE v1.3 was used to perform unsupervised ADMIXTURE analysis on a selection of worldwide modern individuals from the 1240 K + HO dataset, along with the ancient Ibiza individuals. To minimise sample-size imbalances and reduce the influence of missing data in ADMIXTURE analyses, we selected up to 10 high-quality individuals per population with the fewest missing SNPs. Preliminary testing indicated that this number was sufficient to capture the major ancestry components while avoiding overrepresentation of populations with larger sample sizes. In addition, for several populations in the dataset, 8 to 10 individuals represented the maximum number of suitable samples available. The results of this approach show that at K = 7, a low cross-validation (CV) error and a clear differentiation between North Africa, Middle East/Caucasus, Europe, Asia, Mbuti, Hadza and West Africa populations. Next, using populations that maximize the seven different components: East Asian (Han and Japanese), West hunter-gatherers (Mbuti), West African (Yoruba, Mandeka), East hunter-gatherers (Hadza), North African (Mozabite, Saharawi, and Algerian), the Middle Eastern and Caucasus populations (Assyrian, Georgian, and Kurds), and European (Basque, Lithuanians, and Norwegians), we performed a supervised ADMIXTURE analysis on the rest of the modern populations, the ancient Ibiza samples and other ancient samples as references[18,23,29,39,42,128–131], building on the results from the unsupervised admixture analysis. We also run unsupervised ADMIXTURE analyses with the 1K-HGDP dataset and the imputed ancient samples, resulting in similar results (Supplementary Fig. 3). Finally, we run unsupervised ADMIXTURE analyses on both African datasets using the pseudohaploid ancient samples. All the datasets were pruned for linkage disequilibrium (LD) using Plink v1.90 (--indep-pairwise 200 25 0.4)[118], and the results were parsed, aligned, and plotted with PONG[132].

## *f*-Statistics Patterson's analysis

Outgroup f3-statistics of the form $f_3$(Ju'hoansi; X, Y), were computed with ADMIXTOOLS v7.0.1[133]. The Ju'hoansi, a southern African Khoi-San-speaking population, represent one of the earliest diverging lineages of modern humans and have low levels of recent admixture. This highly divergent and minimally admixed population provides a reliable outgroup for measuring shared genetic drift between target populations and is widely used in previous African population genetic studies[49,134]. Pairwise genetic distances ($1-f_3$) were then used to generate classical multidimensional scaling (MDS) plots in R (cmdscale function[135];), visualized with the R package ggplot2[136] by assigning group-specific colours and shapes and reporting the variance explained ($R^2$) as a measure of fit. To minimize biases from data type differences, both ancient and modern individuals were pseudohaploidized, and to control for sample size effects, we analysed a consistent subset of 10 randomly selected individuals per population, as in the ADMIXTURE analysis.

## Metagenomic analysis

Metagenomic screening was performed using the pipeline aMeta v1.0.0 (commit 9bbce53)[21], applying custom filters of 100 species-specific reads (*taxReads*) and 1000 unique *k*-mers with default microbial databases mirroring NCBI nt microbes and selected complete eukaryotic genomes, including human[21,137,138]. For each individual, microbial species received an aMeta authentication score, based on edit distance, evenness of coverage, deamination profile, read length, PMD scores, average nucleotide identity (ANI), and read count, which were retained for downstream analysis. These scores are plotted in aMeta in a heatmap.

We exported the aMeta scores table and re-plotted the heatmap in R with the aMeta plotting script after subsetting to score ≥ 8 and then split the display into likely environmental versus likely host-associated taxa as described below. Likely environmental contaminants and oral

commensals were identified via a four-step exclusion strategy exactly like the one used in reference [23]: (i) BacDive[139] 'Isolation sources' taxon lists were downloaded in May 2025 and searched. Taxa present in lists for aquatic, terrestrial, plant-associated, fungal, clean room or engineered product sources were flagged as environmental. Matches were performed at species level with genus-level fall-back where species entries were unavailable (ii) Microbe Atlas Project[140] prevalence was consulted and taxa with > 1% relative abundance in uncontaminated soils excluding farm and unknown were classified as environmental taxa, (iii) ambiguous cases were checked against the primary literature and (iv) taxa were queried in the expanded Human Oral Microbiome Database (eHOMD)[141] to identify oral commensals.

To explore potential pathogens that fell below our thresholds due to specific genetic characteristics, such as the single-strandedness of some viruses or their fast mutation rate, we applied a secondary layer of detection to further evaluated hits by calculating the unique $k$-mer to read ratio (based on the KrakenUniq v.1.0.4 results generated within aMeta[142,143]) as a proxy for evenness of coverage, using awk. Known human primary pathogens with a ratio of at least 1.5 and at least 200 reads (as opposed to *taxReads* used in aMeta) were evaluated based on available authentication plots. This led to the identification of a hepatitis B virus (HBV) infection in individual s.167 (ratio: 9.3) and primate erythroparvovirus 1, alias human parvovirus B19 (B19V), in individual s.131 (ratio: 1.3).

We then used MaltExtract v.1.7[61] with the same parameters as in aMeta, to extract HBV, B19V, *M. leprae*, *S. pneumoniae* and *P. micra* reads from MALT v.0.6.2[67] alignment files across all individuals with a k-mer to read ratio ≥1 regardless of read count, since those pathogens were now attested at the site. We verified each assignment of these potential detections using BLAST v.2.15.0+ on our cluster against the downloaded NCBI nt database (BLAST DB Version 5) with parameters (-perc_identity 90, -qcov_hsp_perc 80, -evalue 1e-5, -word_size 11, -soft_masking false, -max_target_seqs 50, -max_hsps 1) (Supplementary Data 5–9)[70,71]. Reads were "BLAST-validated" if they had a bit score difference (delta) between the target species and another species of 6 for viruses and 5 for bacteria (given their divergent evolution rate). Vectors and synthetic construct sequences were ignored in this inference. De novo assemblies of the reads were performed using MEGAHIT v.1.2.9[74] with default parameters, and the longest contigs were queried against the NCBI nucleotide database using the BLAST web interface (May 2025), for genotype assignment[70,71]. Additional damage, read length and edit distance plots were generated using DamageProfiler v.1.1 on the extracted reads mapped with bwa aln v.0.7.17[72,73].

### *Mycobacterium leprae* mapping, damage assessment and damage removal

Mapping of *Mycobacterium leprae* reads from individual s.313, together with a reference dataset of ancient and modern genomes (see Supplementary Discussion), was performed using the nf-core/eager v.2.5.0 pipeline[144], along with associated tools for authentication and pre-processing[67,72,73,111,115,116,144–167]. In brief, adapters were trimmed with AdapterRemoval v.2.3.2[150], and reads were aligned to the TN reference genome (GenBank RefSeq: NC_002677.1) with an edit distance parameter -n 0.01 using CircularMapper (http://github.com/apeltzer/CircularMapper), which implements the bwa aln aligner v.0.7.17[75]. BAM filtering was activated and only reads with a mapping quality ≥ 20 and read length ≥ 30 were retained. Duplicates were removed using MarkDuplicates (https://github.com/broadinstitute/picard).

Afterwards, damage removal was applied with fastp v.0.24.0[151] to the adapter-trimmed FASTQ files (for both the shotgun and capture libraries) based on the UDG-treatment type, as verified by visual inspection of damage patterns. Fully UDG-treated libraries showing no visible remaining damage were retained without modification. Partially UDG-treated libraries were trimmed by 1 bp at each end, with a few samples trimmed by 2 bp. Non-UDG-treated libraries were trimmed by 4 bp on each end.

### *Mycobacterium leprae* variant calling and SNP alignment

Trimmed FASTQ files were re-processed with nf-core/eager v.2.5.0 again[144] using the same parameters, except for a stricter bwa aln mismatch rate (-n 0.2), similarly to[81]. This adjustment was possible due to prior damage removal, which allows stricter mapping and helps reduce mismapping from other species.

BAM files were further processed within nf-core/eager v.2.5.0 using GATK3 to call variants with UnifiedGenotyper v.3.5, using the EMIT_ALL_SITES option[157,168] to generate output compatible with MultiVCFAnalyzer v. 0.85.2[145] input requirements.

The outgroup *Mycobacterium lepromatosis* FASTQ SRR1576832 (associated GenBank: JRPY00000000.1) was selected for its high coverage and processed in a separate nf-core/eager run using relaxed parameters: an edit distance of 0.01 (similar to the screening run) and a seed length of 16, in order to retain sufficient SNP coverage, following reference[169]. A VCF file was generated using the same approach.

Afterwards, unzipped variant call files from samples with a mean coverage depth of ≥ 3× were merged using MultiVCFAnalyzer outside nf-core/eager[145]. Only SNPs with a minimum genotype quality of 30 and a depth of at least 3 were retained, applying base frequency thresholds of 90% for both reference and alternate allele calls. Positions not encompassed in the set of 3,124 informative SNPs described in the supplementary material of reference[24] were excluded. *M. lepromatosis* was specified as the outgroup for phylogenetic inference.

### Human parvovirus B19 phylogenetic analysis

Published ancient genomes from Eurasia[75,76] were downloaded and processed alongside the B19V reads extracted with MaltExtract v.1.7 from the Ibiza individuals. Sequences were aligned using bwa aln v.0.7.18 (parameters: -n 0.01 -l 16500 -o 2), deduplicated and filtered to retain reads of ≥ 20 bp in length[72,111]. Consensus FASTA files were generated with ANGSD v.0.940[115] with a minimum depth of 1 and a mapping quality ≥ 20, while removing transition to minimise damage-associated biases (-rmTrans 1). Consensus sequences from the B19V genomes from this study, previously published ancient genomes and modern references were then aligned with MAFFT v.7.526[170] using the options --auto --reorder --adjustdirectionaccurately. Modern reference genomes for the phylogenetic tree analysis were selected based on Dataset 4 from reference[76]. A maximum likelihood tree was reconstructed in IQ-TREE v.2.4.0[77] from the MAFFT alignment under the TIM3 + F + R3 substitution model selected by ModelFinder[78,171], with 100 bootstrap replicates and midpoint rooting.

### *Mycobacterium leprae* genotyping and phylogenetic analysis

SNPs not covered in at least 80%, 85% and 90% of the genomes were excluded using MEGA v.11[172]. The best-fit substitution model (GTR + F) was determined with ModelFinder, as implemented in IQ-TREE v.2.4.0[78,171] and maximum likelihood phylogenies were reconstructed using 100 bootstrap replicates. Note that seven hypermutated strains that failed the χ² test were excluded from phylogenetic analyses[173]. Additionally, maximum parsimony trees were generated in MEGA v.11 using 500 bootstrap replicates and the Subtree-Pruning-Regrafting method[173]. The maximum likelihood and maximum parsimony trees with the highest overall branch support within the 2F clade were selected for interpretation and visualisation. Both trees were rooted using *M. lepromatosis* and visualised in iTOL v.7, where branch colours and annotations were applied[174].

## Data availability

All newly generated sequencing data from this study, including raw FASTQ files and BAM files mapped against GRCh37, have been deposited in the European Nucleotide Archive (ENA) under accession PRJEB92203 (newly generated data, public access). Modern African reference datasets used for comparative analyses were obtained from controlled-access repositories, including the European Genome-

phenome Archive (EGA) and NIH dbGaP. Access to these datasets is restricted due to ethical and consent limitations imposed by the original studies. Researchers may apply for access through the corresponding Data Access Committees via EGA or dbGaP using the accession codes listed below. C.M.S. and C.A.F-L were granted data access to the Modern African reference datasets used for comparative analyses: AfricanNeo Modern datasets, EGAS50000000006 (controlled access, EGA) Fulani datasets, EGAS50000000451 and E-MTAB-8434 (controlled access, EGA/ArrayExpress) Sahel datasets, EGAS00001001610 and EGAS50000000451 (controlled access, EGA) African Genome Variation Project (AGVP) dataset, EGAS00001000959 (controlled access, EGA) EUROTAST dataset, EGAS00001002535 (controlled access, EGA) Amazigh/Berber dataset, EGAS00001003901 (controlled access, EGA) Ethiopia Genome Project dataset, EGAS00001000238 (controlled access, EGA) Chad dataset, EGAS00001001231 (controlled access, EGA) Khoe-San datasets, EGAS00001004459 and E-MTAB-1259 (controlled access, EGA/ArrayExpress) Cameroonian datasets, A-MTAB-679 and A-MTAB-678 (controlled access, ArrayExpress) Swahili dataset, EGAS00001002569 (controlled access, EGA) Mozambique and Angola dataset, E-MTAB-8450 (controlled access, ArrayExpress) WGS Central African RHG dataset, EGAS00001003722 (controlled access, EGA). Authorized NIH DAC granted data access to C.M.S. for the controlled-access genetic data deposited in the NIH dbGAP repository (accession code phs001396. v1.p1 and project ID 19895) [https://www.ncbi.nlm.nih.gov/projects/gap/cgi-bin/study.cgi?study_id=phs001396.v1.p1] (controlled access, dbGaP). Access to these controlled datasets is granted via the corresponding Data Access Committees (DAC) for each study. For individual accession codes, information on applying for access can be found on the respective EGA or dbGaP pages. We also used the following datasets in this study: i) Allen Ancient DNA Resource (AADR). We used the curated ancient human genome dataset compiled by Mallick & Reich (2023)[66], The Allen Ancient DNA Resource (AADR): a curated compendium of ancient human genomes, Harvard Dataverse, V9. DOI: 10.7910/DVN/FFIDCW. As far as we are aware, this dataset does not have a formal accession number. The dataset is publicly available via the AADR repository: https://dataverse.harvard.edu/dataset.xhtml?persistentId=doi:10.7910/DVN/FFIDCW. ii) Curated 1 K + HGDP dataset. We used the dataset described by Koenig et al. (2024)[32], which provides a harmonized resource of deeply sequenced modern human genomes (including 1000 Genomes and HGDP samples). The dataset is publicly accessible at https://gnomad.broadinstitute.org/news/2020-10-gnomad-v3-1-new-content-methods-annotations-and-data-availability/#the-gnomad-hgdp-and-1000-genomes-callset, and instructions for downloading the data are provided at https://github.com/atgu/hgdp_tgp. Both datasets were obtained from publicly available repositories, and we used the versions as released in these publications. All human remains analysed in this study are under the custodianship of the Consell Insular de Ibiza, curated under the scientific direction of Glenda Graziani and Juanjo Marí, the officially designated directors of the archaeological intervention. Access to the remains for future research must be requested from the Consell Insular de Ibiza, which holds legal ownership and permitting authority, in accordance with Spanish heritage legislation (Law 16/1985) and the Balearic Islands regional heritage law (Law 12/1998). The archaeological materials are held in custody by the Archaeological Museum of Ibiza and Formentera, and any research request must be submitted to the Heritage Department of the Consell Insular de Ibiza and authorized jointly with the formal approval of the Archaeological Museum of Ibiza and Formentera.

## Code availability
Custom scripts used in this study are available on GitHub and permanently archived on Zenodo[119].

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

## Acknowledgements

We acknowledge support from the Swedish National Genomics Infrastructure (NGI) in Stockholm funded by Science for Life Laboratory (SciLifeLab), the Knut and Alice Wallenberg Foundation, and the Swedish Research council, and the computations were enabled by resources in project provided by the National Academic infrastructure for Supercomputing in Sweden (NAISS), funded by the Swedish Research council through grant agreement no. 2022-06725. This work was supported by Swedish Research Council project id 2019-00849_VR and ATLAS (Riksbankens Jubileumsfond). C.A.F-L was funded in part by Bertil Lundman's Foundation and the Marcus Borgström Foundation. We also acknowledge Benjamin Guinet, Maria A. Spyrou, Caitlin Mitchell and Emily Gaul for general advice on microbial phylogenetics. Finally, we thank all the reviewers for the time invested in improving our manuscript through their thoughtful and constructive comments.

## Author contributions

Conceptualization: R.R.-V.; A.G.; A.G.R.; Experiment: R.R.-V.; V.L.; A.L.; Data analysis: R.R.-V.; Z.P.; A.S.-M.; R.Y.; C.F.-L.; A.G.R.; N.M.-G.; J.M.; G.G.; A.F.A.; M.V.; L.L.-F.; L.R.A.; P.P.-R.; M.K.; C.M.S.; A.G.; and Writing: R.R.-V.; Z.P. and A.G. with the input of all co-authors.

## Funding

## Competing interests

The authors declare no competing interests.

## Additional information

¹Centre for Palaeogenetics, Stockholm, Sweden. ²Department of Archaeology and Classical Studies, Stockholm University, Stockholm, Sweden. ³Departamento de Historia, Geografía y Comunicación, Universidad de Burgos, Burgos, Spain. ⁴McKusick-Nathans Institute and Department of Genetic Medicine, Johns Hopkins University School of Medicine, Baltimore, MD, USA. ⁵Human Evolution, Department of Organismal Biology, Evolutionary Biology Centre, Uppsala University, Uppsala, Sweden. ⁶Aranzadi Science Society, San Sebastián, Spain. ⁷Cranfield Forensic Institute, Cranfield University, Cranfield, UK.

⁸School of Anthropology and Museum Ethnography, University of Oxford, Oxford, UK. ⁹Independent researcher, Ibiza, Ibiza, Spain. ¹⁰Departament de Ciències de l'Antiguitat i Edat Mitjana, Universitat Autònoma de Barcelona, Barcelona, Spain. ¹¹Consell Insular de Formentera, Balearic Islands, Formentera, Spain. ¹²Department of Geological Sciences, Stockholm University, Stockholm, Sweden. ¹³Departament de Biologia Evolutiva, Ecologia i Ciències Ambientals (BEECA), Facultat de Biologia, Universitat de Barcelona (UB), Barcelona, Spain. ¹⁴Social History of Capitalism research group (SHOC), Vrije Universiteit Brussel (VUB), Brussels, Belgium. ¹⁵Archaeology, Environmental Changes & Geo-Chemistry research group (AMGC), Vrije Universiteit Brussel (VUB), Brussels, Belgium. ¹⁶Department of Coevolution of Land Use and Urbanisation, Max Planck Institute of Geoanthropology, Jena, Germany. ¹⁷Palaeo-Research Institute, University of Johannesburg, Johannesburg, South Africa. ¹⁸Center for the Human Past, Department of Organismal Biology, Uppsala, Sweden. ✉e-mail: ricardo.rodriguez.varela@arklab.su.se; anders.gotherstrom@arklab.su.se

