## [Transparent Peer Review file · Nature Communications]

Genetic diversity and health in medieval Islamic Ibiza revealing North African European and Sub-Saharan ancestries

Corresponding Author: Dr Ricardo Rodríguez-Varela

Version 0:

Reviewer comments:

Reviewer #1

(Remarks to the Author)

• Key results: Please summarise what you consider to be the outstanding features of the work.

This paper presents unpublished results on the origins of the populations that arrived in the Iberian Peninsula during the medieval Muslim period, beginning with the initial 8th-century Islamic invasion and continuing through later waves of settlement. It further traces these movements to the subsequent arrival of related populations in the island of Ibiza. The contribution is highly significant, particularly for its use of ancient DNA analysis to determine geographic provenance. The refinements applied to these genetic analyses—such as accounting for the number of generations elapsed since migration or detecting the dietary influence of marine resources—further enhance the precision and interpretive value of the results.

The historical implications are equally substantial. While many of the findings correspond with information preserved in medieval written sources, such texts must always be approached critically, as they often reflect political, religious, or cultural biases and do not necessarily capture the full demographic reality. Moreover, the Balearic Islands do not possess the written accounts that other places have, in part due to the late conquest of the islands (902/903). Therefore, this paper provides concrete evidence on the origins of Muslim populations, that has far-reaching implications: it deepens our understanding of their racial and ethnic composition, sheds light on patterns of migration and assimilation, and informs interpretations of their cultural, social, and ecological contexts within al-Andalus and the western Mediterranean.

Finally, the radiocarbon dating of the individuals analysed provide the necessary information to set the results in its correct historical frame. For instance: the presence of muladies (non-african muslim population) along with black population from Subsaharian Africa reveal which was the pattern of conversion and acculturation of Iberia after 711. Again, the fact that the research is centered on the Balearic Islands fill that gap that the islands have for this time period.

• Validity: Does the manuscript have flaws which should prohibit its publication? If so, please provide details. NO

• Originality and significance: If the conclusions are not original, please provide relevant references. On a more subjective note, do you feel that the results presented are of immediate interest to many people in your own discipline, and/or to people from several disciplines?

The conclusions are original in what I have mentioned earlier: they provide empiric prove of part of the information that we knew from written sources and archaeological work. However, this new data is much more refined and provides microinsight in information we could infer from the rest of the data but not prove empirically. For instance: the subsaharian population with a precise origin, the number of generations that have passed since the arrival to the Iberian Peninsula and Ibiza. Moreover, north-African and middle-eastern populations have been clearly recorded in the texts but this genetic analysis refines what we know about this influx.

The other thing is that the genetic analysis can trace the pathologies and epidemics. The radiocarbon results provide exact chronology to the bodies which can then be studied in a precise historical context.

• Key Data:

- Radiocarbon dating of all samples analysed which provides the necessary accurate information for historical implications

- Control analysis: to refine results. Super important. Impossible to do it without high resolution research and it offers the corrections needed to have a very full and accurate picture of the situation

- High resolution data of origin: Subsaharian Africa is a whole continent. Research conducted identified different origins both from the west and the east.

- It's focused on an island that has less information about its Muslim past than others.

• Conclusions: Do you find that the conclusions and data interpretation are robust, valid and reliable?

Yes, they are. One of the most enduring and vivid debates concerns the ethnic background of the populations that arrived in the Iberian Peninsula in 711 and in the subsequent centuries. Some researchers argue that these contingents consisted solely of soldiers, with no significant civilian migration, and that these soldiers eventually intermixed with the local population. Others point to the transfer of agricultural knowledge, seeds, technologies, and other cultural elements as evidence of broader population movements. This paper's findings support the latter view, indicating a gradual demographic shift from a predominantly European population to one with substantial North African ancestry. Furthermore, the ability to trace lineages across generations provides additional confirmation of a steady process of migration and intermixing, as well as the presence of European converts from the very beginning of the conquest in the 8th century. In addition, the Balearic Islands—particularly Ibiza—are often underrepresented in the historical record. Written references are relatively scarce, and the islands are frequently treated as a single unit, with their history assumed to follow that of Mallorca. This tendency makes it difficult to isolate the specific historical trajectories of each island, except in certain well-documented events. In this context, the paper's provision of precise, individualized data for Ibiza represents a significant and original contribution, offering new insights into the island's unique role within the broader medieval Muslim Mediterranean world.

• Suggested improvements: Please list additional experiments or data that could help strengthening the work in a revision.

1. The paper would benefit from the inclusion of a photograph of the maqbara (excavation). Such an image would provide a visual context for readers unfamiliar with the site, enhancing their understanding of the archaeological setting. Ideally, this figure should be included either in the introduction or in the section where the burial context is first described.

2. The historical background is informative and provides a strong foundation for the study. However, the introduction section as a whole is somewhat disorganized. There are frequent mentions of results scattered throughout the introduction, sometimes before the historical or methodological framework has been established. This disrupts the logical flow and may confuse readers who expect the introduction to focus on background, research context, and objectives.

I recommend restructuring the introduction and not include the results of the research that are already included in the next section and in the abstract.

Finally, maybe a couple of lines about the research gap that the study is trying to solve and a very brief list/overview of the methodology, without revealing specific results.

• Clarity and context: Is the abstract clear, accessible? Are abstract, introduction and conclusions appropriate

Abstract

The abstract is clear and accurately states the hypothesis, methods, and results. It effectively summarizes the purpose and findings of the research.

Historical Introduction

The historical background is informative and provides a strong foundation for the study.

The introduction does not need to contain detailed results, as these are already well covered in the abstract and in the dedicated results section (comments above)

Main Body

The remainder of the paper is excellent and makes a significant scholarly contribution. The results are especially valuable for the integration of written historical sources and archaeological evidence. This multidisciplinary approach allows for a more nuanced understanding of the kind of population that arrived in Iberia and the Easter Islands of al-Andalus (Balearic) during the Middle Ages, and should be emphasized as one of the study's key strengths

• Please indicate any particular part of the manuscript, data, or analyses that you feel is outside the scope of your expertise, or that you were unable to assess fully.

I am not familiar with the genetic analysis conducted. I am familiar with the time period, questions asked, research about medieval population, debate about origins and influx of population from Africa during the Middle Ages, excavations with similar questions and implications of these types of analysis.

Overall Recommendation

This is a valuable and original study. With a more clearly structured introduction and the addition of relevant visual material, the paper's clarity and impact will be further strengthened. The results presented here will be warmly received by the scholarly community working on al-Andalus, which has long debated—often for decades—key issues such as the processes of Islamisation in Iberia, the extent and nature of population arrivals (whether limited to military elites or involving broader civilian migration), the conversion of the local Iberian—Roman population (muladies), the degree of racial and ethnic intermixing, and the prevalence of specific pathologies. These debates have traditionally relied heavily on medieval textual sources and, to a lesser extent, archaeological excavations. By introducing robust empirical evidence through genetic analysis, this paper provides a new and independent line of inquiry that not only illuminates these long-standing questions but also has the potential to reorient current theoretical frameworks. The study stands as an important contribution that bridges historical narratives and scientific data, offering a more nuanced and evidence-based understanding of population dynamics in medieval al-Andalus, particularly in the margins of the empire (the islands).

Additional questions: when referring to individuals that were used from Northern Iberia for comparison (Burgos, Pais Vasco, etc.) there should be a specific reference to where they are coming from. If there

is one, I couldn't find it.

Reviewer #2

(Remarks to the Author)

In the manuscript "Genetic Diversity and Health in Medieval Islamic Ibiza: Unveiling North African, European, and Sub-Saharan Ancestries", Rodríguez-Varela et al. generate genome-wide data from 13 individuals from an Islamic cemetery in Ibiza dated to 950–1150 CE. The authors infer the geographical ascription of the individuals, providing information on population movements in the Balearic Islands during the Islamic period, which included individuals of North African, European and sub-Saharan African origin. They also explored the metagenomic profiles of the individuals, detecting the presence of several pathogens, including *Mycobacterium leprae*.

I think that the authors have performed a comprehensive exploration of the data and most conclusions are well supported by data analyses. The research question around which this manuscript is centered is interesting and the results are compelling. The manuscript is well-written and it is evident that it has benefited from previous reviews. I have to acknowledge the effort made by the authors to incorporate evidence from different fields in order to better understand the Islamic period in Ibiza. Potentially, this manuscript will be of interest to diverse fields, including palaeogenomics, archaeology, evolutionary biology and population genetics, and it is a good fit for Nature Communications. However, I think that there is one aspect that should be revised before publication.

The RfMix analysis, including the decay of local ancestry, is really interesting, but I'm not sure the approach is appropriate. The ancient individuals, including the Guanche from the Canary Islands, are modeled as the admixture of European, North African and Middle Eastern components. However, because of prehistoric and historical migrations in the region, the Mozabites are the result of admixture from these three components, so what is considered North African regarding local ancestry? Also, given the East to West cline in the ancestral Maghrebi component in North Africa, are Mozabites a reasonable proxy for the individuals arriving in the Iberian Peninsula during the Islamic period? If not, is it possible that the identification of European or North African segments might be skewed? For example, the Guanche individual predates the Islamic expansion in North Africa and can be considered an isolate in the region; however, the local ancestry results show some long segments of European ancestry. How should we interpret those segments? Can they be considered the result of recent European admixture? Or are these segments an artifact created by differences in the admixture patterns between Guanches and Mozabites? Again, although the analysis has the potential to provide interesting information on the Islamic population in Ibiza, I think the authors should consider all these limitations and discuss them in the paper.

Minor comments:

- The authors indicate in the introduction that the "Maqbara of Medina Yabisa" cemetery contains 125 individual burials, but in the results section, they refer to 41 excavated individuals. I assume this is because part of the cemetery has been not excavated yet, but maybe it would be better to clarify this issue in the introduction.

- Figure 1: Probably it is because of space constraints, but the text in Figure 1B is too small. Would it be possible to increase the font size a little?

- Figure 3: It is impossible to see the ADMIXTURE proportions of the last group of ancient Iberians in the plot before zooming in, as it appears as a black rectangle. I would recommend removing the dividing black lines between individuals or removing that section from the general ADMIXTURE plot and only including it in the zoom-in version.

- Second paragraph, page 1 of the Supplementary text: Add a parenthesis before "see that paper...".

- Figure S3, Figure S4 and Figure S11: Same as Figure 3. For the K=2 to K=9 plots, it is very difficult to see the results for the ancient individuals identified with codes 1 to 15 due to the lines that divide each column. I recommend removing the lines between samples to improve the readability of the figures.

Reviewer #3

(Remarks to the Author)

N.B: Best way to view this review is to copy into a text file and use a Markdown (pre)viewer (e.g. your IDE)

Summary

In this palaeogenomics study, Rodríguez-Varela and colleagues performed standard human population genomics and microbial genomics analyses on ancient human remains from a cemetery on Ibiza dated to the al-Andalus period of Iberian history.

For the human DNA, they perform a suite of broad overview analyses to identify source and timing of the ancestral origins of the 13 different individuals yielding sufficient DNA.

For the microbial DNA the authors screened the off-target reads for traces of potential pathogens - both bacteria and viruses, that yielded sufficient genomic coverage from a single individual to place this ancient genome on the phylogenomic tree of *M. leprae*.

Please note my technical expertise is in the microbial genomics, thus I focus my review primarily on this part of the manuscript.

Overall the period the study is generally rather understudied from a palaeogenomics point of view, making the results relevant, particularly given this specific era of Iberian history is an interesting one due to the existing rich historical and archaeological contextual information in addition to the large cultural changes and movements happening during this time frame.

Methodologically, for both the population genomics (to my knowledge) and microbial genomics there is nothing particularly new, but all methods are sound and following standard tools and analyses.

My main issues with the paper (details below) are:

1. Limited integration of the archaeological results (and overextended language in the discussion about) despite being listed as a primary aim
2. Overly strong language in the discussion about the impact of findings of the microbial genomics results, and limited discussion of comparison to other genomes from the period (although the methods are sound!)
3. Very insufficient details on methods for reproducibility of the analyses

General

For the general comments of my review, I will use the criteria specified by the journal

What are the noteworthy results?

The main results of the paper are consolidating historical and archaeological timelines of the settlement and population dynamics within Ibiza during the Islamic rule, refining these models with genetic data.

Contribution of a new ancient *M. leprae* genome from another location in Spain in this time period.

Will the work be of significance to the field and related fields? How does it compare to the established literature? If the work is not original, please provide relevant references.

The methods (smartpca, ADMIXTURE, f3-statistics, KinSHIP; aMeta, raxML, MEGAHIT) and questions are rather routine in the area matching similar recently published papers, but generally the results are contributing to an understudied period.

Does the work support the conclusions and claims, or is additional evidence needed?

Generally the claims made in discussion match the results: in most cases the results are just confirming the existing archaeological (e.g. stable isotopes, radiocarbon) and historical evidence but via a new perspective and providing greater nuance.

Are there any flaws in the data analysis, interpretation and conclusions? Do these prohibit publication or require revision?

I do not see anything preventing publication, however some of the language in the discussion and interpretations are over blown and should be revised, particularly in regards to the ancient microbial results.

In particular, I take issue with statements such as:

> (INTRO) integrate genetic, historical, and archaeological evidence to reconstruct population history

> (DISCUSSION) This study presents a comprehensive archaeological, genetic and metagenomic analysis of individuals from an Islamic necropolis in Ibiza

Despite this, there is actually scant archaeological context described about the necropolis nor the individuals.

The description of the osteological results is extremely minimal even though it provides a huge source of information for that is important on statements on health and disease interpretation - the only mention of this is a single sentence in the discussion (referring to results that are not described elsewhere).

Another archaeological result relevant to the human population genomics analyses would be the characteristics of the graves of the individuals themselves - are there any form of grave goods present? What other links could be made about this and the results from the genomic analyses?

But this is not explored.

Even *_if_* there is very little material left in the graves (something not really discussed), thus making further interpretation difficult, this should at least be stated.

I would argue much of the discussion is combining the broader historical context, which is fine, but it is not integrating the archaeological context of the specific site or individuals.

The other set of too strong language is the impact of the ancient microbial results:

> reveal a diverse range of pathogenic and commensal microbial DNA, offering new insights into the disease landscape of medieval Ibiza

Only 4 taxa are mentioned, 2 are commensal viruses, and *S. pneumoniae* was not discussed further (for reasons that are not clear) - the only major result is a single individual carrying an leprosy genome of a standard genotype of the time period and region.

While I appreciate that the authors were careful with authenticating the report hits, this is definitely not a 'landscape'.

Please be careful of using (what I know is the currently 'in vogue') terminology and keywords within the field that are overly ambitious.

> Altogether, these results suggest that medieval Ibiza was not only genetically diverse, but also embedded in complex networks of disease transmission

I don't think you can state the viruses genetically diverse if they are all from the same genotypes (for B19V), and only two genotypes (HBV).

Also I would be wary to state 'embedded in networks of transmission' given there is no actual analysis of this: there is no analyses on the exact temporal or spatial relationships of the new *M. leprae* genome to the existing ones from the nearby Barcelona genomes from the same period with linking to the specific ancestry or mobility information!

> The presence of *M. leprae*, HBV and B19V in several individuals adds a crucial layer of understanding to the social and environmental context of medieval Ibiza, further illustrating the intricate relationship between migration, health, and genetics in the region.

> underscores how disease was closely intertwined with human mobility and interaction during this period

Again I find the language to be rather flowery.

Neither adding a single *M. leprae* genome that is from a genotype that is spatially and temporally typical, nor detecting very common commensal viruses (which many palaeogenomics studies are now finding everywhere, such as Muehleemann 2018, Kocher 2021 etc.), are adding 'crucial' insights.

With insufficient sample size (for which I do not blame the authors) to do a proper network analysis to compare the 'mobility' of the pathogens in relationship to the ancestry of the individuals, I think this is not exactly illustrating 'intricate relationships' of health.

Indeed - given the lack of any other evidence towards the health status across all individuals osteologically (see above), particularly individual s.313 - it's hard to make any links between ancestry and health status in this study - particularly given individual s.313 is a typical European individual.

Overall, I strongly recommend the authors to include more archaeological and osteological context/results to either the (otherwise very nicely written!) introduction and/or results, and tone down the language about the links with health.

Is the methodology sound? Does the work meet the expected standards in your field?

The methods used are standard in both ancient population-genomic and ancient microbial fields, and the authors report sufficient ancient DNA authentication metrics (damage patterns, coverage, contamination estimations, evenness of coverage)

The authors have demonstrated awareness of potential issues, for example investigating potential technical artefacts in the detection of *M. leprae* in another sample (e.g., index hopping as described in the supplementary text)

Is there enough detail provided in the methods for the work to be reproduced?

While the choices of methodology and analyses is sound, the work is **definitely not reproducible**.

The methods description only describes the names tools and only sometimes the versions - but even these alone definitely does make not make the results reproducible.

There are rarely references to options, settings, or (metagenomic) reference databases.

I strongly encourage the authors to upload all code, scripts, config files, and commands to e.g. a GitHub/GitLab/Bitbucket repository and archive this on Zenodo.

The only exception to this is if most of the tools were run with default settings, in which case this should be mentioned for each tool, but this would make me question the quality of the analyses.

Note the reporting summary file is also quite poorly described, missing many of the tools - however I recognise this is a terribly implemented system by Nature for better reporting so I do not criticise the authors on this.

Specific

Text

- Page 3, para 4: 'Tools such as', although I know you developed aMeta, just cite the different tools that can be used rather than singling out your own tool given you say 'such as', your tool may not always be the best (as you demonstrate for viruses here!)

- Page 4, para 2: 'From the 41 individuals...' this is one example of where I was surprised by the lack of archaeological description of the graves/archaeological context - particularly as you say you integrate archaeological results

- Page 5, para 2: 'We present the estimated' - I found it a bit jarring the results started with the genetic results and then jumped back to the osteological results. I would move the osteology to the top of the results and expand this to include more archaeological descriptions - particularly as you refer to how the DNA was extracted/sequenced here (i.e., after you already started describing genetic results)

- Page 13, para 3: 'To explore...' How did you define which pathogens you screened for? A reader might think something like - well did you really only detect these 4 species, maybe you missed an obvious one that I could go back and re-analyse to find (which they should have the freedom to do so!)

- Page 13, para 3: 'defined as likely...' What reasoning did you use to define *B. lata*, *C. tetani* etc. as contaminations?

- Page 13, para 3: 'detected in individuals' be more precise in your language - without further validation (trees, unique marker genes), you have not detected *S. pneumoniae* and *M. leprae* at this stage. You have just found reads that matched/aligned to the reference genome etc.; these could still be theoretically commensal relatives at this stage (particularly for *S. pneumoniae* given *Streptococcus* is such a broad genus).

- Page 13, para 3: 'Only *Streptococcus*...' Why do you mention *S. pneumoniae* here and then never again? Why did you exclude this from any further analysis? I find this a glaring unexplained omission

- Page 13, para 3: 'Only *Streptococcus*...' It is also important to describe (something that was not done) what is in your database. If you look at NCBI RefSeq, *S. pneumoniae* is massively overrepresented with having more than 8000 genomes, whereas there is only *M. leprae* genome - is this why you ignore *S. pneumoniae* - was it a false positive?

As proof:

```
```bash
$ wget https://ftp.ncbi.nlm.nih.gov/genomes/refseq/assembly_summary_refseq.txt
$ sed -i 's/#assembly/assembly/g' assembly_summary_refseq.txt
```
```

in R

```
```r
> library(tidyverse)
— Attaching core tidyverse packages
```

```
tidyverse 2.0.0 —
✓ dplyr 1.1.4 ✓ readr 2.1.5
✓ forcats 1.0.0 ✓ stringr 1.5.1
✓ ggplot2 3.5.1 ✓ tibble 3.2.1
✓ lubridate 1.9.3 ✓ tidyr 1.3.1
✓ purrr 1.0.2
— Conflicts
```

```
tidyverse_conflicts() —
✖ dplyr::filter() masks stats::filter()
✖ dplyr::lag() masks stats::lag()
i Use the conflicted package to force all conflicts to become errors
> raw <- read_tsv("~/Downloads/assembly_summary_refseq.txt")
Rows: 465987 Columns: 1
— Column specification
```

```
Delimiter: "\t"
chr (1): ## See ftp://ftp.ncbi.nlm.nih.gov/genomes/README_assembly_summary.txt for a description of the columns in this file.
```

i Use `spec()` to retrieve the full column specification for this data.

i Specify the column types or set `show\_col\_types = FALSE` to quiet this message.

Warning message:

One or more parsing issues, call `problems()` on your data frame for details, e.g.:

```
dat <- vroom(...)
```

```
problems(dat)
```

```
> data_raw <- read_tsv("~/Downloads/assembly_summary_refseq.txt", skip=1)
> data_raw |> select(organism_name) |> arrange(organism_name) |> group_by(organism_name) |> summarise(n = n()) |> arrange(desc(n))
A tibble: 108,361 × 2
```

```

organism_name n
<chr> <int>
1 Escherichia coli 41955
2 Klebsiella pneumoniae 24660
3 Staphylococcus aureus 14518
4 Pseudomonas aeruginosa 11329
5 Acinetobacter baumannii 10476
6 Streptococcus pneumoniae 8972
7 Listeria monocytogenes 5502
8 Mycobacterium tuberculosis 5359
9 Enterococcus faecium 4583
10 Salmonella enterica 4212
> data_raw |> select(organism_name) |> arrange(organism_name) |> group_by(organism_name) |> summarise(n = n()) |> arrange(desc(n)) |> filter(organism_name == 'Mycobacterium leprae')
A tibble: 1 x 2
 organism_name n
 <chr> <int>
1 Mycobacterium leprae 1

```

- Page 14, para 1: 'Screening below...' Why did you start looking below your thresholds? It seems a bit contradictory to set one then 'ignore' it - it comes across as fishing for results. What is the problem with the original thresholds? EDIT: Later on you justify this that aMeta isn't good for Viruses - but you should briefly state this here (e.g. 'We then looked below this threshold due to issues with the aMeta score for detecting viruses (see discussion)') as I immediately became suspicious on the first reading of this.
- Page 14, para 1: 'Detection of HBV' I don't understand what you mean by - 'Detection of HBV in s.167, with additional HBV reads using MaltExtract in <...>' additional to what? What is the difference between detection and just having reads?
- Page 14, para 1: 'Damage plots count only be generated' - is this the MaltExtract damage plots, or did you do alignment and then use of e.g. mapDamage?
- Page 14, para 1: 'longest contigs' -> you've not said you did \_De novo\_ assembly until the next sentence (if that is what you are referring to?)
- Page 14, para 1: 'De novo assembly and BLASTn' why do you mention BLASTn as the tool specifically and not the tool you used for assembly?
- Page 14, para 1: 'Interestingly, HBV' why is it interesting that it's 3/4 being women? You have a low total count of individuals, so just through stochastic sampling this may not be particularly meaningful
- Page 14, para 2: 'reads respectively' I'm a little confused why you give very few statistics about \_S. pneumoniae\_ and \_M. leprae\_ but are going into so much detail about the metagenomic results of the viruses, could you go back and provide more of these basic stats for the bacteria too?
- Page 14, para 2: same again as above, slightly more detail about the assembly would be nice (at least tool name)
- Page 14, para 3: 'atypical coverage profiles' what do you mean by this exactly? Is this depth of breadth coverages? What are you expecting in 'normal' ones, and how do these atypical ones look like? It would be nice to see more in the discussion what the limitations are of aMeta with viruses (something that I do not really see), and given the authors are also the developers of that pipeline, how do you plan to (or could theoretically) address this - this could be a useful knowledge for the wider aMeta user community to know how to address this limitation.
- Page 14, para 3: 'This bias may be...' it's nice you go into more detail on the last sentence with my previous comment, but you should lead the transition of bacteria to virus results on this page with this, as this explains to the reader why you went below the original aMeta threshold, and expand on it in the discussion (as I explain above)
- Page 14, para 4: 'Targeted enrichment' why did you only capture \_M. leprae\_ even though you mentioned it at the beginning of the metagenomics results section?
- Page 14, para 4: 'substantially increased' You mention substantially increase read yield to 128k, but you didn't report how many reads you had in the shotgun data in the previous section (see my point a couple above about giving more basic screening statistics for the bacteria)
- Page 15, para 1: 'manual genotyping' what does 'manual' mean in this case? This rings alarm bells to me, and what that involves is not further expanded in the methods sections - are you literally just looking visually at each position and making a decision? What rules did you apply to picking each allele?
- Page 15, para 1: 'The Ibiza genome' Don't change the name to Ibiza genome, refer to it via the individual name as you've otherwise done throughout otherwise it will become confusing (or if you insist, say something like 'Ibiza genome from individual s.131').
- Page 15, para 2: 'In parallel...' please explain your reasoning to also make a maximum parsimony tree (which is arguably more simplistic than maximum likelihood, why not just Neighbour-Joining tree, or go the other way and make Bayesian trees [I know why, but other readers may not understand])?
- Page 16, para 1: 'Comprehensive archaeological...' This is too strong - it is definitely not comprehensive: there is no detailed description of archaeological context of each individual (grave goods, positioning etc) nor the site, the 'genetics' is purely population genomics - nothing on e.g. genetic health, and ultimately after screening the 'metagenomics' consists of a tree of only single species - not of the other potential pathogens. I would remove 'comprehensive'
- Page 16, para 1: 'offering new insights...' I would be wary of saying you have given an insight into the health status given you've only identified the presence of the microbe, while it is most likely that the individual did suffer despite no osteological signatures (in the sense the to have sufficient load to detect in an ancient DNA sample, there must have been a large number microbial cells). I would feel more amendable if you said 'infection status' or something similar. Alternatively you should provide more archaeological/osteological context about each individual so you have two sources of proof about the state of the health of the individual
- Page 16, para 2: 'our study focuses on...' this is the first time you mention the cemetery is urban, this compounds my criticism that you state this is a comprehensive study but actually there is very little archaeological description or context about the site nor burials
- Page 16, para 2: 'Individuals s.157 (XX)' why do you mention the biological sex in the first line? You don't do this for any other individual and does not seem to be relevant for any of the points made in this paragraph?
- Para 17, para 4: 'diverse range of' I would not agree you've revealed a \_diverse\_ range of pathogen and commensal microbes, you've reported 2 bacteria (only one actually analysed) and 2 very common viruses (unless this is what you mean by commensal?), if you want to state this sentence like this about this you should discuss more about the wider metagenomic profile, even below the aMeta score.
- Para 17, para 4: 'although osteological evidence...' you say evidence of leprosy has been found in other cemeteries, but you've not said so far if your individuals displayed this - this should be described in the introduction or results in an expanded archaeology/osteological description
- Para 18, para 2: 'Interestingly...' Again, you've only now mentioned that there is no differentiation in grave treatment, but you've not described this in the introduction (if previously published elsewhere) or in the results (if new)
- Para 18, para 2: see my point two above ('although osteological...') - this should have been described much earlier! And what do you mean by that the available remains are not sufficient? This is a critical bit of information that should be described in the results! It gives more context why you could not expand more on this topic, you should not hide this buried in the discussion!
- Para 18, para 5: 'This burden of infection' This might be being very nitpicky on my part, but I find the two sentences in the middle a little contradictory: you say burden of infection (implying a higher 'suffering'), but if it's the same as everywhere in else in Europe at the time, then then really it is just the basic microbial background of individuals. Do you have evidence that these viruses (in particular) are actually having a negative effect on the individual? Can you really call this disease? If they are everywhere they could be a normal part of our flora and it's more about abundance or disequilibrium that then causes health problems. Your results can't really show what the \_level\_ abundance of them are, and thus can't say anything about the \_burden\_ of these microbes are on the individuals (note that even though I've said this, I do not have a clinical background so apologies if burden of infection has a specific term in the literature that you have applied here that I'm not aware of)
- Para 18, para 5: 'crucial layer' people having very common microbes on their body is not new at all, so I don't think this is a 'crucial' layer, I would remove
- Para 18, para 5: the last sentence is too broad, you've not really made any real links or described implications between mobility/ancestry and the microbes, something you can only say when you have a large stratified number of genomes and strain analyses - all you are saying in this manuscript is this single genotype is present in this island - you have no other comparison or deeper analyses e.g. comparing with individuals on the continent in the same time frame (which again I don't blame you, there are not sufficient public data either to do this currently), but I would tone this down - i.e. you have not shown any intricacies.
- Page 19, para 2: 'genomic traces' don't downplay yourselves here, a 3x \_M. leprae\_ genome is not traces!
- Page 19, para 2: 'intertwined...' once again, you are not demonstrating where the \_M. leprae\_ came from per se (did the genotype come from the 'native' Iberians on the continent, or travelled purely through the lineage infecting newcomers from al-Andalus?) so I don't think you have demonstrated 'intricacies' of mobility and interaction, I would tone this down
- Page 19, para 3: 'By integrating...' I don't think you've really integrated pathogen detection and genomic ancestry, at least in what I have read here. You've not discussed in depth between the specific link of the microbial genotypes with the ancestry of the individuals they are coming from: your whole discussion is actually surprisingly not integrated across the different methods reported here, you have one section on ancestry and one on the microbes (other than vaguely broad 'people from Islamic cemeteries').
- Page 19, para 5: 'These methods were complemented by...' where are the results on this?! These are not described at all!
- Page 20, para 1: 'established protocols...' Was this following exactly with the same reagents (and in some cases critical consumables)? Please state if so, otherwise this is not reproducible! I would rather see a link to the "exact" protocol you used?
- Page 20, para 2: 'as described in...' same again: was this protocol followed using exactly the same reagents and consumables? Please state if so otherwise this is not reproducible! I would rather see a link to the "exact" protocol you used?
- Page 20, para 3: '(Design ID...):' This is very good, I commend the authors on being so specific! Thank you!
- Page 20, para 3: 'Novaseq S6000' I think that is a typo, shouldn't it be just 6000? But also, what kit was used? Was it paired-end, and what was the number of imaging cycles? This can be important information for users who may want to reanalyse your data (e.g. can be important for \_de novo\_ assembly of the metagenomic data)
- Page 21, para 1: 'were demultiplexed...' with what tool/settings/version? This isn't useful otherwise
- Page 21, para 1: 'Adapters were trimmed' was all adapter trimming with default parameters?
- Page 21, para 1: 'Using Burrows-Wheeler...' was BWA mapping also performed for the microbial work? Or is this just for the human pop gen?
- Page 21, para 1: 'merged using Samtools' which subcommand? There are many different ones?
- Page 21, para 1: 'a modified version' If modified, please make your modified version available on GitHub or similar, or describe the exact modifications made (but the former greatly preferred), otherwise this work is not reproducible
- Page 21, para 2: 'PMDtools', what version/settings?
- Page 21, para 2: 'contamination estimates...' why not refer to the two tools by name as you are doing elsewhere? What about versions and settings?
- Page 21, para 3: 'TrimBam from BamTuils' what version of BamTuils? What settings?
- Page 21, para 7: 'applied depending on specific analysis' what exactly does this mean? What were the criteria and when applied? Be explicit
- Page 22, para 2: 'we standardised the SNP identifiers' how exactly did you do this? With scripts in R or python? This seems like an important file to make available for people to reproduce your analyses
- Page 22, para 4: 'The Iberian component included 66...' why is there such an imbalance in the number of individuals between the European representatives and the Africans? Because of the greater genetic diversity in the latter? Please provide a justification (at least for non-specialists such as myself)
- Page 23, para 2: 'computed the covariance' - how were you doing this, in custom R/python scripts? Please make available if so.
- Page 23, para 6: 'NgsRelate, Kin, READ (v2)...' Why the versions for READ, and not the others, please at least be consistent (or rather versions and settings for every tool used here! Or make your scripts/commands available on e.g. a GitHub repo so a reader can look directly and so you can save time in writing!) - Please check this across the entire of the methods section as this keeps coming up (I won't mention this again, but definitely keeps happening post this comment)
- Page 24, para 2: 'choosing up to 10 individuals' why 10? Please justify
- Page 24, para 3: 'Using the Namibian' why did you pick the Namibian individual?
- Page 24, para 4: 'pipeline aMeta' - what settings, and more importantly for metagenomics, which database!? This is a hugely important thing in metagenomics and greatly influences your output! Furthermore, as aMeta is using a snakemake, it should be easy to provide the exact configuration file you used, correct?
- Page 25, para 1: 'were retained for downstream analysis' how exactly did you do this filtering, in R/Python? And just for clarity: are all of those criteria are considered \_within\_ the authentication score, or

you were filtering for the aMeta score AND the rest of the criteria but manually?

- Page 25, para 1: 'specific cases' - does this only refer to *M. leprae* not other species otherwise not described (assuming you're referring to the index hopping section of the SI)?
- Page 25, para 2: 'Calculating the unique...' How did you calculate the k-mer to read ratio? Do you have a script or notebook?
- Page 25, para 2: 'Known human primary pathogens' why did you go away from the aMeta score? Please justify
- Page 25, para 3: 'We then used MaltExtract, within aMeta or manually?
- Page 25, para 3: 'BLASTn on HPC' what is this exactly? I've not heard of it and I don't see it referenced in the BLAST+ paper? What database?
- Page 25, para 3: 'NCBI nucleotide databases' what date/version of the NCBI nt database did you use? This database is updated on a regular basis
- Page 25, para 3: 'Additional damage' - from what files did you exactly generate these from? DamageProfiler only takes BAM/SAM files (I believe)? Or from MaltExtract? Or mapping reads back to your assemblies?
- Page 25, para 4: 'nf-core/eager...' as with snakemake above, please provide the config file and/or command used, note that nf-core/eager should provide a full list of versions of all tools and citations used within the pipeline somewhere that you can use here.
- Page 25, para 5: 'damage removal was applied.' to which FASTQ files were you applying this too? Post-capture or raw I'm assuming post-capture but this is not described? Is this still happening within nf-core/eager or manually?
- Page 25, para 6: 'more stringent edit distance' are you referring to bwa aln mapping here?
- Page 26, para 2: 'Mycobacterium lepromatosis FASTQ' what GenBank assembly/strain is this associated with, is this the designated *M. lepromatosis* reference genome? And if not has it been validated as a good quality outgroup for your phylogenomic analyses?
- Page 26, para 3: why the 90% cutoff? If you've got a 3X genome why not 66% or even just 100%?
- Page 26, para 4: 'Manual genotyping' Same again about manual genotyping as I commented in the results section, please describe what you mean by this and what were your criteria/rules for making each call
- Page 26, para 5: 'B19V reads extracted' Why only the the specific B19V reads for assembly, what if you've lost some conserved reads to closely related taxa reducing your coverage?
- Page 26, para 5: 'Sequences were aligned using' is this with nf-core/eager as above or manually?
- Page 26, para 5: 'fields' typo!

### ### Figures

As a general criticism: the population genomics figures are often quite poor quality in the sense the size ratio between the plots and texts are poor. For example, pretty much all the PCA figures it's impossible to read or distinguish anything because:

1. the main text figures are too small (so it's hard to distinguish between so many clusters when there are so many colours)
2. the font size is tiny so it's extremely difficult to read. The supplementary figures are slightly better as they are larger, but the label and axis text are almost always too small.

In contrast: the microbial genomics figures are generally much more readable and and balanced - for example Figure 9. In cases where the 'contents' of the figure is large and detailed, the font is naturally small. However these has a balanced ratio with the figure components themselves (there is not enough free whitespace to make the text any bigger - such as the large trees in the supplementary), making it acceptable. However in the case of the pop-gen figures there is a lot of free space to make the fonts bigger enough to make them readable..

My only criticism of the microbial genomics figures is the on the two *M. leprae* trees it's very difficult to find the Ibiza genomes as the tree is so large - the red font is not enough to catch a readers eye - please use a larger indicator (e.g. a star)

Specific comments:

- Fig. 3: I have no idea what the span of the ancient iberians is referring to, nor was it immediately clear what areas the 'Zoom' was referring to, I found the different font sizes across all the plot distracting too (e.g. the smaller sized pale blue 'pairwise similarity' was very hard to read)

### ### Supplementary Tables

- General: a summary of what each table is referring to would be nice (and ideally a legend for each table what each column is referring to)
- TableS1 appears to have broken coverage column headers for 'Damage' and 'aDamage' (I'm not sure what they mean, nor what the ']' standards for)
- TableS1: please make cleaner/computer readable columns by putting units in headers when they are all the same (e.g. for the dates)
- Table S2d: the column headers appear to be missing? I'm not sure what each one is referring to
- Table S6 and S7 starts referring to samples with '[A,B,D]' suffixes... what are these?

### ### Misc

One last half-throw away potentially poorly thought through comment as I submit this: have the authors considered the wider impact these results may have in the present day social tensions between 'Western' and Islamic societies?

Have you thought about the potential ramifications or implications of reporting an association between - what could be interpreted on a very shallow reading of the paper by e.g. Daily Mail journalists and readers - between Islamic people and pathogenic disease?

Generally I have found ancient population genomicists in the past have tried to 'hide' between the 'purity' of the science method when reporting their results on ancestry to avoid having to consider what modern society may make of it. This may half work in older time periods but as we (rightly, and is a strength of this paper) move into less studied historical periods, where society today feels a much closer 'connection', I think primarily ancient population genomicists groups who start to combine ancestry and cultures with microbes (that are often seen as 'dirty' or 'bad') should try to be more careful in how they report things for this reason.

For example, as I have mentioned above in my specific comments - when describing the *M. leprae* genome you don't really discuss the detection of this in the context of the ancestry this specific individual has (i.e. falling within a typical European area of the PCA space). It might be good to make this more explicit, as otherwise a (e.g.) Daily Mail journalist will shallowly skim read the paper as: 'Islamic invaders into Ibiza from the Arab countries brought Leprosy with to the Island!' When in reality, we do not have the sample size or temporal granularity to know know where that specific lineage of Leprosy comes from, but for sure the individual carrying *M. leprae* is not from the first wave of invaders.

Anyway, like I said throw-away rambly comment, but maybe some food-for-thought?

### Reviewer #4

(Remarks to the Author)

The article is a very interesting attempt to generate and interpret ancient DNA, radiocarbon dating, and metagenomic data to explore ancestry and disease among medieval Islamic inhabitants of Ibiza (then part of Al-Andalus). It reconstructs population dynamics, admixture, and health during a period of strong North African and Mediterranean connectivity.

I will not comment on the genetic component that is not my expertise, only on the dating part that is.

The paper claims to present radiocarbon dates from 12 individuals (and not 13 as stated in the radiocarbon paragraph of the main text?). While direct dating is used as a major prior to the hypothesis the paper seeks to answer, there is minimal information regarding the C-14 ages. The S1 and S2 tables simply include Lab code, C-14 age and std error. No information on the chemical parameters of the extraction, no total collagen yield, C yield post combustion, to allow us to assess the dates independently. As the text/ SI stands the information provided is inadequate and I would urge the authors to rectify this.

It seems to me that some of conclusions lie on the claim of two individuals being Almoravid arrivals (~1115 CE). In order to achieve further chronological precision, the authors combine radiocarbon dating with dietary isotopes to correct for marine reservoir effects. This is a relatively standard method, but admittedly imperfect. Since marine organisms have older apparent radiocarbon ages than terrestrial ones (because the ocean's carbon reservoir exchanges more slowly with the atmosphere), if a person's diet included significant marine protein, their tissues will reflect that older carbon — making their radiocarbon age artificially old.

To correct for this, the  $\delta^{13}\text{C}$  and  $\delta^{15}\text{N}$  values in bone collagen are used to estimate how much dietary protein came from marine vs. terrestrial sources (using a Bayesian mixing models, and then this correction is applied using local  $\Delta\text{R}$  values (the regional marine reservoir offset).

As the authors rightfully claim, it may be necessary for coastal contexts like Ibiza, to apply such as correction because marine food was part of the diet. The stable isotope approach provides a quantitative proxy for marine intake, and Bayesian models (as used in the paper) incorporate uncertainty. This is especially important, since two individuals had relatively high  $\delta^{15}\text{N}$  values (consistent with marine intake).

One of my concerns is that the sample size claiming marine intake is small (2). I don't quite understand why only these 2 were corrected for marine input, and not the others who may have more limited fish protein in their diet? The authors say these 2 have the more recent radiocarbon dates, but this is not quite true either. I may be missing something, but it'd be good to get an answer. Would, for example, if all individuals are calibrated with a marine reservoir correction, have different age ranges? Could the authors provide with a calibration of all individuals, with and without marine diet correction?

Ultimately, Bayesian estimates should be seen as probabilistic estimates, not precise calendar anchors, and while the posterior calculation (80-95% postdating 1115CE) is high, the means of reaching to this estimate are somehow unclear to me. It may be the words in the main text, but the SI does not provide more answers.

Finally, a minor point, but both in the main text and SI, there is a sentence that "...from  $\delta^{13}\text{C}$  and  $\delta^{15}\text{N}$  values obtained from the same dentine collagen used for aDNA and radiocarbon analyses". The dentine collagen is collagen, ie protein, it is not used for aDNA analysis. The aDNA derives from the tooth dentine.

Overall, a table dedicated to the C-14 data only is rather important. There are specific guidelines in the literature what such a table should include. I would like to understand a bit more why and how the marine reservoir was applied, and to whom, and also see detailed tables with the calibration of these dates, with/without a marine correction, as well as all relevant chemical apartments of extraction and combustion.

The paper is indeed a nice piece of interdisciplinary work, but some parts have been given unequal weight.

Version 1:

Reviewer comments:

Reviewer #1

(Remarks to the Author)

All requested revisions have now been completed, and I am satisfied that the manuscript has been improved accordingly. The authors have addressed the points raised in my review in a clear and thorough manner, strengthening the structure, clarity, and interpretive framework of the paper. In its current form, the manuscript presents a robust and well-supported analysis, and I am pleased to confirm that my comments have been fully considered.

Reviewer #2

(Remarks to the Author)

The authors have considered the suggestions proposed by all the reviewers and answered all our questions in detail. I think this version of the manuscript is appropriate for publication in Nature Communications.

Congratulations to the authors on their excellent work.

Reviewer #3

(Remarks to the Author)

I thank the authors for addressing my extensive comments comprehensively - particularly in toning down some of the language, improved inclusion of the specific archaeological context of the burials, and limited reproducibility due to a lack of code/scripts and description of parameters in some place.

I particularly commend the author(s) particularly those responsible for the metagenomics section for the well structured and described list of scripts (the population genomics ones are still a little thin, but still more than many publications these days).

I am satisfied my comments have been sufficiently addressed for publication.

I list here a few minor remaining points, however they do not prevent publication:

**METAGENOMIC ANALYSIS**

1. I would remove reference to the 'orange complex'. This is a rather out of date concept now, and does not add anything - you can just state it is a pathobiont.
2. Do not use the term 'isolate' unless you are talking specifically about culturing (something impossible with dead organisms)

**DISCUSSION**

1. 'we identified molecular evidence of infection \*\*to\*\* primary pathogens' -> this reads oddly to me, I think you mean 'by' not 'to'?

**LIBRARY PREPARATION AND SEQUENCING**

1. 'with the following specifics settings' -> typo on 'specifics', should be 'specific'

**CAPTURE ENRICHMENT FOR M. LEPRAE IN S.313**

1. 'Final sequencing was performed on a NovaSeq S6000' -> typo still remains here even though correct in first sentence of next section, should be Illumina NovaSeq 6000

**CODE AVAILABILITY**

1. Don't forget to include the Zenodo archive DOI as well as the GitHub URL (mentioned in the rebuttal letter but not in the text it seems)

Reviewer #4

(Remarks to the Author)

I would like to thank the authors for addressing the issues regarding the chronology and new radiocarbon data presented in the paper. I am happy with the revisions and enjoyed reading the paper and the author interpretation of rather complex data.

**Open Access** This Peer Review File is licensed under a Creative Commons Attribution 4.0 International License, which permits use, sharing, adaptation, distribution and reproduction in any medium or format, as long as you give appropriate credit to the original author(s) and the source, provide a link to the Creative Commons license, and indicate if changes were made.

The images or other third party material in this Peer Review File are included in the article's Creative Commons license, unless indicated otherwise in a credit line to the material. If material is not included in the article's Creative Commons license and your intended use is not permitted by statutory regulation or exceeds the permitted use, you will need to obtain permission directly from the copyright holder. To view a copy of this license, visit <https://creativecommons.org/licenses/by/4.0/>

**This is the point-by-point answer to the reviews, corresponding to the manuscript Rodríguez-Varela et al.**

We thank the reviewers for their thoughtful and constructive comments. We have carefully revised the manuscript to address all points raised, which have significantly improved the clarity and impact of the study. Below, we respond to each comment in detail.

**REVIEWER COMMENTS**

**Reviewer #1** (Remarks to the Author)

- Key results: Please summarise what you consider to be the outstanding features of the work.

This paper presents unpublished results on the origins of the populations that arrived in the Iberian Peninsula during the medieval Muslim period, beginning with the initial 8th-century Islamic invasion and continuing through later waves of settlement. It further traces these movements to the subsequent arrival of related populations in the island of Ibiza. The contribution is highly significant, particularly for its use of ancient DNA analysis to determine geographic provenance. The refinements applied to these genetic analyses—such as accounting for the number of generations elapsed since migration or detecting the dietary influence of marine resources—further enhance the precision and interpretive value of the results.

The historical implications are equally substantial. While many of the findings correspond with information preserved in medieval written sources, such texts must always be approached critically, as they often reflect political, religious, or cultural biases and do not necessarily capture the full demographic reality. Moreover, the Balearic Islands do not possess the written accounts that other places have, in part due to the late conquest of the islands (902/903). Therefore, this paper provides concrete evidence on the origins of Muslim populations, that has far-reaching implications: it deepens our understanding of their racial and ethnic composition, sheds light on patterns of migration and assimilation, and informs interpretations of their cultural, social, and ecological contexts within al-Andalus and the western Mediterranean.

Finally, the radiocarbon dating of the individuals analysed provide the necessary information to set the results in its correct historical frame. For instance: the presence of muladies (non-african muslim population) along with black population from Subsaharian Africa reveal which was the pattern of conversion and acculturation of Iberia after 711. Again, the fact that the research is centered on the Balearic Islands fill that gap that the islands have for this time period.

- Validity: Does the manuscript have flaws which should prohibit its publication? If so, please provide details. NO

- Originality and significance: If the conclusions are not original, please provide relevant references. On a more subjective note, do you feel that the results presented are of immediate interest to many people in your own discipline, and/or to people from several disciplines?

The conclusions are original in what I have mentioned earlier: they provide empiric prove of part of the information that we knew from written sources and archaeological work. However, this new data is much more refined and provides microinshigt in information we could infere from the rest of the data but not prove empirically. For instance: the subsaharian population with a precise origin, the number of generations that have passed since the arrival to the Iberian Peninsula and Ibiza. Moreover, north-African and middle-eastern populations have been clearly recorded in the texts but this genetic

analysis refines what we know about this influx.

The other thing is that the genetic analysis can trace the pathologies and epidemics. The radiocarbon results provide exact chronology to the bodies which can then be studied in a precise historical context.

- Key Data:

- Radiocarbon dating of all samples analysed which provides the necessary accurate information for historical implications

- Control analysis: to refine results. Super important. Impossible to do it without high resolution research and it offers the corrections needed to have a very full and accurate picture of the situation

- High resolution data of origin: Subsaharian Africa is a whole continent. Research conducted identified different origins both from the west and the east.

- It's focused on an island that has less information about its Muslim past than others.

- Conclusions: Do you find that the conclusions and data interpretation are robust, valid and reliable?

Yes, they are. One of the most enduring and vivid debates concerns the ethnic background of the populations that arrived in the Iberian Peninsula in 711 and in the subsequent centuries. Some researchers argue that these contingents consisted solely of soldiers, with no significant civilian migration, and that these soldiers eventually intermixed with the local population. Others point to the transfer of agricultural knowledge, seeds, technologies, and other cultural elements as evidence of broader population movements. This paper's findings support the latter view, indicating a gradual demographic shift from a predominantly European population to one with substantial North African ancestry. Furthermore, the ability to trace lineages across generations provides additional confirmation of a steady process of migration and intermixing, as well as the presence of European converts from the very beginning of the conquest in the 8th century.

In addition, the Balearic Islands—particularly Ibiza—are often underrepresented in the historical record. Written references are relatively scarce, and the islands are frequently treated as a single unit, with their history assumed to follow that of Mallorca. This tendency makes it difficult to isolate the specific historical trajectories of each island, except in certain well-documented events. In this context, the paper's provision of precise, individualized data for Ibiza represents a significant and original contribution, offering new insights into the island's unique role within the broader medieval Muslim Mediterranean world.

- Suggested improvements: Please list additional experiments or data that could help strengthening the work in a revision.

We would like to thank Reviewer 1 for their valuable comments and suggestions.

1. The paper would benefit from the inclusion of a photograph of the maqbara (excavation). Such an image would provide a visual context for readers unfamiliar with the site, enhancing their understanding of the archaeological setting. Ideally, this figure should be included either in the introduction or in the section where the burial context is first described.

A plan of the site has now been added as Figure 1b in the Introduction, depicting the burials of the individuals analysed in this study.

2. The historical background is informative and provides a strong foundation for the study. However,

the introduction section as a whole is somewhat disorganized. There are frequent mentions of results scattered throughout the introduction, sometimes before the historical or methodological framework has been established. This disrupts the logical flow and may confuse readers who expect the introduction to focus on background, research context, and objectives.

I recommend restructuring the introduction and not include the results of the research that are already included in the next section and in the abstract.

*Finally, maybe a couple of lines about the research gap that the study is trying to solve and a very brief list/overview of the methodology, without revealing specific results.*

We thank Reviewer 1 for the constructive feedback. We would like to clarify that the current version of the Introduction does not include any of our own research results; rather, it presents historical, archaeological, and previously published contextual information necessary to set the scene. In particular, we summarize and cite earlier archaeological papers published in Spanish about the site under study and genetic literature to establish the broader demographic and cultural background of medieval Ibiza, and to provide a rationale for our research questions.

To improve clarity, we have revised the text to ensure a clearer separation between background/context and our study aims. We have also slightly restructured the Introduction to more clearly highlight the research gap and to summarize the methodology without implying results. These changes have notably improved the logical flow and readability of the introduction, in line with the reviewer's suggestions.

- Clarity and context: Is the abstract clear, accessible? Are abstract, introduction and conclusions appropriate

Abstract

The abstract is clear and accurately states the hypothesis, methods, and results. It effectively summarizes the purpose and findings of the research.

Historical Introduction

The historical background is informative and provides a strong foundation for the study.

The introduction does not need to contain detailed results, as these are already well covered in the abstract and in the dedicated results section (comments above)

Main Body

The remainder of the paper is excellent and makes a significant scholarly contribution. The results are especially valuable for the integration of written historical sources and archaeological evidence. *This multidisciplinary approach allows for a more nuanced understanding of the kind of population that arrived in Iberia and the Easter Islands of al-Andalus (Balearic) during the Middle Ages, and should be emphasized as one of the study's key strengths*

We are especially pleased that the reviewer highlighted the integration of historical and archaeological evidence, as one of our main aims was precisely to bring these disciplines together with genetic data. We have made sure we emphasize this multidisciplinary perspective more clearly in the revised version, underlining how the combination of disciplines allows us to refine our understanding of the populations that arrived in Iberia and the Balearic Islands during the Middle Ages in Europe.

- Please indicate any particular part of the manuscript, data, or analyses that you feel is outside the scope of your expertise, or that you were unable to assess fully.

I am not familiar with the genetic analysis conducted. I am familiar with the time period, questions asked, research about medieval population, debate about origins and influx of population from Africa during the Middle Ages, excavations with similar questions and implications of these types of analysis.

#### Overall Recommendation

This is a valuable and original study. With a more clearly structured introduction and the addition of relevant visual material, the paper's clarity and impact will be further strengthened. The results presented here will be warmly received by the scholarly community working on al-Andalus, which has long debated—often for decades—key issues such as the processes of Islamisation in Iberia, the extent and nature of population arrivals (whether limited to military elites or involving broader civilian migration), the conversion of the local Iberian–Roman population (*muladíes*), the degree of racial and ethnic intermixing, and the prevalence of specific pathologies. These debates have traditionally relied heavily on medieval textual sources and, to a lesser extent, archaeological excavations. By introducing robust empirical evidence through genetic analysis, this paper provides a new and independent line of inquiry that not only illuminates these long-standing questions but also has the potential to reorient current theoretical frameworks. The study stands as an important contribution that bridges historical narratives and scientific data, offering a more nuanced and evidence-based understanding of population dynamics in medieval al-Andalus, particularly in the margins of the empire (the islands).

Additional questions: when referring to individuals that were used from Northern Iberia for comparison (Burgos, Pais Vasco, etc.) there should be a specific reference to where they are coming from. If there is one, I couldn't find it.

Thank you for this comment. In addition to the cited articles, we have now included specific details regarding the provenance of these individuals.

Added to the manuscript (*page 8, line 256-260*): “To further explore the genetic ancestry of these individuals, we performed genotype imputation on the ancient Ibiza samples alongside two controls: a pre-European contact individual from Tenerife (Canary Islands) (gun011; 42) representing a North African source, and a pre-Islamic early medieval individual from Las Gobas (Condado de Treviño, Burgos) in northern Iberia (Ido039, individual 26; 23) representing a local Iberian source.”

#### Reviewer #2 (Remarks to the Author)

In the manuscript "Genetic Diversity and Health in Medieval Islamic Ibiza: Unveiling North African, European, and Sub-Saharan Ancestries", Rodríguez-Varela et al. generate genome-wide data from 13 individuals from an Islamic cemetery in Ibiza dated to 950–1150 CE. The authors infer the geographical ascription of the individuals, providing information on population movements in the

Balearic Islands during the Islamic period, which included individuals of North African, European and sub-Saharan African origin. They also explored the metagenomic profiles of the individuals, detecting the presence of several pathogens, including *Mycobacterium leprae*.

I think that the authors have performed a comprehensive exploration of the data and most conclusions are well supported by data analyses. The research question around which this manuscript is centered is interesting and the results are compelling. The manuscript is well-written and it is evident that it has benefited from previous reviews. I have to acknowledge the effort made by the authors to incorporate evidence from different fields in order to better understand the Islamic period in Ibiza. Potentially, this manuscript will be of interest to diverse fields, including paleogenomics, archaeology, evolutionary biology and population genetics, and it is a good fit for Nature Communications. However, I think that there is one aspect that should be revised before publication.

The RFmix analysis, including the decay of local ancestry, is really interesting, but I'm not sure the approach is appropriate. The ancient individuals, including the Guanche from the Canary Islands, are modeled as the admixture of European, North African and Middle Eastern components. However, because of prehistoric and historical migrations in the region, the Mozabites are the result of admixture from these three components, so what is considered North African regarding local ancestry? Also, given the East to West cline in the ancestral Maghrebi component in North Africa, are Mozabites a reasonable proxy for the individuals arriving in the Iberian Peninsula during the Islamic period? If not, is it possible that the identification of European or North African segments might be skewed? For example, the Guanche individual predates the Islamic expansion in North Africa and can be considered an isolate in the region; however, the local ancestry results show some long segments of European ancestry. How should we interpret those segments? Can they be considered the result of recent European admixture? Or are these segments an artifact created by differences in the admixture patterns between Guanches and Mozabites? Again, although the analysis has the potential to provide interesting information on the Islamic population in Ibiza, I think the authors should consider all these limitations and discuss them in the paper.

We thank the reviewer for raising these important points. Our RFMix results are consistent with the ancestry patterns inferred from ADMIXTURE and PCA using three independent datasets (Human Origins, 1k+HGDP, and the extended Africa2 panel). This convergence suggests that the main ancestry signals are robust, even if some local ancestry segments are occasionally misassigned.

We agree that using Mozabites and Bedouins as our primary North African reference populations is a limitation, especially given the complex prehistoric and historical admixture in North Africa. Although additional modern North African genomes exist, the available data lack sufficient sample size, geographic representation, or SNP overlap with our ancient dataset to be incorporated into the RFMix framework without greatly reducing marker density and reference diversity. We therefore acknowledge that Mozabites carry both European and Near Eastern related ancestries, and that some overlap between these components is expected across the Mediterranean. This inherently constrains the resolution at which closely related ancestries can be separated.

The reviewer's question regarding the Guanche individual gun011 is particularly insightful. In the ADMIXTURE analyses using Human Origins and 1K Genomes + HGDP, gun011 shows a small European component, which agrees with the RFMix inference (more sensitive). We interpret these not as evidence of recent European admixture, but rather as reflecting either (i) ancient European-related ancestry already present in pre-European Canary Islanders (Serrano et al., 2023; Rodríguez-Varela et al., 2017), or (ii) reference bias due to shared European ancestry in Mozabites and other North African populations. Another plausible explanation, consistent with the first hypothesis, is that these small segments reflect ancient European gene flow into North Africa during earlier periods (e.g., the Roman era), which may have persisted in the ancestors of gun011 before their isolation in the Canary Islands. Thus, both biological and methodological factors could contribute to the apparent European signal.

We now emphasize these points in the revised manuscript and clarify that our RFMix results should be interpreted in comparative rather than absolute terms. Importantly, potential reference-related misassignments do not critically affect our temporal inferences: tract-length patterns remain informative for admixture dating even when some local ancestry calls are imperfect. Most methods based on tract-length distributions can be biased by small erroneous segments that artificially shorten tracts; instead, our approach relies on the decay of ancestry covariance, which is less sensitive to such local inaccuracies.

Added to the manuscript (*page 13 and 14, line 402-423*):

“The local ancestry analyses using RFMix yielded results broadly consistent with the ADMIXTURE and PCA patterns obtained from three independent datasets (Human Origins, 1k+HGDP, and the extended Africa2 panel), supporting the robustness of our main ancestry inferences. However, we acknowledge that using Mozabites and Bedouins as the sole North African reference populations and proxies for medieval Berber and Arab populations entering into Iberia is a limitation. Mozabites genomes include both European and Near Eastern components. Consequently, some ancestry segments may have been misassigned as “European”. In the ADMIXTURE analyses of this study using Human Origins and the 1K-HGDP datasets, the pre-European individual from Tenerife (Guanche, gun011) also shows small percentages of European ancestry (Supplementary Fig. 3 and Fig. 4), in agreement with the RFMix results (Fig.5 and 6). We interpret these not as evidence of recent European admixture, but rather as reflecting either (i) ancient European-related ancestry already present in pre-European Canary Islanders42,45, or (ii) reference bias due to shared European ancestry in Mozabites. Another plausible explanation, consistent with the first hypothesis, is that these small segments reflect ancient European gene flow into North Africa during earlier periods (e.g., the Roman era), which may have persisted in the ancestors of gun011 before their isolation in the Canary Islands. Thus, both biological and methodological factors could contribute to the apparent European signal. We therefore stress that our RFMix results should be read in comparative rather than absolute terms. Importantly, these potential misassignments do not critically affect our temporal inferences: tract-length patterns remain informative for admixture dating, as our approach, based on ancestry covariance decay, is

less sensitive to local misclassification than methods relying directly on tract-length distributions.”

Minor comments:

- The authors indicate in the introduction that the “Maqbara of Madina Yabisa” cemetery contains 125 individual burials, but in the results section, they refer to 41 excavated individuals. I assume this is because part of the cemetery has been not excavated yet, but maybe it would be better to clarify this issue in the introduction.

We thank the reviewer for pointing this out. Indeed, the cemetery is estimated to contain around 125 burials, but we obtained permission to study 41 individuals in this study. We have clarified this distinction in the introduction and results to avoid confusion in page 6 line 171.

- Figure 1: Probably it is because of space constraints, but the text in Figure 1B is too small. Would it be possible to increase the font size a little?

To improve readability, we have increased the font size of Figure 1b, now called Figure 2.

- Figure 3: It is impossible to see the ADMIXTURE proportions of the last group of ancient Iberians in the plot before zooming in, as it appears as a back rectangle. I would recommend removing the dividing black lines between individuals or removing that section from the general ADMIXTURE plot and only including it in the zoom-in version.

We thank the reviewer for this helpful comment. We have modified Figure 3 by removing the dividing black bars between individuals, which improves the visibility of the ADMIXTURE proportions for the last group of ancient Iberians.

- Second paragraph, page 1 of the Supplementary text: Add a parenthesis before "see that paper...".

We have added the parenthesis before “see that paper...” as requested.

- Figure S3, Figure S4 and Figure S11: Same as Figure 3. For the K=2 to K=9 plots, it is very difficult to see the results for the ancient individuals identified with codes 1 to 15 due to the lines that divide each column. I recommend removing the lines between samples to improve the readability of the figures.

We have removed the dividing black bars between individuals, which improves the visibility of the ADMIXTURE proportions for the last group of ancient Iberians.

Reviewer #3 (Remarks to the Author)

N.B: Best way to view this review is to copy into a text file and use a Markdown (pre)viewer (e.g. your IDE)

## Summary

In this palaeogenomics study, Rodríguez-Varela and colleagues performed standard human population genomics and microbial genomics analyses on ancient human remains from a cemetery on Ibiza dated to the al-Andalus period of Iberian history.

For the human DNA, they perform a suite of broad overview analyses to identify source and timing of the ancestral origins of the 13 different individuals yielding sufficient DNA,

For the microbial DNA the authors screened the off-target reads for traces of potential pathogens - both bacteria and viruses, that yielded sufficient genomic coverage from a single individual to place this ancient genome on the phylogenomic tree of *M. leprae*.

Please note my technical expertise is in the microbial genomics, thus I focus my review primarily on this part of the manuscript.

Overall the period the study is generally rather understudied from a palaeogenomics point of view, making the results relevant, particularly given this specific era of Iberian history is an interesting one due to the existing rich historical and archaeological contextual information in addition to the large cultural changes and movements happening during this time frame.

Methodologically, for both the population genomics (to my knowledge) and microbial genomics there is nothing particularly new, but all methods are sound and following standard tools and analyses.

My main issues with the paper (details below) are:

- 1. Limited integration of the archaeological results (and overextended language in the discussion about) despite being listed as a primary aim*
- 2. Overly strong language in the discussion about the impact of findings of the microbial genomics results, and limited discussion of comparison to other genomes from the period (although the methods are sound!)*
- 3. Very insufficient details on methods for reproducibility of the analyses*

Thank you for these valuable comments. We have addressed all three points as follows:

- 1. Archaeological integration:** We have expanded the manuscript to better integrate the archaeological results throughout the text and moderated the language in the Discussion to align with the evidence.
- 2. Microbial genomics:** The Discussion has been revised to use more cautious language regarding the impact of the microbial genomics results.
- 3. Methods for reproducibility:** We have included detailed descriptions of the methods, software and versions, specifying where and how analyses were performed, to ensure full reproducibility. We also create a GitHub repository (<https://github.com/quenllavarela/medieval-iberia-genomics>) with our scripts and archive this on Zenodo). v1.0.0 [quenllavarela/medieval-iberia-genomics: scripts used in: Genetic Diversity and Health in Medieval Islamic Ibiza: Unveiling North African, European, and Sub-Saharan Ancestries. DOI: 10.5281/zenodo.17661881](https://zenodo.org/record/17661881)

## General

For the general comments of my review, I will use the criteria specified by the journal

### What are the noteworthy results?

The main results of the paper are consolidating historical and archaeological timelines of the settlement and population dynamics within Ibiza during the Islamic rule, refining these models with genetic data.

Contribution of a new ancient *M. leprae* genome from another location in Spain in this time period.

### Will the work be of significance to the field and related fields? How does it compare to the established literature? If the work is not original, please provide relevant references.

The methods (smartpca, ADMIXTURE, f3-statistics, KinSHIP; aMeta, raxML, MEGAHIT) and questions are rather routine in the area matching similar recently published papers, but generally the results are contributing to an understudied period.

### Does the work support the conclusions and claims, or is additional evidence needed?

Generally the claims made in discussion match the results: in most cases the results are just confirming the existing archaeological (e.g. stable isotopes, radiocarbon) and historical evidence but via a new perspective and providing greater nuance.

### Are there any flaws in the data analysis, interpretation and conclusions? Do these prohibit publication or require revision?

I do not see anything preventing publication, *however some of the language in the discussion and interpretations are over blown and should be revised, particularly in regards to the ancient microbial results.*

In particular, I take issue with statements such as:

> (INTRO) integrate genetic, historical, and archaeological evidence to reconstruct population history

> (DISCUSSION) This study presents a comprehensive archaeological, genetic and metagenomic analysis of individuals from an Islamic necropolis in Ibiza

Despite this, there is actually scant archaeological context described about the necropolis nor the individuals.

The description of the osteological results is extremely minimal even though it provides a huge source of information for that is important on statements on health and disease interpretation - the only mention of this is a single sentence in the discussion (referring to results that are not described elsewhere).

Another archaeological result relevant to the human population genomics analyses would be the characteristics of the graves of the individuals themselves - are there any form of grave goods present? What other links could be made about this and the results from the genomic analyses?

But this is not explored.

Even *if* there is very little material left in the graves (something not really discussed), thus making

further interpretation difficult, this should at least be stated.

I would argue much of the discussion is combining the broader historical context, which is fine, **but it is not integrating the archaeological context of the specific site or individuals.**

We appreciate the reviewer's observation regarding the limited archaeological contextualisation. We adjusted the tone in the Introduction and Discussion to avoid overstatement. We have added further information about the site's archaeology wherever possible, including a plan of the site (Fig. 1b), a description of the burial types, body orientations, and the absence of grave goods, all of which are consistent with Islamic funerary traditions. We toned down the introduction statement pointed out by reviewer #3 with "combine genomic data with dated archaeological context and relevant historical background to investigate population history" and the discussion statement with "This study combines genomic and metagenomic data from dated burials at an Islamic necropolis in Ibiza with documented archaeological context, offering new insights into the demographic landscape of Al-Andalus and providing finer resolution in genetic ancestry. In our pathogen screen we detect authenticated ancient DNA signals consistent with hepatitis B virus and *Mycobacterium leprae*, and low-level parvovirus B19 signals, the latter most likely reflecting persistence after infection.". In the Introduction (*page 3 line 107-115*):

"The archaeological site located on Bartomeu Vicent Ramon street corresponds to a sector of the "*Maqbara of Madina Yabisa*", a main urban Muslim cemetery, where 125 individual burials dated between 925 and 1150 CE were excavated. The graves, consisting of simple earth pits (darīh type), align with Islamic funerary law prescribing unadorned burials, with the exception of one burial (UE 153)8, which yielded personal ornaments (two silver rings) (Fig. 1B)9. Most individuals were placed on their right side, facing southeast toward Mecca, and the cemetery shows evidence of prolonged use, with occasional overlapping burials (Fig. 1B). Osteological assessment indicates a demographically mixed population, with both sexes and all age groups represented, good preservation, and limited evidence of trauma or skeletal pathologies9,10."

In *page 6 line 170-187* of the results:

"Permission was granted to study 41 individuals, but due to bad preservation and tooth availability, only 30 were sampled (Supplementary Table 1) from the 125 burials excavated at the Maqbara of Madina Yabisa (Ibiza, Spain). From these, sufficient autosomal DNA was successfully recovered from 13 individuals, enabling population genomic and metagenomic analyses (Fig. 1b). The majority of the graves are oriented (W-E) along an axis between 250° and 284° W, showing no fixed spatial order within the necropolis, which likely reflects its long-term and somewhat irregular use9. However, a group of ten graves stands out due to its distinct arrangement: they are aligned parallel to each other, evenly spaced, and oriented NW–SE, with the individuals facing toward the southeast (115°–137°). Among the thirteen individuals that yielded ancient DNA, two belong to this southeast-oriented group (s. 107 and s.103) (Fig. 1B). We present the estimated age at death and stature based on osteological examination of these 13 skeletons (see Methods). Our results indicate that all individuals

were adults at the time of death, with estimated statures ranging from 145 cm to 176 cm (Supplementary Table 1). No grave goods, trauma or diagnostic skeletal pathology were observed in the sequenced individuals. However, the absence of certain elements, such as the facial bones in individual S.313, makes it difficult to rule out the presence diseases such as leprosy. For the genetic analyses, only teeth were sampled.”

The other set of too strong language is the impact of the ancient microbial results:

> reveal a diverse range of pathogenic and commensal microbial DNA, offering new insights into the disease landscape of medieval Ibiza

Only 4 taxa are mentioned, 2 are commensal viruses, and *S. pneumoniae* was not discussed further (for reasons that are not clear) - the only major result is a single individual carrying an leprosy genome of a standard genotype of the time period and region.

While I appreciate that the authors were careful with authenticating the report hits, this is definitely not a 'landscape'.

Please be careful of using (what I know is the currently 'in vogue') terminology and keywords within the field that are overly ambitious.

We revised the wording to avoid overstatement. In the manuscript, we replaced the “diverse range... disease landscape” sentence with a neutral enumeration of authenticated pathogen detections: hepatitis B virus, human parvovirus B19, *Mycobacterium leprae* and *Streptococcus pneumoniae*. We also added brief constraints reflecting sample size and within-tooth sampling. We note one authenticated oral microbe, *Parvimonas micra*, and clarify that our sampling derives from within teeth (dentine/pulp), which targets blood-borne agents rather than the oral surface microbiome present in calculus or plaque. Our filter for likely environmental contaminants might be stringent for oral microbiome detection, which is not the focus of our study. *Streptococcus pneumoniae* passed authentication. Given its potential asymptomatic carriage, we report the detection but focus downstream analyses on primary pathogens, since we are especially interested in the diseases of the individuals, which explains why we did not mention it again. We realise now that we should have mentioned it again in the discussion based on reviewer #3’s comments. Here is the new paragraph in the discussion page 23 line 731-737:

“Alongside genetic ancestry, our metagenomic analyses identify authenticated pathogen signals for hepatitis B virus, human parvovirus B19, *Mycobacterium leprae* and *Streptococcus pneumoniae*, which can be carried asymptotically. The probability of pathogen detection is limited by the small sample size ( $n = 13$ ) and identification of oral microbiome taxa is further constrained by within-tooth sampling, which favours blood-borne pathogens. We authenticate one oral microbe, *Parvimonas micra*, which is part of the orange complex, associated with periodontitis71”

> Altogether, these results suggest that medieval Ibiza was not only genetically diverse, but also embedded in complex networks of disease transmission

I don't think you can state the viruses genetically diverse if they are all from the same genotypes (for B19V), and only two genotypes (HBV).

Also I would be wary to state 'embedded in networks of transmission' given there is no actual analysis of this: there is no analyses on the exact temporal or spatial relationships of the new *M. leprae* genome to the existing ones from the nearby Barcelona genomes from the same period with linking to the specific ancestry or mobility information !

Thank you for pointing this out. We remove the “genetically diverse” phrasing and to avoid any ambiguity we remove the inference about transmission networks, we replaced that sentence in page 25 lines 811-812 with: “The pathogens detected in this study provide insights into health conditions at the necropolis and establish a baseline for future comparative work.”

> The presence of *M. leprae*, HBV and B19V in several individuals adds a crucial layer of understanding to the social and environmental context of medieval Ibiza, further illustrating the intricate relationship between migration, health, and genetics in the region.

> underscores how disease was closely intertwined with human mobility and interaction during this period

Again I find the language to be rather flowery.

Niether adding a single *M. leprae* genome that is from a genotype that is spatially and temporal typical, nor detecting very common commensals viruses (which many palaeogenomics studies are now finding everywhere, such as Muehleman 2018, Kocher 2021 etc.), are adding 'crucial' insights.

With insufficient sample size (for which I do not blame the authors) to do a proper network analysis to compare the 'mobility' of the pathogens in relationship to the ancestry of the individuals, I think this is not exactly illustrating 'intricate relationships' of health.

Indeed - given the lack of any other evidence towards the health status across all individuals osteologically (see above), particularly individual s.313 - it's hard to make any links between ancestry and health status in this study - particularly given individual s.313 is a typical European individual.

We agree the original wording was too strong. We removed phrases such as “crucial layer” and “embedded in networks of transmission” and limited our conclusions to what the data support. We now state that detections of *Mycobacterium leprae*, hepatitis B virus and human parvovirus B19 provide direct ancient DNA evidence for these pathogens on medieval Ibiza and are consistent with broader ancient DNA reports of HBV and B19V. B19V in adults is consistent with persistence and *Streptococcus pneumoniae* may reflect asymptomatic carriage rather than disease. We retain a limited health framing based on the evidence available in this dataset: authenticated detection of two primary pathogens HBV and *M. leprae* that are not expected as potential commensals or persistent pathogens after infection. Given the sample size and design, we do not infer ancestry–health relationships or pathogen transmission networks.

Overall, I strongly recommend the authors to include more archaeological and osteological context/results to either the (otherwise very nicely written!) introduction and/or results, and tone down the language about the links with health.

Thank you for this suggestion. As noted in our previous response, we have expanded the manuscript by adding further archaeological and osteological context in the Introduction (including the plan of the archaeological site in Fig. 1b) and in the Results section. To avoid redundancy, we do not repeat those sentences here, but they have been incorporated into the manuscript as described. We have also moderated the language regarding links with health to more accurately reflect the available evidence and avoid overinterpretation.

### Is the methodology sound? Does the work meet the expected standards in your field?

The methods used are standard in both ancient population-genomic and ancient microbial fields, and the authors report sufficient ancient DNA authentication metrics (damage patterns, coverage, contamination estimations, evenness of coverage)

The authors have demonstrated awareness of potential issues, for example investigating potential technical artefacts in the detection of *M. leprae* in another sample (e.g., index hopping as described in the supplementary text)

Thank you for this positive assessment.

### Is there enough detail provided in the methods for the work to be reproduced?

While the choices of methodology and analyses is sound, the work is **definitely not reproducible**. The methods description only describes the names tools and only sometimes the versions - but even these alone definitely does not make the results reproducible.

There are rarely references to options, settings, or (metagenomic) reference databases.

I strongly encourage the authors to upload all code, scripts, config files, and commands to e.g. a GitHub/GitLab/Bitbucket repository and archive this on Zenodo.

The only exception to this is if most of the tools were run with default settings, in which case this should be mentioned for each tool, but this would make me question the quality of the analyses.

Thank you for this suggestion, we will provide all in-house scripts used in our analyses on GitHub (<https://github.com/quenllavarela/medieval-iberia-genomics>) and archived in Zenodo. For the other tools, we clarify in the Methods section whenever settings different from the default were applied. Default settings were used only where explicitly indicated, and we have ensured that all relevant details regarding versions and parameters are fully documented to maintain reproducibility. We recognise that some metagenomic steps were finalised close to submission, and a few tool versions were not captured in the text. We have now added the missing version.

Note the reporting summary file is also quite poorly described, missing many of the tools - however I recognise this is a terribly implemented system by Nature for better reporting so I do not criticise the authors on this.

Thank you for this observation. We have now added the previously missing tools to the reporting summary file.

## Specific

### Text

- Page 3, para 4: 'Tools such as', although I know you developed aMeta, just cite the different tools that can be used rather than singling out your own tool given you say 'such as', your tool may not always be the best (as you demonstrate for viruses here!)

We have now added additional tools and their references. The paragraph in *page 3 and 4 lines 132-135* now reads: “In parallel, ancient metagenomics provide valuable insights into the health and disease environments of past societies. Pipelines such as HOPS 20, aMeta 21 or the combination of mapping followed by ngsLCA 22 and bamdam (<https://github.com/bdesanctis/bamdam>) enable the identification of microbial communities and ancient pathogens in archaeological human remains.”

- Page 4, para 2: T 'From the 41 individuals...' this is one example of where I was surprised by the lack of archaeological description of the graves/archaeological context - particularly as you say you integrate archaeological results

We have now added the previously mentioned paragraphs describing the graves and their archaeological context in both the Introduction and the Results, in order to better contextualize the individuals included in our analyses.

- Page 5, para 2: 'We present the estimated' - I found it a bit jarring the results started with the genetic results and then jumped back to the osteological results. I would move the osteology to the top of the results and expand this to include more archaeological descriptions - particularly as you refer to how the DNA was extracted/sequenced here (i.e., after you already started describing genetic results)

We have moved the osteological analysis and archaeological context section in results to the top and included more osteological and archaeological information in this section.

- Page 13, para 3: 'To explore...' How did you define which pathogens you screened for? A reader might think something like - well did you really only detect these 4 species, maybe you missed an obvious one that I could go back and re-analyse to find (which they should have the freedom to do so!)

We thank the reviewer for prompting clarification. We screened all taxa returned by aMeta and retained for reporting those with authentication scores  $\geq 8$ , as in Rodríguez-Varela et al. 2024, because scores of 7 in this dataset yielded a very large number of false positives. The threshold is dataset-dependent and  $\geq 7$  may be appropriate in more degraded material. We applied an exclusion strategy rather than a preset pathogen list to classify likely environmental taxa and oral commensals (now expanded in Methods). The Methods now detail the four steps (BacDive 'Isolation sources' lists, Microbe Atlas soil prevalence  $> 1\%$ , primary literature checks, and eHOMD queries). Please note that Supplementary Fig. 13 contains the authentication score  $\geq 8$  heatmap for direct consultation.

For accuracy, because their presence was attested by sufficiently covered samples, we re-evaluated site-attested primary pathogens with a k-mer/read ratio  $\geq 1$  across all individuals regardless of read count (now mentioned in Methods) increasing the potential detections for these pathogens. This is how we identified index hopping affecting one individual during the leprosy screening.

However, we acknowledge that pathogen detection is influenced by several factors such as database choice, method and threshold choices. Additional taxa may have been present but remained undetected under our analytical conditions, although our two layers of authentication (aMeta scores and the KrakenUniq *k*-mer/read ratio) have proven effective at recovering pathogens, including several novel or less frequently reported in ancient DNA data as shown in Rodríguez-Varela et al. 2024. We also re-evaluated site-attested pathogens across all individuals as a study-level cross-check. This has now been detailed in the introductory paragraph of the metagenomics results section and better introduced at each of the different steps of the method section. To ensure transparency and facilitate further exploration, all metagenomic data and analysis outputs are publicly available (<https://github.com/quenllavarela/medieval-iberia-genomics>), allowing other researchers to reanalyse the datasets and apply alternative screening criteria if desired.

- Page 13, para 3: 'defined as likely...' What reasoning did you use to define *B. lata*, *C. tetani* etc. as contaminations?

We agree the information was scarce and have expanded that section as noted above. As detailed now in the Methods, we classified likely environmental taxa using a four-step exclusion strategy. Applying this:

- *B. lata* was flagged as terrestrial by BacDive 'Isolation sources' and is reported mainly from non-human contexts. It did not meet our authentication criteria in this dataset.
- *Clostridium tetani* exceeded the Microbe Atlas soil prevalence threshold and is a well-known soil-associated anaerobe. We frequently see *C. tetani* in ancient tooth datasets from different sites. It is vaccine-preventable and boosters are recommended after soil-contaminated wounds.

In short, both taxa were treated as environmental under our criteria. We would be interested in deeper study of such signals, but in most cases, they are likely environmental. Evidence for active infection would require orthogonal analyses such as ancient proteomics.

- Page 13, para 3: 'detected in individuals' be more precise in your language - without further validation (trees, unique marker genes), you have not *\_detected\_ S. pneumoniae\_ and M. leprae\_* at this stage. You have just found reads that matched/aligned to the reference genome etc.; these could still be theoretically commensal relatives at this stage (particularly for *S. pneumoniae\_* given *Streptococcus* is such a broad genus).

Thank you for this point. We use “detected” only for taxa that pass a KrakenUniq unique k-mer pre-screen followed by our predefined authentication criteria as described in the method section, not for any read match. In this dataset that denotes taxa with aMeta score  $\geq 8$  supported by damage, coverage and identity metrics, plus visual inspection of the full authentication plots. Reflecting this, the Results now state that among the authenticated taxa only *Streptococcus pneumoniae* and *Mycobacterium leprae* met our criteria for pathogen detection in individuals s.155 and s.313, were confirmed by plot inspection, and were classified as a potential pathogen and a primary pathogen, respectively. Note that *Streptococcus pneumoniae* received a score of 9, which was the highest score in this dataset. For viruses, species-level assignment is further supported by BLAST validation and for B19V and *Mycobacterium leprae* by phylogenetic placement. Although *S. pneumoniae* is not the main focus of this study as mentioned earlier, we have added an extra BLASTn check of the MALT-extracted Streptococcus reads and have improved the blast summarisation script for the tables. The results are consistent with *S. pneumoniae* (Supplementary Table 6). We emphasise that low coverage can still meet our detection definition even though it limits downstream analyses such as phylogenetic reconstruction and unique marker gene analyses. In such cases, and where a taxon is not the study focus, species validation can rely on BLASTn validation of assigned reads as in this study or competitive mapping across all species within the genus as shown in Rodríguez-Varela et al. 2024. In our experience competitive mapping and BLASTn gave concordant outcomes, and we use BLASTn here because it avoids building full genus databases, which is time and storage consuming.

- Para 13, para 3: 'Only Streptococcus...' Why do you mention *S. pneumoniae* here and then never again? Why did you exclude this from any further analysis? I find this a glaring unexplained omission

Thank you for flagging this. *Streptococcus pneumoniae* was authenticated and reported, but it was not a primary focus. Given its frequent asymptomatic carriage, we treated it as an authenticated detection rather than a target for downstream phylogenetics. We now make this explicit in the Results and include it in the BLAST validation table (Supplementary Table 6), and keep it in the Discussion.

- Page 13, para 3: 'Only Streptococcus...' It is also important to describe (something that was not done) what is in your database. If you look at NCBI RefSeq, *S. pneumoniae* is massively overrepresented with having more than 8000 genomes, whereas there is only *M. leprae* genome - is this why you ignore *S. pneumoniae* - was it a false positive?

Thank you for raising this. As part of the reproducibility updates we now specify the default aMeta databases in Methods. For pathogen screening, aMeta uses a KrakenUniq classifier database built from an NCBI nt-based microbial library with selected complete eukaryotic genomes, including human. For alignment and authentication, a MALT index is built dynamically from an nt-derived Bowtie2 FASTA and restricted to taxa returned by KrakenUniq. The database composition is described in more detail in the aMeta paper. We agree that public repositories are unevenly represented across taxa, but nt includes extensive

Streptococcus representation and our authentication does not rely on database counts alone. aMeta integrates edit distance, coverage evenness, post-mortem damage, read length, PMD and ANI, with a KrakenUniq unique k-mer prescreen. We have also added a BLASTn validation of the MALT-extracted Streptococcus reads and the results are consistent with *Streptococcus pneumoniae* (Supplementary Table 6). The detection is therefore not a false positive. We treat *S. pneumoniae* as an authenticated detection rather than a target for phylogenetics because it is frequently carried asymptotically and was not the focus of this study.

- Page 14, para 1: 'Screening below...' Why did you start looking below your thresholds? It seems a bit contradictory to set one then 'ignore' it - it comes across as fishing for results. What is the problem with the original thresholds? EDIT: Later on your justify this that aMeta isn't good for Viruses - but you should briefly state this here (e.g. 'We then looked below this threshold due to issues with the aMeta score for detecting viruses (see discussion)') as I immediately became suspicious on the first reading of this.

We appreciate the concern. We now routinely use this secondary layer of authentication because there is no one-size-fits-all detection method across organisms. Small viral genomes, atypical coverage profiles and faster mutation rates can depress aMeta scores despite real signal. Sub-threshold viral hits were treated as candidates for manual follow-up and were reported only after independent validation. In this study, HBV and B19V were validated by BLAST, and B19V was additionally placed phylogenetically. We now state this rationale at first mention in Results as you suggested. Here is the new result paragraph *page 17 and 18 lines 498-537*:

“Since our aim was to recover pathogens spanning a wide range of genomic characteristics (e.g., genome size, partial single-strandedness, divergence from reference due to mutation rate, pathogen load), we applied two layers of authentication: a first one using aMeta, and a second one using an evenness of coverage proxy based on the KrakenUniq results generated within aMeta, similarly to 23. In addition, whenever a pathogen is attested at the site, we reevaluate its presence at low coverage across all individuals by analysing all the reads assigned to it via MALT 70. For the first layer of authentication, we focused on microbial taxa authenticated through a combination of metrics summarized into authentication scores by aMeta 21 (see Methods). Among taxa with authentication scores  $\geq 8$  (Supplementary Fig. 13), several were identified as likely environmental contaminants (e.g. *Burkholderia lata*, *Clostridium tetani*, and *Ralstonia solanacearum*), and one as part of the oral microbiome and the orange complex, which is associated with periodontitis 71 *Parvimonas micra* in individual s.133 (Supplementary Fig. 14-15). Among the authenticated taxa, only *Streptococcus pneumoniae* and *Mycobacterium leprae* met our criteria for pathogen detection, in individuals s.155 and s.313, and were confirmed by visual inspection of the full authentication plots and classified as a potential pathogen and a primary pathogen, respectively (Supplementary Fig. 16-17). Notably, *S. pneumoniae*, a pathogen known to cause pneumonia, is typically carried asymptotically by ~10% of healthy adults 72. Summary statistics for the host-associated species with a score of  $\geq 8$  can be found in Supplementary Tables 5-9.

To increase the detection of viral pathogen and potential eukaryotic parasites, we performed a second layer of authentication below the aMeta authentication score thresholds based on an evenness of coverage proxy ( $k$ -mer/reads ratio; see Methods). This led to the detection of Hepatitis B virus (HBV) in individual s.167 (Supplementary Fig. 18), with potential detections ( $k$ -mer/reads ratio  $\geq 1$ ) of additional HBV reads extracted using MaltExtract and validated by BLASTn (e-value  $1e-5$ ; bit delta  $\geq 6$  interspecies for viruses) in s.155, s.117 and s.315 (525, 143, 50 and 7 reads respectively; Supplementary Table 5)20,73,74. This was performed because the aMeta scoring system, can be less sensitive for viral taxa with particular genetic characteristics (see end of section). We mapped all extracted reads with `bwa aln`75, and generated deamination plots with DamageProfiler76. We then performed *de novo* assembly with MEGAHIT77, and BLASTn73 of the longest HBV contigs indicating genotype D (likely D4) for s.167 and genotype A for s.155 (e-value 0 and 98.4% identity). Because coverage was low, damage plots could only be generated for the partial genomes of s.167 and s.155 to validate ancient status (Supplementary Figs. 19–20). The longest contigs measured 1,669 bp for s.167 and 1,586 bp for s.155 out of ~3,200 bp.”

- Page 14, para 1: 'Detection of HBV' I don't understand what you mean by - 'Detection of HBV in s.167, with additional HBV reads using MaltExtract in <...>' additional to what? What is the difference between detection and just having reads?

To clarify that sentence, we now distinguish between “detected” and “potential detection.” “Detected” denotes samples that pass validation based on authentication scores and/or visual investigation of the authentication plots knowing the genetic characteristic of the species. Site-attested primary pathogens are then re-evaluated under the thresholds across all individuals with  $k$ -mer/reads ratio  $\geq 1$ , and the ones yielding reads for that species with MaltExtract were described as potential detection when they had less than 100 reads and therefore didn’t have authentication plots for validation.

- Page 14, para 1: 'Damage plots count only be generated' - is this the Malt Extract damage plots, or did you do alignment and then use of e.g. mapDamage?

Thank you for flagging this. We have clarified in the Results that the damage plots in Supplementary Fig. 19-20 and Fig 22-25 extracted using MaltExtract were generated after mapping with `bwa aln` to the species reference, followed by deamination and fragment length profiling with DamageProfiler. We clarify this in *Page 18 lines 527-550*

“Additional potential detections were identified in s.155, s.117, s.315, s.133, s.101 and s.197 ( $k$ -mer/reads ratio  $\geq 1$ ), with HBV reads extracted using MaltExtract20 (Supplementary Table 8). Those reads were submitted to BLASTn validation (e-value  $1e-5$ ; bit delta  $\geq 6$  interspecies for viruses) yielding in total 525, 143, 50 and 7 non-ambiguous HBV-validated reads from s.167, s.155, s.117 and s.315 respectively. (Supplementary Table 8)20,73,74. We implemented this second layer of authentication because the aMeta scoring system can be less sensitive for viral taxa with particular genetic characteristics (see end of section). We mapped all extracted reads with `bwa aln`75, and generated deamination plots with DamageProfiler76 (Supplementary

Figs. 19-20). We also performed *de novo* assembly of the extracted reads with MEGAHIT77 and used BLASTn73 on the longest HBV contigs, which indicated genotype D (likely D4) for s.167 and genotype A for s.155 (both with e-value 0 and 98.4% identity). Due to low coverage, damage plots could only be generated for the partial genomes of s.167 and s.155 to validate ancient status (Supplementary Figs. 19–20). The longest contigs measured 1,669 bp for s.167 and 1,586 bp for s.155 out of ~3,200 bp.

Similarly, human parvovirus B19 (primate erythroparvovirus 1; B19V) was detected in individuals s.131, s.133, s.157 and s.315, with 1339, 906, 713 and 477 reads respectively, identified through MaltExtract and confirmed by BLASTn (e-value 1e-5; bit delta  $\geq$  6 interspecies for viruses) (Supplementary Table 6)20,73,74. Traces of B19V reads (<10) were found in 5 more individuals and BLASTn-validated (4, 1, 5, 2 and 1) without ambiguous assignments (bit delta  $\geq$  6 interspecies) (Supplementary Table 6). Authentication plots for s.131 (Supplementary Fig. 21), together with complementary damage plots for all four individuals (Supplementary Figs. 22–25), confirm the ancient origin of the partial genomes of B19V. *De novo* assembly using MEGAHIT77 yielded longest contigs of 342 bp for s.131 and 210 bp for s.315. BLAST analysis of these contigs indicated genotype 2. The longest contigs from the other two individuals (s.133: 174 bp; s.157: 304 bp) were insufficient for confident genotype attribution by BLAST.”

- Page 14, para 1: 'longest contigs' -> you've not said you did *\_De novo\_* assembly until the next sentence (if that is what you are referring to?)

The sentence has now been reordered to mention *de novo* analyses before the contigs length.

- Page 14, para 1: 'De novo assembly and BLASTn' why do you mention BLASTn as the tool specifically and not the tool you used for assembly?

Thank you for flagging this. We now explicitly state the assembler, which was MEGAHIT in *Page 18 line 533*.

- Page 14, para 1: 'Interestingly, HBV' why is it interesting that it's 3/4 being women? You have a low total count of individuals, so just through stochastic sampling this may not be particularly meaningful

We agree with the reviewer that, given the low number of HBV-positive individuals, the observation that three out of four were women is likely due to stochastic variation rather than a meaningful pattern. We have therefore removed this sentence from the manuscript.

- Page 14, para 2: 'reads respectively' I'm a little confused why you give very few statistics about *\_S. pneumoniae\_* and *\_M. leprae\_* but are going into so much detail about the metagenomic results of the viruses, could you go back and provide more of these basic stats for the bacteria too?

We thank the reviewer for this helpful suggestion. We have now added the same basic screening statistics for *M. leprae*, *S. pneumoniae* and also *Parvimonas micra* in the Supplementary Tables (Table 5-7) and briefly refer to these values in the Results text alongside the viral statistics. *Page 19 lines 565-585*: “In the case of *Streptococcus pneumoniae*, the evenness of coverage proxy did not increase detection of additional isolates. Reads assigned by MALT to this species were extracted from most individuals in this study ( $k$ -mer/reads ratio  $> 1$ ). The BLASTn validation confirmed accurate species detection for the isolate in s.155, which had an aMeta score of 9. Among the extracted reads, 1902 were validated as *S. pneumoniae*, 871 were classified as ambiguous and 95 were assigned to another species (e-value  $1e-5$ ; bit delta  $\geq 5$  interspecies for bacteria; Supplementary Table 6). The other signals did not yield a majority of *S. pneumoniae*-assigned reads compared to either the ambiguous category or the reads assigned to other species.

Similarly, *Parvimonas micra* was validated in both FASTQ batches of individual s.133, which had an aMeta authentication score of 8 (s.133\_A and s.133\_B), with a majority of reads validated by BLASTn, whereas the reads extracted for this taxon in other individuals were predominantly assigned to other species (e-value  $1e-5$ ; bit delta  $\geq 5$  interspecies for bacteria; Supplementary Table 5).

For *Mycobacterium leprae*, we only could validate the detection in individual s.313 (Supplementary Table 7 and Supplementary Discussion). We extracted 11,887 reads using MaltExtract20, from which 10,545 were BLAST-validated73 (e-value  $1e-5$ ; bit delta  $\geq 5$  interspecies for bacteria; Supplementary Table 7). This *M. leprae* detection was low coverage, with at most 29,000 mapped reads and a mean depth of  $\sim 0.54\times$  (first eager run, see Methods). This sample was therefore selected for capture enrichment.”

- Page 14, para 2: same again as above, slightly more detail about the assembly would be nice (at least tool name)

Thanks, tool name added.

- Page 14, para 3: 'atypical coverage profiles' what do you mean by this exactly? Is this depth of breadth coverages? What are you expecting in 'normal' ones, and how do these atypical ones look like? It would be nice to see more in the discussion what the limitations are of aMeta with viruses (something that I do not really see), and given the authors are also the developers of that pipeline, how do you plan to (or could theoretically) address this - this could be a useful knowledge for the wider aMeta user community to know how to address this limitation.

Thank you for pointing this out. We now clarify what we meant by “atypical coverage”. HBV and B19V are DNA viruses whose genome architecture leads to uneven breadth of coverage in double-stranded libraries. HBV is partially double-stranded with a single-stranded gap, and B19V is single-stranded with palindromic terminal repeats that form hairpins. In practice, this means that double-stranded regions, or hairpin-stabilised ends in B19V, are over-represented, while single-stranded regions are sparsely covered. In aMeta, uneven breadth of coverage

reduces the authentication score, and viruses also tend to show lower average nucleotide identity and different edit-distance curves than bacteria, which further depresses the score. In our data, both HBV and B19V fell below the default aMeta threshold for this reason despite having sufficient read counts and typical deamination, and we therefore validated them with BLASTn and, for B19V, phylogenetic placement. We have added a short paragraph in the Discussion outlining these limitations and how we plan to mitigate them for viral taxa. *Page 24 lines 759-76:*

“In addition to *M. leprae*, we identified hepatitis B virus (HBV) and human parvovirus B19 (B19V), respectively responsible for hepatitis B and the fifth disease (colloquially called slapped-cheek syndrome). They were detected using a proxy for evenness of coverage (see Methods). Indeed, HBV’s partially single-stranded genome and B19V’s single-stranded genome with palindromic terminal repeats produce atypical coverage profiles that stringent evenness of coverage thresholds can filter out. Similarly, faster mutation rates in viruses can lead to false negatives when using identity filters. Nonetheless, the deamination patterns observed for both viruses support their authenticity.”

- Page 14, para 3: 'This bias may be...' it's nice you go into more detail on the last sentence with my previous comment, but you should lead the transition of bacteria to virus results on this page with this, as this explains to the reader *\_why\_* you went below the original aMeta threshold, and expand on it in the discussion (as I explain above)

Thanks, we have explained why those viruses might be less easy to detect in the comment above and in the result section and have removed that last sentence.

- Page 14, para 4, 'Targeted enrichment' why did you only capture *\_M. leprae\_* even though you mentioned it at the beginning of the metagenomics results section?

We do not have capture sets for all pathogens, and we purchased the *M. leprae* set because we have found it several times across our projects. For this study, which aims to document archaeology, genetics and health, we consider reporting low coverage detections sufficient without capture. The inferred HBV and B19V genotypes are not divergent from known diversity for the period, so we did not pursue evolutionary analysis here. As noted, *S. pneumoniae* may reflect carriage, and we are not aware of a capture kit for this species. We also think it is better in site overviews to report detected taxa to provide a complete picture, so that others can target them later for evolutionary work. This has already happened, with teams contacting us to follow up on detections.

- Page 14, para 4: 'substantially increased' You mention substantially increase read yield to 128k, but you didn't report how many reads you had in the shotgun data in the previous section (see my point a couple above about giving more basic screening statistics for the bacteria)

Thank you for pointing this out. We now explicitly report the pre-capture read count and depth for *M. leprae* in the metagenomics analysis result section and have rephrased this

sentence in the phylogenetics section, *page 20 lines 609-612*:

“Targeted enrichment of *Mycobacterium leprae* using the myBaits Custom Community Panel from Arbor Biosciences 82 increased the read yield from 29,000 to 128,800 (roughly 4.4-fold increase), enabling a mean genome coverage depth of 3.75× and a breadth of 76.2% after relaxed mapping (3.21× and 72.4% after FASTQ trimming and strict mapping; (see Methods).”

- Page 15, para 1: 'manual genotyping' what does 'manual' mean in this case? This rings alarm bells to me, and what that involves is not further expanded in the methods sections - are you literally just looking visually at each position and making a decision? What rules did you apply to picking each allele?

Thank you for raising this. We have removed the term “manual genotyping” from the Results and Methods and now report genotype assignment solely from the phylogenetic reconstruction. Briefly, prior to finalising the pipeline, we did a quick sanity check using IGV against the ~80 lineage-informative SNPs in Monot et al. 2009: sites without coverage were ignored, and at covered sites we kept the major allele. This preliminary check already indicated genotype 2F, which matches the phylogenetic placement. The tree-based assignment is the basis for all statements in the revised manuscript.

- Page 15, para 1: 'The Ibiza genome' Don't change the name to Ibiza genome, refer to it via the individual name as you've otherwise done throughout otherwise it will become confusing (or if you insist, say something like 'Ibiza genome from individual s.131').

We have replaced the term “Ibiza genome” with the sample code s.313, as recommended, to ensure consistency and avoid confusion. We also note that in the same paragraph we had now specified “the *M. leprae* genome retrieved from individual s.313”, so the terminology is now consistent.

- Page 15, para 2: 'In parallel..' please explain your reasoning to also make a maximum parsimony tree (which is arguably more simplistic than maximum likelihood, why not just Neighbour-Joining tree, or go the otherway and make Bayesian trees [\_I\_ know why, but other readers may not understand])?

We thank the reviewer for helping us to make our manuscript more accessible. We have now added an explanation in the Results section where we mention Maximum Parsimony. In *page 20 lines 627-631*. “We used parsimony because it selects the tree with the most parsimonious changes and *M. leprae* changes slowly and mostly by single-base mutations with very limited homologous recombination via horizontal gene transfer. In practice parsimony and maximum likelihood give very similar trees for *M. leprae*, as shown in 84”.

- Page 16, para 1: 'Comprehensive archaeological..' This is too strong - it is definitely not comprehensive: there is no detailed description of archaeological context of each individual (grave goods, positioning etc) nor the site, the 'genetics' is purely population genomics - nothing on e.g.

genetic health, and ultimately after screening the 'metagenomics' consists of a tree of only single species - not of the other potential pathogens. I would remove 'comprehensive'

We have removed the term “comprehensive” and added new information regarding the archaeological context of each individual (e.g., grave goods, positioning) as well as the overall site. This information has been included in a new paragraph in the first section of the Results, under Osteological and Archaeological Context.

- Page 16, para 1: 'offering new insights...' I would be wary of saying you have given an insight into the health status given you've only identified the presence of the microbe, while it is most likely that the individual did suffer despite no osteological signatures (in the sense that to have sufficient load to detect in an ancient DNA sample, there must have been a large number of microbial cells). I would feel more amendable if you said 'infection status' or something similar. Alternatively you should provide more archaeological/osteological context about each individual so you have two sources of proof about the state of the health of the individual

We have removed “health status of the individuals” for this sentence.

- Page 16, para 2: 'our study focuses on...' this is the first time you mention the cemetery is urban, this compounds my criticism that you state this is a comprehensive study but actually there is very little archaeological description or context about the site nor burials

This is now addressed: more complete archaeological information has been added in the Introduction and the Osteological and Archaeological Context section, including details about the urban nature of the cemetery and additional context for the burials.

- Page 16, para 2: 'Individuals s.157 (XX)' why do you mention the biological sex in the first line? You don't do this for any other individual and does not seem to be relevant for any of the points made in this paragraph?

We have removed the reference to biological sex from this sentence.

- Para 17, para 4: 'diverse range of' I would not agree you've revealed a \_diverse\_ range of pathogen and commensal microbes, you've reported 2 bacteria (only one actually analysed) and 2 very common viruses (unless this is what you mean by commensal?), if you want to state this sentence like this about this you should discuss more about the wider metagenomic profile, even below the aMeta score.

We have removed the term “diverse range” to more accurately reflect the data reported.

- Para 17, para 4: 'although osteological evidence...' you say evidence of leprosy has been found in other cemeteries, but you've not said so far if your individuals displayed this - this should be described in the introduction or results in an expanded archaeology/osteological description

This is now addressed: more complete archaeological information has been added in the Osteological and Archaeological Results section.

- Para 18, para 2: 'Interestingly...!' Again, you've only now mentioned that there is no differentiation in grave treatment, but you've not described this in the introduction (if previously published elsewhere) or in the results (if new)

This is now addressed: more complete archaeological information has been added in the Osteological and Archaeological Context section.

- Para 18, para 2: see my point two above ('although osteological...') - this should have been described much earlier! And what do you mean by that the available remains are not sufficient? This is a critical bit of information that should be described in the results! It gives more context why you could not expand more on this topic, you should not hide this buried in the discussion!

We agree that this information is essential for contextualizing the osteological assessment and have now moved it to the Results section. The Results now include the following sentence: *page 6, lines 183-187:*

“No grave goods, trauma or diagnostic skeletal pathology were observed in the sequenced individuals. However, the absence of certain elements, such as the facial bones in individual s.313, makes it difficult to rule out diseases like leprosy.”

We have also revised the relevant sentence in the Discussion to read:

*page 24 lines 755-757:*

“Although the available skeletal remains, notably lacking certain elements such as the facial bones in individual s.313, are not sufficient to completely exclude the presence of leprosy osteologically, the studied bones do not show any apparent signs of the disease.”

- Para 18, para 5: 'This burden of infection' This might be being very nitpicky on my part, but I find the two sentences in the middle a little contradictory: you say burden of infection (implying a higher 'suffering'), but if it's the same as everywhere in else in Europe at the time, then then really it is just the basic microbial background of individuals. Do you have evidence that these viruses (in particular) are actually having a negative effect on the individual? Can you really call this disease? If they are everywhere they could be a normal part of our flora and it's more about abundance or disequilibrium that then causes health problems. Your results can't really show what the *\_level\_*/abundance of them are, and thus can't say anything about the *\_burden\_* of these microbes are on the individuals (note that even though I've said this, I do not have a clinical background so apologies if burden of infection has a specific term in the literature that you have applied here that I'm not aware of)

We recognise that the burden of infection might sound like it implies a higher suffering and have now replaced it by the rate of infection. We were comparing this rate of infection with today, where medicine makes it possible to fight off these diseases. Note however that HBV

and *Mycobacterium leprae* are never commensal microbes and are primary pathogens, which means that they would necessarily create an infection in the host. Like with most diseases, if the immune system manages to fight off the infection, then, it might never show symptoms specific enough to diagnose a specific disease especially on the skeleton where only a few chronic diseases can leave traces, like leprosy, tuberculosis, syphilis and more rarely brucellosis. Once the infection is resolved, there should be no more traces of the pathogen in the body, at least this is what is the case for most diseases today, except for the presence of specialised antibodies from the immune system. Therefore, we argue that when there is detection, although at low coverage, there is infection, but we cannot assess how sick the person was. For Human parvovirus B19, this is different, since it generally creates the fifth disease in childhood and then persists in the body without having a known effect.

- Para 18, para 5: 'crucial layer' people having very common microbes on their body is not new at all, so I don't think this is a 'crucial' layer, I would remove

We have removed the term “crucial layer” as suggested. And changed the sentence in *page 24 and 25 lines 784-798* to:

“Overall, we identified molecular evidence of infection to primary pathogens in at least 3 of the 13 individuals analysed, involving *M. leprae* and HBV. Including the two individuals with low-coverage HBV signals raises this number to 5 and a rate of 38%, which is comparable to proportions reported in broad-spectrum pathogen screening studies at other medieval sites23,78. When we also consider likely infections with *S. pneumoniae* and *P. micra*, the number of individuals with at least one detectable pathogen increases to 7, or about 54%.

Because B19V in adults most likely reflects persistence after childhood infection, we treat it separately. In our material, four individuals yielded partial B19V genomes and a further five individuals carried very low coverage traces. This translates into a minimum of 31% (4 of 13) and a maximum of 69% (9 of 13) of individuals with detectable B19V DNA. Given the high prevalence of lifelong B19V persistence in modern populations96, the true proportion in our sample probably lies toward the higher end of this range. Our metagenomic analyses provide direct evidence for the presence of these pathogens in medieval Ibiza, establishing a foundation for future comparative work to explore their wider historical and epidemiological patterns.”

- Para 18, para 5: the last sentence is too broad, you've not really made any real links or described implications between mobility/ancestry and the microbes, something you can only say when you have a large stratified number of genomes and strain analyses - all you are saying in this manuscript is this single genotype is present in this island - you have no other comparison or deeper analyses e.g. comparing with individuals on the continent in the same time frame (which again I don't blame you, there are not sufficient public data either to do this currently), but I would tone this down - i.e. you have not shown any intricacies.

We agree with the reviewer that our data do not allow us to establish detailed links between mobility, ancestry, and microbial infections. We have therefore toned down the conclusion

and rephrased the relevant paragraph accordingly. This paragraph has been modified as described in our response above.

- Page 19, para 2: 'genomic traces' don't downplay yourselves here, a 3x *M. leprae* genome is not traces!

We thank the reviewer for this comment. We agree that the wording “genomic traces” may understate our results. We have revised the text removing “genomic traces”.

- Page 19, para 2: 'intertwined...' once again, you are not demonstrating where the *M. leprae* came from per se (did the genotype come from the 'native' Iberians on the continent, or travelled purely through the a lineage infecting newcomers from al-Andalus?) so I don't think you have demonstrated 'intricacies' of mobility and interaction, I would tone this down

We thank the reviewer for this comment. We agree that our data do not allow us to definitively determine the origins of the *M. leprae* genotypes or fully reconstruct the detailed patterns of population mobility and interaction. Accordingly, we have toned down the original statement and revised the text to avoid over-interpreting the data, presenting the pathogen results in a more cautious and accurate manner.

- Page 19, para 3: 'By integrating...' I don't think you've really integrated pathogen detection and genomic ancestry, at least in what I have read here. You've not discussed in depth between the specific link of the microbial genotypes with the ancestry of the individuals they are coming from: your whole discussion is actually surprisingly not integrated across the different methods reported here, you have one section on ancestry and one on the microbes (other than vaguely broad 'people from islamic cemeteries).

We thank the reviewer for this comment. We agree that the manuscript does not fully integrate pathogen detection with host genomic ancestry. Accordingly, we have toned down the original statement and revised the text to present the ancestry and microbial results more separately, avoiding overstatement of integration.

- Page 19, para 5: 'These methods were complemented by...' where are the results on this?! These are not described at all!

We thank the reviewer for pointing out that the results of these complementary osteological assessments were not sufficiently described. We have now expanded this section to include a clearer description of the analyses performed and the type of information obtained. The revised text in *page 26 lines 858-869* now reads:

“The analysis of the human remains focused on estimating the minimum number of individuals and reconstructing their biological profiles, including age-at-death, biological sex,

stature, and the presence of any pathological alterations or trauma that could inform on past living conditions and funerary practices.

Sex was estimated primarily from pelvic and cranial morphology97-99, complemented by metric assessments100. Age-at-death estimation was based on skeletal development101 and dental formation102,103 as well as the morphology of the pubic symphysis, auricular surface, and rib ends when preserved104-106. These methods were complemented by observations of dental wear, ante-mortem tooth loss, and age-related pathological conditions.

Stature was estimated, when possible, from the maximum length of long bones (preferably the femur), applying the regression equations developed by107 and108.”

The methods used here, summarizes standard qualitative assessments routinely included in osteological studies. These observations serve as supportive indicators for age and health estimation but do not typically yield quantifiable results suitable for further reporting.

Therefore, their inclusion in the text is intended to acknowledge the application of these standard methods rather than to present new data.

- Page 20, para 1: 'established protocols...' Was this following exactly with the same reagents (and in some cases critical consumables)? Please state if so, otherwise this is not reproducible! I would rather see a link to the **\*\*exact\*\*** protocol you used?

We thank the reviewer for this comment. While we cited the original study (109) to acknowledge the developers of this DNA extraction approach, we would like to clarify that the protocol described in our Methods section reflects exactly the reagents and consumables used in our study. Specifically, 1 ml of digested extract was combined with 13 ml of binding buffer containing 5 M guanidine hydrochloride, 40% (v/v) isopropanol, 0.05% Tween 20, and 90 mM sodium acetate (pH 5.2). DNA purification was performed using 50-ml silica columns (Roche, High Pure Viral Nucleic Acid Large Volume Kit), and DNA was eluted in 45 µl of Elution Buffer (EB; Qiagen). Blank controls were included at each step.

We have rephrased the text in the Methods to make it clear that these reagents and consumables were used exactly as described, ensuring full reproducibility of the extraction procedure.

- Page 20, para 2: 'as described in...' same again: was this protocol followed using exactly the same reagents and consumables? Please state if so otherwise this is not reproducible! I would rather see a link to the **\*\*exact\*\*** protocol you used?

Similar to the previous point, we cited the original study (Meyer, M. & Kircher, M. 2010) to acknowledge the developers of this DNA library approach. We would like to clarify that the protocol described in our Methods section exactly reflects the reagents and consumables used in our study.

We have rephrased the text in the Methods section (*page 27, line 903*) to make this clear, adding the phrase “with the following specific settings:”.

- Page 20, para 3: '(Design ID:...): This is very good, I commend the authors on being so specific! Thank you!

Thank you.

- Page 20, para 3: 'Novaseq S6000': I Think that is a typo, shouldn't it be just 6000? But also, what kit was used? Was it paired-end, and what was the number of imaging cycles? This can be important information for users who may want to reanalyse your data (e.g. can be important for *\_de novo\_* assembly of the metagenomic data)

We have now added to method the following sentences in *page 28 lines 933-936*:

“Purified libraries were pooled in equimolar concentrations and sequenced on an Illumina NovaSeq 6000 at the SciLifeLab National Genomics Infrastructure (SciLifeLab-NGI) in Stockholm using an S4 flowcell with XP clustering; either the S4-300 (v1.5) kit or the S4-200 XP kit was used, both supporting standard paired-end 2×150 bp reads.”

- Page 21, para 1: 'were demultiplexed..' with what tool/settings/version? This is isn't useful otherwise

Sequencing reads were demultiplexed based on their sample index at SciLifeLab-NGI. We have now clarified this in the Methods section, *page 28 lines 936-938*, as follows:

“Sequencing reads were demultiplexed during BCL to FASTQ conversion using bcl2fastq v2.20.0.422 (CASAVA), with a Sanger / phred33 / Illumina 1.8+ quality scale at SciLifeLab-NGI.”

- Page 21, para 1: 'Adapters were trimmed' was all adapter trimming with default parameters?

We have now added to the text the details of the adapter trimming and read merging in *page 28 lines 938-941*. “Adapters were trimmed using Cutadapt v. 2.3112 (`--quality-base 33 --nextseq-trim=15 --overlap 3 -e 0.2 --trim-n --minimum-length 15:15`), and fastq reads were merged using FLASH v. 1.2.11 (`--min-overlap 11 --max-overlap 150 --allow-outies`)”.

- Page 21, para 1: 'Using Burrows-Wheeler...' was BWA mapping also performed for the microbial work? Or is this just just for the human pop gen?

Yes, we applied BWA for the human population-genetics alignments (mapping reads to the human reference genome). To avoid confusion, we rename this section in methods from Sequencing and Data Processing to Sequencing and Human Data Processing

For microbial work, however, we followed the aMeta workflow and only used BWA to map the reads extracted with MaltExtract afterwards, which we have clarified in the method and the results section.

- Page 21, para 1 'merged using Samtools' which subcommand? There are many different ones?

We have now clarified in the text that the BAM files were merged using *samtools merge* in line 944 page 28

- Page 21, para 1: 'a modified version' If modified, please make your modified version available on GitHub or similar, or describe the exact modifications made (but the former greatly preferred), otherwise this work is not reproducible

Thanks for pointing this out. The modifications we made to the original script were purely cosmetic or minor functional additions and do not alter the input or core logic. Specifically, the modifications are:

- Addition of a `--count_file` option to record counts of sequences;
- Tie-breaking in `calc_consensus()` using `random()` when multiple bases have equal scores.

We have now incorporated these details directly into the Methods section to ensure clarity and avoid maintaining multiple versions of the script for the same output. As stated on page 28, lines 943–948: “Fastq files from different sequencing runs of the same library were merged using Samtools v1.17 (`samtools merge`)113, followed by consensus read creation using the script `FilterUniqueSAMCons.py` (<https://bioinf.eva.mpg.de/fastqProcessing/> with the following modifications: addition of a `--count_file` option to record counts of sequences, and tie-breaking in `calc_consensus()` using `random()` when multiple bases have equal scores.”

- Page 21, para 2: 'PMDtools', what version/settings?

We used PMDtools v0.60, as now specified in the text, with the default settings.

- Page 21, para 2: 'contamination estimates...'; why not refer to the two tools by name as you are doing elsewhere? What about versions and settings?

We thank the reviewer for the comment. We have now updated the text to specify the two mtDNA contamination estimation tools by name: one following Green et al. (2008) (ref 115) and the other using ContamMix (ref 116). The versions and settings used for each tool have also been added to the Methods section for clarity and reproducibility.

- Page 21, para 3: 'TrimBam from BamTuils' what version of Bamutils? What settings?

We used bamUtil v1.0.15 (TrimBam), now specified in the text. Analyses were performed with the default settings.

- Page 21, para 7: 'applied depending on specific analysis' what exactly does this mean? What were the criteria and when applied? Be explicit

To address this comment, we revised the paragraph for clarity and removed the vague phrasing. The original text read:

“Variant- and individual-level filters were applied depending on the specific analysis. For example, individual s.155 was excluded from local ancestry analysis due to excessive missing data, as RFMix v1.5.4 does not tolerate missing genotypes. Other samples with relatively higher levels of missingness, such as s.313 and the two sub-Saharan African individuals, were retained but analysed separately to avoid bias.”

We have now simplified the paragraph to state the explicit criteria and circumstances (*page 29, lines 992–995*):

“Individual s.155 was excluded from local ancestry analysis due to excessive missing data, as RFMix v1.5.4 does not tolerate missing genotypes. Individuals s.313 and s.197 with relatively higher levels of missingness, were retained but analysed separately to avoid bias”

This revision removes unnecessary wording and makes the criteria explicit. These cases (s.155, s.197) were the only exceptions, and we highlight them here for transparency.

- Page 22, para 2 'we standardised the SNP identifiers' how exactly did you do this? With scripts in R or python? This seems like an important file to make available for people to reproduce your analyses

The script to prepare the datasets are now available on GitHub (<https://github.com/quenllavarela/medieval-iberia-genomics>) and archived in Zenodo.

- Page 22, para 4: 'The Iberian component included 66...' why is there such a imbalance in the number of individuals between the European representatives and the Africans? Because of the greater genetic diversity in the latter? Please provide a justification (at least for non-specialists such as myself)

We appreciate the reviewer’s comment. The imbalance in sample sizes reflects underlying differences in genetic diversity across regions and the need to capture this diversity in the reference panels. Sub-Saharan Africa harbours the highest levels of human genetic diversity worldwide; therefore, a larger number of reference individuals is required to represent this diversity adequately and provide stable ancestry estimates. In contrast, European populations show substantially lower genetic diversity, and a smaller number of individuals is sufficient to capture the relevant variation.

For the European/Iberian component, we included 33 CEU and 33 Basque individuals (66 total) to match the size of the North African/Middle Eastern component (33 Mozabite + 33 Bedouin). For the sub-Saharan African component, we included multiple populations (Luhya, Yoruba, Gambian, Sierra Leonean) with ~100 individuals each to span the broader range of African genetic variation that emerged in our PCA and ADMIXTURE analyses. Exploratory RFMix runs confirmed that these were the groups most consistently detected and informative for our dataset.

We add in the methods section *page 30, lines 1018-1028*: “We selected the following reference populations from the 1K-HGDP dataset to represent plausible ancestral sources for the Ibiza individuals. The Iberian component included 33 CEU and 33 Basque individuals, while North African/Middle Eastern ancestry was represented by 33 Mozabite and 33 Bedouin individuals. Because Sub-Saharan Africa harbours the highest levels of human genetic diversity, we included larger reference panels from this region: 97 Luhya (East Africa), and 97 Yoruba, 97 Gambian, and 98 Sierra Leonean individuals (West Africa). These populations were chosen based on PCA and ADMIXTURE results, which identified them as the most informative sources of African-related ancestry in the ancient individuals. The larger Sub-Saharan African panels ensure adequate representation of regional diversity, whereas the smaller but balanced European and North African panels sufficiently capture variation relevant to our dataset.”

- Page 23, para 2: 'computed the covariance' - how were you doing this, in custom R/python scripts?

Yes, the covariance was computed using the `cov` function from the base stats package in R. The R script has now been made available at (<https://github.com/quenllavarela/medieval-iberia-genomics>) to ensure reproducibility and is referenced in the Methods section.

- Page 23, para 6: 'NgsRelate, Kin, READ (v2)...' Why the versions for READ, and not the others, please at least be consistent (or rather versions and settings for every tool used here! Or make your scripts/commands available on e.g. a GitHub repo so a reader can look directly and so you can save time in writing!) - Please check this across the entire of the methods section as this keeps coming up (I won't mention this again, but definitely keeps happening post this comment)

We thank the reviewer for pointing this out. The versions of all software tools are now included in the Methods section for consistency. In addition, we have expanded the Methods section to provide details on parameters used when analyses were not run in default mode. The metagenomic scripts, as well as the parameters and in-house scripts for the population genetics analyses, are deposited in Zenodo and GitHub to ensure transparency and reproducibility.

- Page 24, para 2: 'choosing up to 10 individuals' why 10? Please justify

We thank the reviewer for this comment. We selected up to 10 individuals per population to balance representation across groups while minimising the impact of missing data, as ADMIXTURE is sensitive to uneven sample sizes and differential data quality. Preliminary tests showed that using up to 10 high-quality individuals per population was sufficient to capture the major ancestry components without inflating cluster contributions from larger populations. In addition, for several populations in the dataset, 8 to 10 individuals represented the maximum number of suitable samples available. We have added this justification to the Methods section *page 32 lines 1098-1102*: “To minimise sample-size imbalances and reduce the influence of missing data in ADMIXTURE analyses, we selected up to 10 high-quality individuals per population. Preliminary testing indicated that this number was sufficient to capture the major ancestry components while avoiding overrepresentation of populations with larger sample sizes. In addition, for several populations in the dataset, 8 to 10 individuals represented the maximum number of suitable samples available.”

- Page 24, para 3: 'Using the Namibian' why did you pick the Namibian individual?

We thank the reviewer for this comment. We chose the Namibian Ju’hoansi as the outgroup for our  $f_3$ -statistics because this population is deeply divergent from other African populations included in our analyses. The Ju’hoansi, part of the southern African Khoi-San-speaking groups, represent one of the earliest diverging lineages of modern humans within Africa and have relatively low levels of recent admixture with Bantu-speaking or Eurasian populations (Schlebusch et al., 2017). Using a highly divergent and minimally admixed population as the outgroup allows us to reliably measure shared genetic drift between populations X and Y without bias from recent gene flow. Moreover, the Ju’hoansi are widely used as an outgroup in African population genetic studies (e.g., Fortes-Lima et al., 2024), providing precedent and comparability with previous research. This rationale has been added to the Methods section. *Page 32 lines 1121-1125*:

“The Ju’hoansi, a southern African Khoi-San-speaking population, represent one of the earliest diverging lineages of modern humans and have low levels of recent admixture. This highly divergent and minimally admixed population provides a reliable outgroup for measuring shared genetic drift between target populations and is widely used in previous African population genetic studies53,135.”

- Page 24, para 4: 'pipeline aMeta' - what settings, and more importantly for metagenomics, which database!?! This is a hugely important thing in metagenomics and greatly influences your output! Furthermore, as aMeta is using a snakemake, it should be easy to provide the exact configuration file you used, correct?

We understand the concern. aMeta requires fewer configuration options than pipelines such as nf-core/eager. The only user-set parameters are the read and unique k-mer thresholds applied after KrakenUniq to keep likely true positives for dynamically building the MALT reference used for mapping. These are already specified in the Methods (100 taxReads and

1,000 unique k-mers). In addition, some analyses can be turned off if not required. We turned off mapDamage after bowtie2 mapping, since that branch is a quick-check path without the LCA layer, but this does not influence the results we report. Another user might keep it enabled for a quick check. We have now added the config file to the GitHub of this project.

Our config.yaml is the same as the example in the aMeta README, except for absolute paths to our server. If you meant the Snakemake profile, we will make the one we used available on the project GitHub as an example, noting that profiles are highly cluster-specific, unlike Nextflow configuration files.

- Page 25, para 1: 'were retained for downstream analysis' how exactly did you do this filtering, in R/Python? And just for clarity: are all of those criteria are considered `_within_` the authentication score, or you were filtering for the aMeta score AND the rest of the criteria but manually?

Thanks for flagging this. All metrics listed (edit distance, coverage evenness, read length, deamination profile, PMD, ANI and read count) are inputs to the aMeta authentication score. We did not re-apply these criteria outside aMeta. For downstream analyses we retained only taxa with an aMeta score  $\geq 8$ . The heatmap was generated in R using the aMeta plotting script after subsetting the aMeta score table to score  $\geq 8$  inside it. We then annotated taxa as likely environmental or likely host-associated as described in *page 33 lines 1141-1152*.

- Page 25, para 1: 'specific cases' - does this only refer to `_M. leprae_` not other species otherwise not described (assuming you're referring to the index hopping section of the SI)?

Thank you, this sentence has now been corrected in *page 17 and 18 lines 503-505* to: "In addition, whenever a pathogen is attested at the site, we reevaluate its presence at low coverage across all individuals by analysing all the reads assigned to it via MALT70 when the evenness of coverage proxy is bigger than 1."

- Page 25, para 2: 'Calculating the unique...' How did you calculate the k-mer to read ratio? Do you have a script or notebook?

We calculated it with awk, by simply dividing #4 by column #2 in the aMeta KrakenUniq output. However, a notebook is a good idea, and we have now reworked through our steps and generated an .md in the github for that part of the analysis.

- Page 25, para 2: 'Known human primary pathogens' why did you go away from the aMeta score? Please justify

We recognise that this needed justification and have now explained the two different layers of authentication we used in our analyses earlier in this response and have expanded on it in complement to the discussion in the result section, while specifying why briefly before the secondary layer of authentication in the method section.

- Page 25, para 3: 'We then used MaltExtract', within aMeta or manually?

We have now specified that MaltExtract was run with the same parameters outside aMeta. It is the exact same command, however, it allows us to extract all reads for a species in all individuals, not only in the ones that passed the 100 taxReads and 1000 k-mers threshold.

- Page 25, para 3: 'BLASTn on HPC' what is this exactly? I've not heard of it and I don't see it referenced in the BLAST+ paper? What database?

We thank the reviewer for pointing this out. The phrase “BLASTn on HPC” referred to the use of the standard BLASTn program from the NCBI BLASTn suite (v2.15.0+), which we executed on our institution’s high-performance computing (HPC) cluster to efficiently process multiple samples in parallel. The searches were performed against the NCBI nucleotide (nt) database Version 5 of the NCBI nt database), which had been downloaded locally to the cluster. We have clarified this in the revised method text in *page 33 lines 1167-1171*:

“We verified each assignment of these potential detections using BLAST v.2.15.0+ on our cluster against the downloaded NCBI nt database (BLAST DB Version 5) with parameters (-perc\_identity 90, -qcov\_hsp\_perc 80, -evalue 1e-5, -word\_size 11, -soft\_masking false, -max\_target\_seqs 50, -max\_hsps 1) (Supplementary Table 5-9) 73,74.”

- Page 25, para 3: 'NCBI nucleotide databases' what date/version of the NCBI nt database did you use? This database is updated on a regular basis

Thanks. See previous response for the paragraph that we have included in the manuscript to address this comment. We used the online version of BLAST in May 2025 and have now clarified it in the text.

- Page 25, para 3: 'Additional damage' - from what files did you exactly generate these from? DamageProfiler only takes BAM/SAM files (I believe)? Or from MaltExtract? Or mapping reads back to your assemblies?

Thanks for flagging this. We have now corrected that section to clarify that we used bwa aln to map the MaltExtract reads back to the species and generate the damage plots with DamageProfiler in the method section. *Page 33 line 1164-1167*: “We then used MaltExtract v.1.7 64 with the same parameters as in aMeta, to extract HBV, B19V, *M. leprae*, *S. pneumoniae* and *P. micra* reads from MALT v.0.6.2 70 alignment files across all individuals with a k-mer to read ratio  $\geq 1$  regardless of read count, since those pathogens were now attested at the site.” *and page 34 line 1176-1178*: “Additional damage, read length and edit distance plots were generated using DamageProfiler v.1.1 on the extracted reads mapped with bwa aln v.0.7.17 75,76.”

- Page 25, para 4: 'nf-core/eager...' as with snakemake above, please provide the config file and/or command used, note that nf-core/eager should provide a full list of versions of all tools and citations used within the pipeline somewhere that you can use here.

Thanks. The config file and submission scripts are now uploaded to the GitHub repository for this project. We have also added the full list of tools and versions provided by nf-core eager to the first mention of it in the method section.

- Page 25, para 5: 'damage removal was applied..' to which FASTQ files were you applying this too? Post-capture or raw I'm assuming post-capture but this is not described? Is this still happening within nf-core/eager or manually?

Thank you for pointing this out. Damage trimming with fastp was applied to the adapter-trimmed FASTQ files for both the shotgun and capture libraries, outside the nf-core/eager workflow. We first inspected deamination patterns to determine the effective UDG treatment of each library, then ran fastp on the adapter-trimmed FASTQs and used these damage-trimmed files as input for the final nf-core/eager runs. We have clarified this in the Methods section *page 34, lines 1181 and 1197*.

- Page 25, para 6: 'more stringent edit distance' are you referring to bwa aln mapping here?

Thank you for the comment. Yes, this refers specifically to the bwa aln -n parameter. We have now replaced the term “edit distance” with the correct BWA terminology and describe it as a stricter mismatch rate in the revised manuscript. In *page 34, line 1200-1201* we have added “Trimmed FASTQ files were re-processed with nf-core/eager v.2.5.0 again145 using the same parameters, except for a stricter bwa aln mismatch rate (-n 0.2) similarly to84.”

- Page 26, para 2: 'Mycobacterium lepromatosis FASTQ' what GenBank assembly/strain is this associated with, is this the designated *\_M. lepromatosis\_* reference genome? And if not has it been validated as a good quality outgroup for your phylogenomic analyses?

Thank you for the question. The outgroup dataset corresponds to the *Mycobacterium lepromatosis* assembly with GenBank accession JRPY00000000.1. This is the same assembly used as the *M. lepromatosis* reference in previous phylogenomic studies of *M. leprae* and *M. lepromatosis*, including Fotakis et al. 2020 and Bonczarowska et al. 2022, where it is treated as the designated outgroup for comparative analyses. Several other studies (for example Pfrengle et al. 2021 and Urban et al. 2024) refer to *M. lepromatosis* as the outgroup without specifying a different assembly, and we found no alternative high-quality reference consistently used in leprosy phylogenomics.

We have now added the GenBank accession to the Methods (*page 34, line 1209-10*) and clarified that our choice follows the precedent set in these published phylogenomic analyses.

- Page 26, para 3: why the 90% cutoff? If you've got a 3X genome why not 66% or even just 100%?

Thank you for the comment. The 90% cutoff refers to the minimum fraction of reads supporting the called allele at each SNP position in each genome in MultiVCFAnalyzer, not to the mean genome-wide coverage. Since *M. leprae* is a haploid, predominantly clonal bacterium, we expect very little within-sample polymorphism at true SNP positions. We therefore required at least 90% of reads to support the reference or the alternate allele in each genome to minimise spurious SNP calls caused by sequencing error, residual damage or low-level contamination, and this threshold was applied uniformly across all genomes in the analysis. A lower fraction such as two reads out of three ( $\approx 66\%$ ) would admit many sites where the signal is driven by a small number of discordant reads, while a 100% requirement would be unrealistically strict once occasional errors in higher coverage genomes are taken into account.

- Page 26, para 4: 'Manual genotyping' Same again about manual genotyping as I commented in the results section, please describe what you mean by this and what were your criteria/rules for making each call

Thank you for the comment. As clarified in our response to your earlier point on “manual genotyping,” we used IGV only as a preliminary sanity check and do not base any conclusions on manual allele picking. We have now updated the Methods to remove this wording since genotype assignment relies on the two phylogenetic reconstructions.

- Page 26, para 5: 'B19V reads extracted' Why only the the specific B19V reads for assembly, what if you've lost some conserved reads to closely related taxa reducing your coverage?

Thank you for the comment. B19V reads for assembly and consensus calling were extracted with MaltExtract. We intentionally restricted the dataset to reads assigned to human parvovirus B19 in order to minimise mismapping. Given that there are no common, very closely related parvoviruses expected in human bone and teeth, the number of true B19V reads that would be classified only at a higher taxonomic node is likely very small and largely confined to highly conserved, phylogenetically uninformative regions.

In addition, B19V has long inverted terminal repeats, so many genuine reads from these regions map to multiple positions on the linear reference and receive a MAPQ of 0 in BWA. Using a standard high MAPQ filter on all mapped viral reads would therefore preferentially remove genuine B19V reads from the ITRs. Our strategy of first using a competitive taxonomic assignment and then mapping only the B19V assigned reads allows us to keep these fragments while maintaining good specificity. This may slightly reduce depth in the most conserved regions, but has negligible impact on the consensus sequence and phylogenetic placement while reducing the risk of incorporating non B19V reads.

- Page 26, para 5: 'Sequences were aligned using' is this with nf-core/eager as above or manually?

Thank you for the comment. This step was performed outside nf-core/eager. For the B19V analysis we manually aligned the extracted B19V reads and the published ancient genomes with BWA aln using the parameters given in the Methods, then deduplicated and filtered the alignments and generated consensus FASTA files with ANGSD.

- Page 26, para 5: 'fiels' typo! Fixed

### ### Figures

As a general criticism: the population genomics figures are often quite poor quality in the sense the size ratio between the plots and texts are poor. For example, pretty much all the PCA figures it's impossible to read or distinguish anything because:

1. the main text figures are too small (so it's hard to distinguish between so many clusters when there are so many colours)
2. the font size is tiny so it's extremely difficult to read. The supplementary figures are slightly better as they are larger, but the label and axis text are almost always too small.

We appreciate this helpful feedback. All population genomics figures (including the PCA plots) have now been revised to improve readability. Specifically, we have increased the overall figure size in the main text, standardized and enlarged font sizes for labels and axes, and adjusted colour schemes to better distinguish clusters.

In contrast: the microbial genomics figures are generally much more readable and and balanced - for example Figure 9. In cases where the 'contents' of the figure is large and detailed, the font is naturally small. However these has a balanced ratio with the figure components themselves (there is not enough free whitespace to make the text any bigger - such as the large trees in the supplementary), making it acceptable. However in the case of the pop-gen figures there is a lot of free space to make the fonts bigger enough to make them readable..

My only criticism of the microbial genomics figures is the on the two *M. leprae* trees it's very difficult to find the Ibizan genomes as the tree is so large - the red font is not enough to catch a readers eye - please use a larger indicator (e.g. a star)

Specific comments:

- Fig. 3: I have no idea what the span of the ancient iberians is referring to, nor was it immediately clear what areas the 'Zoom' was referring to, I found the different font sizes across all the plot distracting too (e.g. the smaller sized pale blue 'pairwise similarity' was very hard to read)  
Fig 3 has now be changes we homogenize the text code and size and indicate the time grade and the exact populations used in the zoom

Thank you for pointing this out. Figure 3 has now been revised: we have standardized the font style and size across the plot, clarified the time scale, and specified the exact populations included in the zoomed-in section.

### ### Supplementary Tables

- General: a summary of what each table is referring to would be nice (and ideally a legend for each table what each column is referring to)

We have now added a description of each table from the Supplementary Information. In addition, we provide a legend for each table specifying what the table refers to and expand in the headers what each column represents.

- TableS1 appears to have broken coverage column headers for 'tDamage' and 'aDamage' (I'm not sure what they mean, nor what the `}` standards for)

tDamage refers to transitions caused by cytosine deamination, specifically C→T substitutions at the 5' end and G→A substitutions at the 3' end of reads. The headers have now been corrected to 5'Damage and 3'Damage. The stray `}` was a typo, thank you for noticing this.

- TableS1: please make cleaner/computer readable columns by putting units in headers when they are all the same (e.g. for the dates)

Units are now just in the header of Supplementary Table 1

- Table S2d: the column headers appear to be missing? I'm not sure what each one is referring to

Column headers are now added to the new Supplementary Table 2

- Table S6 and S7 starts referring to samples with `{A,B,D}` suffixes... what are these?

When the same sample and library were sequenced across multiple lanes, generating separate FASTQ files labeled as sample\_name\_A, \_B, \_C, and \_D. These suffixes indicate the sequencing lane or batch for each file, which are uploaded to the ENA under the same sample name. This has now been clarified in Supplementary Tables 5 to Table 9

### ### Misc

One last half-throw away potentially poorly thought through comment as I submit this: have the authors considered the wider impact these results may have in the present day social tensions between 'Western' and Islamic societies?

Have you thought about the potential ramifications or implications of reporting an association between - what could be interpreted on a very shallow reading of the paper by e.g. Daily Mail journalists and readers - between Islamic people and pathogenic disease?

Generally I have found ancient population genomicists in the past have tried to 'hide' between the 'purity' of the science method when reporting their results on ancestry to avoid having to consider what modern society may make of it. This may half work in older time periods but as we (rightly, and is a strength of this paper) move into less studied Historical periods, where society today feels a much closer 'connection', I think primarily ancient population genomicists groups who start to combine ancestry and cultures with microbes (that are often seen as 'dirty' or 'bad') should try to be more careful in how they report things for this reason.

For example, as I have mentioned above in my specific comments - when describing the *M. leprae* genome you don't really discuss the detection of this in the context of the ancestry this *specific* individual has (i.e. falling within a typical European area of the PCA space). It might be good to make this more explicit, as otherwise a (e.g.) Daily Mail journalist will shallowly skim read the paper as: 'Islamic invaders into Ibiza from the Arab countries brought Leprosy with to the Island!' When in reality, we do not have the sample size or temporal granularity to know where that specific lineage of Leprosy comes from, but for sure the individual carrying *M. leprae* is not from the first wave of invaders.

Anyway, like I said throw-away rambling comment, but maybe some food-for-thought?

We thank the reviewer for raising this important point. In the revised Discussion we now explicitly state that individual s.313, who carries *M. leprae*, has predominantly European-like ancestry and that our data do not allow us to attribute the presence of this pathogen to any specific group, migration event, or “wave” of conquest. We also emphasise that our sample size and temporal resolution are insufficient to draw conclusions about links between ancestry, culture, and infection, and we caution against extrapolating pathogen detections to modern social or religious identities. We will likewise be careful in any press communication to avoid simplistic narratives.

Reviewer #4 (Remarks to the Author):

The article is a very interesting attempt to generate and interpret ancient DNA, radiocarbon dating, and metagenomic data to explore ancestry and disease among medieval Islamic inhabitants of Ibiza (then part of Al-Andalus). It reconstructs population dynamics, admixture, and health during a period of strong North African and Mediterranean connectivity.

I will not comment on the genetic component that is not my expertise, only on the dating part that is.

The paper claims to present radiocarbon dates from 12 individuals (and not 13 as stated in the radiocarbon paragraph of the main text?). While direct dating is used as a major prior to the hypothesis the paper seeks to answer, there is minimal information regarding the C-14 ages. The S1 and S2 tables simply include Lab code, C-14 age and std error. No information on the chemical parameters of the extraction, no total collagen yield, C yield post combustion, to allow us to assess

the dates independently. As the text/ SI stands the information provided is inadequate and I would urge the authors to rectify this.

It seems to me that some of conclusions lie on the claim of two individuals being Almoravid arrivals (~1115 CE). In order to achieve further chronological precision, the authors combine radiocarbon dating with dietary isotopes to correct for marine reservoir effects. This is a relatively standard method, but admittedly imperfect. Since marine organisms have older apparent radiocarbon ages than terrestrial ones (because the ocean's carbon reservoir exchanges more slowly with the atmosphere), if a person's diet included significant marine protein, their tissues will reflect that older carbon — making their radiocarbon age artificially old.

To correct for this, the  $\delta^{13}\text{C}$  and  $\delta^{15}\text{N}$  values in bone collagen are used to estimate how much dietary protein came from marine vs. terrestrial sources (using a Bayesian mixing models, and then this correction is applied using local  $\Delta R$  values (the regional marine reservoir offset).

As the authors rightfully claim, it may be necessary for coastal contexts like Ibiza, to apply such as correction because marine food was part of the diet. The stable isotope approach provides a quantitative proxy for marine intake, and Bayesian models (as used in the paper) incorporate uncertainty. This is especially important, since two individuals had relatively high  $\delta^{15}\text{N}$  values (consistent with marine intake).

One of my concerns is that the sample size claiming marine intake is small (2). I don't quite understand why only these 2 were corrected for marine input, and not the others who may have more limited fish protein in their diet? The authors say these 2 have the more recent radiocarbon dates, but this is not quite true either. I may be missing something, but it'd be good to get an answer.

Would, for example, if all individuals are calibrated with a marine reservoir correction, have different age ranges? Could the authors provide with a calibration of all individuals, with and without marine diet correction?

Ultimately, Bayesian estimates should be seen as probabilistic estimates, not precise calendar anchors, and while the posterior calculation (80-95% postdating 1115CE) is high, the means of reaching to this estimate are somehow unclear to me. It may be the words in the main text, but the SI does not provide more answers.

Finally, a minor point, but both in the main text and SI, there is a sentence that "...from  $\delta^{13}\text{C}$  and  $\delta^{15}\text{N}$  values obtained from the same dentine collagen used for aDNA and radiocarbon analyses". The dentine collagen is collagen, ie protein, it is not used for aDNA analysis. The aDNA derives from the tooth dentine.

Overall, a table dedicated to the C-14 data only is rather important. There are specific guidelines in the literature what such a table should include. I would like to understand a bit more why and how the marine reservoir was applied, and to whom, and also see detailed tables with the calibration of these dates, with/ without a marine correction, as well as all relevant chemical apartments of extraction and combustion.

The paper is indeed a nice piece of interdisciplinary work, but some parts have been given unequal weight.

We thank the reviewer for the thoughtful and constructive comments on the radiocarbon component. We have revised the manuscript and SI accordingly, emphasizing transparent reporting, per-individual calibration (with and without marine correction), and probabilistic interpretation.

1. We corrected and confirmed that 12 individuals were directly  $^{14}\text{C}$  dated instead of 13. The main text now correctly states 12. See *page 6, lines 196-199*:

“Twelve of the thirteen individuals with sufficient DNA for genomic analyses were directly radiocarbon dated at the Tandem Laboratory, Uppsala University, to the 10th–12th centuries CE. Individual s.155, which yielded lower genomic coverage, was not dated (Supplementary Tables 1 and 2).”

2. We created a new radiocarbon figure (Supplementary Figure 1), and a table (Supplementary Table 2). This last includes for each individual: ID, archaeological element, tissue (tooth dentine collagen), lab code,  $^{14}\text{C}$  BP  $\pm 1\sigma$ ,  $\delta^{13}\text{C}$  and  $\delta^{15}\text{N}$ , calibration curve(s),  $\Delta\text{R}$ , software/version, and calibrated 68.3% / 95.4% HPDs. In line with reporting guidelines, Supplementary Table 2 also includes fields for atomic C:N provided by the Radiocarbon Dating Laboratory of Uppsala. We requested from the laboratory per-sample %C, %N, collagen yield (mg and %) and post-combustion/graphitization carbon yields. These values are not reported by the lab for this batch and remain unavailable despite request.

In methods, *page 27, lines 871-879*, we added this section: “Radiocarbon pre-treatment followed the standard Uppsala Tandem Laboratory (Uppsala, Sweden) protocol for bone collagen (report p5077). Surfaces were mechanically cleaned (scraping or sand-blasting), ultrasonically washed in boiled distilled water (pH 3), ground, and demineralized in 0.8 M HCl at  $\sim 10^\circ\text{C}$  for 30 min (fraction A). The insoluble residue was gelatinized in distilled water at pH 3 for 10 h at  $90^\circ\text{C}$ , producing an insoluble fraction C and a soluble fraction D. The soluble “fraction D”, containing most of the original collagen, was combusted to  $\text{CO}_2$ , graphitized using an Fe catalyst, and dated by AMS at Uppsala Tandem Laboratory.”

3. We have extended the MRE assessment to all 12 directly dated individuals, s117 and s197. For each person, we estimated marine dietary protein with *simmr* from dentine-collagen  $\delta^{13}\text{C}/\delta^{15}\text{N}$  using a clearly documented baseline (terrestrial herbivores and Mediterranean fish; Methods/SI). The marine-fish fraction posteriors span 17.7–34.7% across individuals (mean 21.8, median 20.0). Each  $^{14}\text{C}$  determination was then calibrated in OxCal v4.4.4 under a Terrestrial model (IntCal20), and a Mixed model (IntCal20-Marine20) with per-individual  $p_{\text{marine}}$  (mean  $\pm$  SD) and regional  $\Delta\text{R} = -42 \pm 30$   $^{14}\text{C}$  yr (Banyuls). Supplementary Table 2 now reports, for every individual, the calibrated HPDs both before and after MRE.

4. In agreement with Reviewer #4, Bayesian calibrations are probabilistic estimates, not fixed anchors. Under the individual mixed model: individual s.197 is now robustly post-1115 CE (HPD68.3% 1165–1280 CE); and individual s.117 straddles 1115 (HPD68.3% 1051–1264 CE): compatible with, but not definitive for, a post-1115 placement. Interestingly, most individuals also straddle 1115. However, individual s.313 remains earlier (HPD68.3% 896–1046 CE). Because the dated material is permanent molar dentine, formed decades before death, the date of death is expected to post-date the calibrated dentine window, which strengthens a post-1115 placement for s.197 and increases the plausibility for s.117, while we acknowledge residual uncertainty (especially at 95.4%).

*We have added the following third paragraph to the Supplementary Discussion:* “Under the mixed model, s.197 is robustly post-1115 CE (HPD68.3% = 1165–1280 CE), whereas s117 straddles the 1115 threshold (HPD68.3% = 1051–1264 CE): compatible with, but not conclusive for, a post-1115 placement. Most other individuals likewise straddle 1115 at 68.3%, and s.313 remains earlier (HPD68.3% = 896–1046 CE). Because the dated material is permanent molar dentine, formed decades before death in mature adulthood, dates of death would be expected to fall later than the dentine formation windows, which further supports a post-1115 placement for s.197 and increases the plausibility for s117, while we acknowledge remaining uncertainty at 95.4%. These results are overall consistent with the independent archaeological span proposed for the cemetery (ca. 950–1150 CE) (Dury et al. 2019), with one clearly early burial (s.313) and one clearly late burial (s.197), while the remainder fall broadly within the 11th–12th centuries when marine-diet uncertainty is considered.”

5. We have corrected the text to state that aDNA was extracted from tooth dentine powder, whereas radiocarbon and stable isotopes were measured on dentine collagen (protein).

*We have added the following introductory paragraph to the Supplementary Discussion:* “To assess potential marine offsets, we estimated marine dietary protein for all twelve (n = 12) directly dated individuals using the Bayesian mixing package *simmr* (R 4.5.0) (Govan et al. 2023). Mixtures were the dentine-collagen  $\delta^{13}\text{C}$  and  $\delta^{15}\text{N}$  values for each individual [...].”

We hope these revisions meet the reviewer’s requests for transparent methods, full per-individual reporting, consistent application of MRE, and appropriately cautious interpretation.

We thank all the reviewers for the time invested in improving our manuscript through their thoughtful and constructive comments. All requested revisions have now been completed, and we are grateful for the positive assessments of the revised version.

In response to the final round of comments, the following changes were implemented in the revised manuscript.

In the Metagenomic Analysis section, references to the “orange complex” were removed, as this concept is outdated and did not add interpretive value, and the organism is now described as a pathobiont.

The term “isolate” was removed or replaced throughout the manuscript with “ancient pathogen.”

In the Discussion, the phrasing “molecular evidence of infection to primary pathogens” was corrected to “by primary pathogens.”

In the Library Preparation and Sequencing section, the typographical error “specifics settings” was corrected to “specific settings.”

In the Capture Enrichment for *M. leprae* in S.313 section, the sequencing platform was corrected to Illumina NovaSeq 6000.

Finally, in the Code Availability section, we added the archived Zenodo DOI alongside the GitHub repository reference, ensuring long-term accessibility of all custom scripts used in this study.

We thank the reviewers again for their careful evaluation and constructive feedback, which have significantly strengthened the quality, clarity, and robustness of the manuscript.

On behalf of all co-authors,

The Authors